# ARF1 compartments direct cargo flow via maturation into recycling endosomes

Alexander Stockhammer[1,3], Petia Adarska[1,3], Vini Natalia [1], Anja Heuhsen[1], Antonia Klemt[1], Gresy Bregu[1], Shelly Harel[1], Carmen Rodilla-Ramirez[1], Carissa Spalt[1], Ece Özsoy[1], Paula Leupold [1], Alica Grindel[1], Eleanor Fox[1], Joy Orezimena Mejedo [1], Amin Zehtabian [1], Helge Ewers [1], Dmytro Puchkov [2], Volker Haucke [1,2] & Francesca Bottanelli [1] ✉

Cellular membrane homoeostasis is maintained via a tightly regulated membrane and cargo flow between organelles of the endocytic and secretory pathways. Adaptor protein complexes (APs), which are recruited to membranes by the small GTPase ARF1, facilitate cargo selection and incorporation into trafficking intermediates. According to the classical model, small vesicles would facilitate bi-directional long-range transport between the Golgi, endosomes and plasma membrane. Here we revisit the intracellular organization of the vesicular transport machinery using a combination of CRISPR-Cas9 gene editing, live-cell high temporal (fast confocal) or spatial (stimulated emission depletion) microscopy as well as correlative light and electron microscopy. We characterize tubulo-vesicular ARF1 compartments that harbour clathrin and different APs. Our findings reveal two functionally different classes of ARF1 compartments, each decorated by a different combination of APs. Perinuclear ARF1 compartments facilitate Golgi export of secretory cargo, while peripheral ARF1 compartments are involved in endocytic recycling downstream of early endosomes. Contrary to the classical model of long-range vesicle shuttling, we observe that ARF1 compartments shed ARF1 and mature into recycling endosomes. This maturation process is impaired in the absence of AP-1 and results in trafficking defects. Collectively, these data highlight a crucial role for ARF1 compartments in post-Golgi sorting.

Eucaryotic cells compartmentalize biochemical reactions within membrane bound organelles. Material exchange occurs either via transport carriers that bud from a donor and fuse with an acceptor compartment or via direct contact between membranes. The Golgi apparatus and endosomal organelles are crucial for maintaining cellular membrane homoeostasis as they coordinate trafficking highways at the intersection between exocytic and endocytic traffic. According to current models, communication between the Golgi and various endosomes is mediated by clathrin-coated vesicles[1,2]. Vesicle formation is orchestrated by a complex protein machinery, which ensures the specificity of cargo selection and guides trafficking intermediates to their correct destination. For this, key cargo adaptors such as the adaptor protein complex 1 (AP-1) are recruited to membranes by the ADP-ribosylation factor 1 (ARF1)[2–4], where they can recruit cargo and coat proteins such as clathrin to drive membrane curvature and vesicle formation[1,5].

When imaged as fusion proteins or with antibodies, clathrin and adaptors define punctate structures throughout the cytoplasm of the cell, supporting a vesicular model for cargo exchange between the

[1]Institute of Chemistry and Biochemistry, Freie Universität Berlin, Berlin, Germany. [2]Leibniz Forschungsinstitut für Molekulare Pharmakologie (FMP), Berlin, Germany. [3]These authors contributed equally: Alexander Stockhammer, Petia Adarska. ✉e-mail: francesca.bottanelli@fu-berlin.de

**Fig. 1 | ARF1 compartments are the major site of non-endocytic clathrin assembly. a**, Live-cell confocal and STED imaging of ARF1[EN]-Halo/SNAP-CLCa[EN] HeLa cells labelled with CA-JF$_{571}$ and BG-JFX$_{650}$ show association of clathrin to ARF1 compartments. **b**, Live-cell confocal imaging of ARF1[EN]-eGFP/AP2μ[EN]-SNAP/Halo-CLCa[EN] HeLa cells labelled with CA-JF$_{552}$ and BG-JFX$_{650}$, highlighting association of (i) non-endocytic clathrin with ARF1 compartments and (ii) endocytic clathrin with AP-2. **c**, Quantification of clathrin association with ARF1 and/or AP-2. In total, 10 cells from three independent experiments were analysed, replicates are shown in different colours and each small dot represents a single cell, s.d. error bars.

**d–f**, Time-lapse confocal spinning-disk imaging of ARF1[EN]-Halo/SNAP-CLCa[EN] HeLa cells labelled with CA-JF$_{552}$ and BG-JFX$_{650}$ (**d**), highlights movement of clathrin together with ARF1 compartments (**e**) and detachment of ARF1 compartments from the TGN together with clathrin (**f**). **g,h**, Clathrin is found at sites of fission of ARF1 compartments when they detach from the TGN (**g**) and in the cell periphery (**h**). Selected frames are shown; a video was taken with a frame rate of 5 frames per second. BG, benzylguanine (SNAP-tag substrate); CA, chloroalkane (HaloTag substrate). Scale bars, 10 μm (confocal overview), 5 μm (STED image in **a**) and 1 μm (crops). Source numerical data are available in source data.

Golgi, endosomes and the plasma membrane (PM)[5–8]. In contrast to shuttling vesicles and full-collapse fusion events, intracellular organelle communication through a kiss-and-run mechanism was proposed[9]. Various secretory and endocytic recycling cargoes have been found to transit through tubulo-vesicular compartments[10,11], some of which were shown to be decorated with clathrin and AP-1 (refs. [12–14]). Secretory cargo flow from the Golgi to the PM may occur via direct tubular carriers or clathrin-decorated tubules that deliver their content to endosomes via yet unknown mechanisms[13,15]. Tubulo-vesicular endosomes harbouring the adaptors AP-1 and AP-3 as well as clathrin have been identified using immuno-electron microscopy but have not been characterized in detail[16,17]. Cargo sorting from sorting endosomal compartments was also shown to occur via tubular domains responsible for cargo sequestration[18]. At early endosomes, cargo enrichment and tubulation of the membrane depend on sorting nexins and sorting complexes such as retromer, retriever and the CCC complex[19]. This process is important to sequester cargoes destined for recycling to the PM and the Golgi away from early endosomes, which subsequently mature into late endosomes and finally fuse with lysosomes to delivery their content[20–22].

Investigating the mechanisms and dynamics of proteins sorting out of the Golgi and within the endo-lysosomal system remains challenging, urging a combination of approaches aimed at investigating dynamic events and the underlying ultrastructure in near-physiological conditions. First, live-cell imaging is indispensable not only because of the transient and highly dynamic nature of trafficking events, but also because post-Golgi structures, such as tubulo-vesicular compartments, are lost upon fixation[23]. Second, over-expression of fusion proteins, which is often used for making proteins accessible for live-cell imaging, holds the potential for artefacts[24], as too high protein levels can affect protein localization and function, making endogenous tagging crucial. Third, visualization of sorting events in the perinuclear area where organelles are tightly packed has proven to be a difficult task, demanding super-resolution imaging techniques suitable for imaging live specimens[13,25,26]. Here, we utilize CRISPR-Cas9 technology to endogenously tag various sorting machinery components involved in post-Golgi trafficking with the self-labelling enzymes HaloTag and SNAP[27]. Employing various imaging techniques, such as three-dimensional (3D) correlative light electron microscopy, fast live-cell confocal and super-resolution stimulated emission depletion (STED) microscopy, we characterized multi-functional tubulo-vesicular ARF1 sorting compartment harbouring different adaptor proteins and clathrin. Trafficking assays and CRISPR-Cas9 mediated knock-outs point at a role for ARF1 compartments in endocytic recycling (downstream of early endosomes marked by Rab5) and post-Golgi secretory traffic. Notably, for both secretory and endocytic trafficking, cargo is observed first in ARF1 compartments and subsequently in recycling endosomes (REs). This prompted us to investigate how cargo is transferred from ARF1 compartments to REs. We show that ARF1 compartments undergo maturation into REs via a mechanism that depends on AP-1, as loss of AP-1 inhibits maturation and causes trafficking defects. Advancing previous thinking in this field, our findings suggest a model where cargo sorting is mediated by a dynamic tubular network that connects the trans-Golgi network (TGN) with endo-lysosomes and the PM[20,28,29].

## Results

### Clathrin is associated with ARF1 compartments

Previous studies have identified TGN-derived tubulo-vesicular compartments defined by the small GTPase ARF1 (ref. [23]). Notably, these compartments were found to harbour clathrin nanodomains. This observation raised the possibility that they could be hubs for clathrin-coated vesicle budding and define a tubulo-vesicular network responsible for post-Golgi cargo sorting. First, we recapitulated our initial observations[23] and we applied super-resolution live-cell STED microscopy on ARF1[EN]-Halo/SNAP-CLCa[EN] (endogenous) knock-in (KI) HeLa cells to highlight the close association of clathrin with ARF1 compartments (Fig. 1a). Additionally, we observed ARF1-positive clathrin compartments in various cell types (Extended Data Fig. 1a,b). To further characterize ARF1 compartments, we first wanted to test whether non-endocytic clathrin is exclusively associated with ARF1 compartments. For this, we created an ARF1[EN]-eGFP/Halo-CLCa[EN]/AP2μ[EN]-SNAP triple KI cell line (Fig. 1b). Quantification of clathrin association with either AP-2 (endocytic)[30] or ARF1 revealed that most of the non-endocytic and non-Golgi clathrin decorates ARF1-positive membranes (Fig. 1c). We took advantage of fast confocal live-cell imaging to get a better understanding of the dynamics of clathrin on ARF1 compartments (Fig. 1d and Supplementary Video 1). Notably, we could not visualize any clathrin-coated vesicles budding from ARF1 compartments. Instead, we observed clathrin clusters translocating together with the closely associated membrane (Fig. 1e). Notably, when we looked at ARF1 compartments emerging from the TGN, we found clathrin associated with the detaching tubule (Fig. 1f). Without visualization of the underlying ARF1 membrane, these events could have easily been mistaken for a clathrin-coated vesicle moving within the cytoplasm or budding from the TGN. Clathrin localized at the fission site on ARF1 compartments, suggesting that clathrin and associated machinery may be responsible for the recruitment of fission factors. We analysed more than 100 fission events and found that clathrin was present at >90% of the fission sites (Fig. 1g,h and Extended Data Fig. 1c).

To better understand ARF1 compartment membrane organization and clathrin association, we used 3D correlative light electron microscopy (3D CLEM). ARF1[EN]-Halo/SNAP-CLCa[EN] KI cells were labelled with cell-permeable dyes and imaged post-fixation by scanning confocal microscopy (Fig. 2a), embedded, and a small region of interest was visualized using focused ion beam (FIB)-scanning electron microscopy (SEM) (Fig. 2b). Alignment of confocal and FIB-SEM images enabled the identification of ARF1 compartments at an isotropic resolution of 7 nm (Fig. 2c(i)–(iii) and Supplementary Video 2). The underlying tubulo-vesicular compartments displayed a pearled morphology that is reminiscent of the morphology of the ER–Golgi intermediate tubular compartments that mediate ER export[31]. In contrast to the clathrin-coated vesicular structures, which have a confined diameter between 70 and 90 nm, the diameter of the non-clathrin-coated ARF1 compartment varied between 20 nm and 180 nm (Fig. 2d). Clathrin-positive membranes are preferably located at sites with high

**Fig. 2 | Tubulo-vesicular nature of ARF1 compartments revealed by 3D CLEM.** **a**, Slice of a confocal z-stack of ARF1[EN]-Halo/SNAP-CLCa[EN] HeLa cell labelled with CA-JF[552] and BG-JFX[650] that was chosen for CLEM. The area that was imaged with FIB-SEM is highlighted with a yellow outline. Segmented ARF1 compartments (i–iii) are shown. **b**, Overlay of the 3D projection of the confocal stack and FIB-SEM image (green box). FIB-SEM image was obtained with 7 nm isotropic resolution. **c**, Segmentation of individual ARF1 compartments with clathrin. Shown are 2–3 exemplary slices of the FIB-SEM image, the FIB-SEM image with outlines from the segmented area, overlays of fluorescence of ARF1 and clathrin with the FIB-SEM image, a representative slice of the confocal image of the ARF1 compartments and the 3D rendering from the ARF1 compartments. **d**, Diagram showing variation in tubule diameter of the ARF1 compartment and the clathrin-coated areas. Data for the diagram was obtained from the three ARF1 compartments shown in **c** that were measured at different parts of the tubule (100 percentile box plot, tubule: 17 nm minimum, 61 nm centre, 172 nm maximum; clathrin-coated: 66 nm minimum, 78 nm centre, 87 nm maximum). **e**, Single slices (7 nm increments) of the FIB-SEM dataset, including segmentation, highlight the connection of clathrin-coated and non-coated part of the ARF1 compartment (arrow highlights neck of non-clathrin-coated and clathrin-coated ARF1 compartment). Scale bars, 10 μm (overview in **a**) and 500 nm (crops in **c**). Source numerical data are available in source data.

intrinsic curvature and are directly connected to the ARF1 compartment via a membranous neck (Fig. 2e). In summary, these data identify ARF1 compartments as the major site of clathrin recruitment. This finding motivated us to further characterize the identity and function of these compartments.

## AP-1 and AP-3 localize to segregated nanodomains on ARF1 compartments

Post-Golgi protein sorting relies on the membrane recruitment of different cargo adaptors, such as AP-1 or AP-3. Both adaptors were previously reported to localize to the TGN[5,6,14] and endosomal membranes[12,17,32],

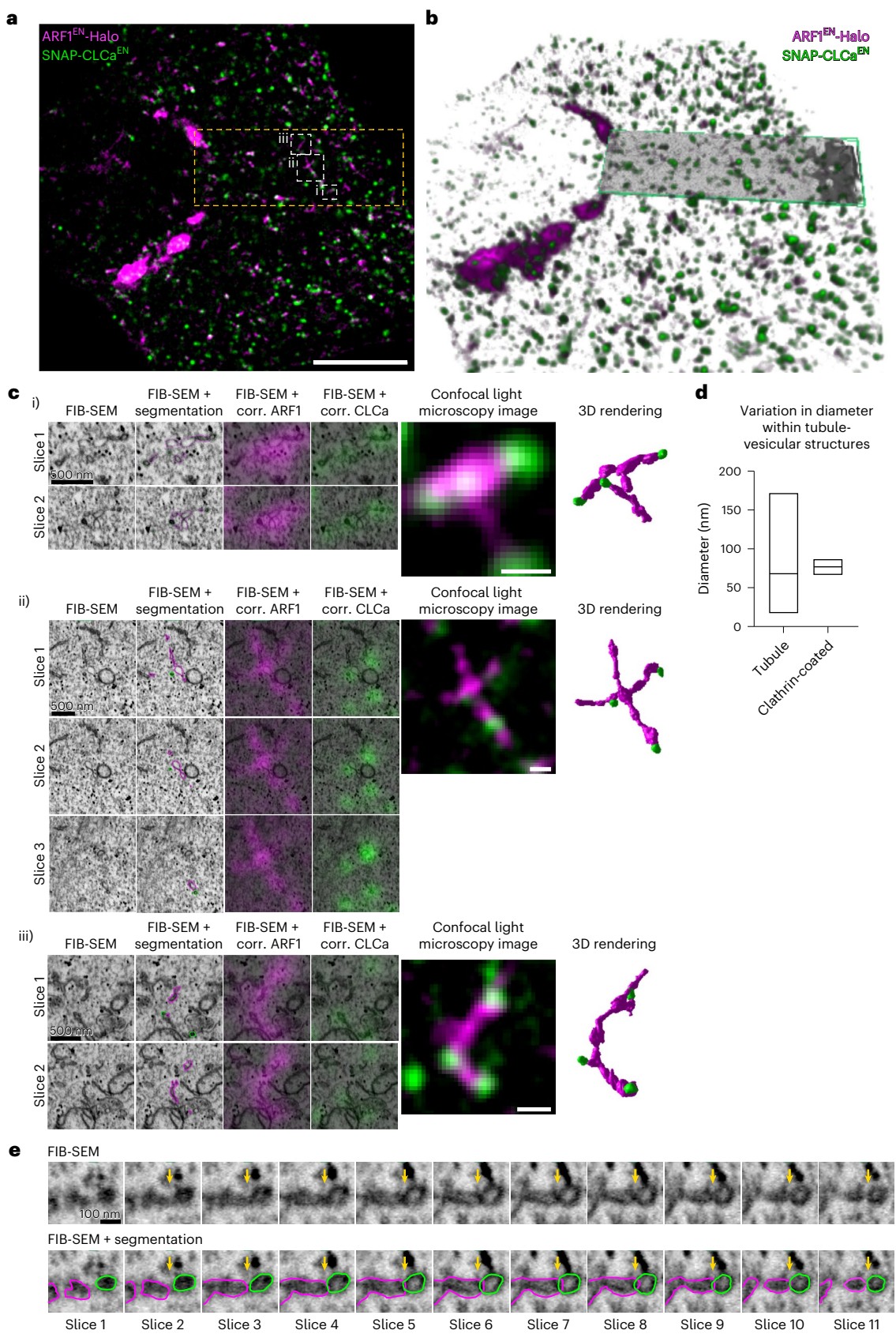

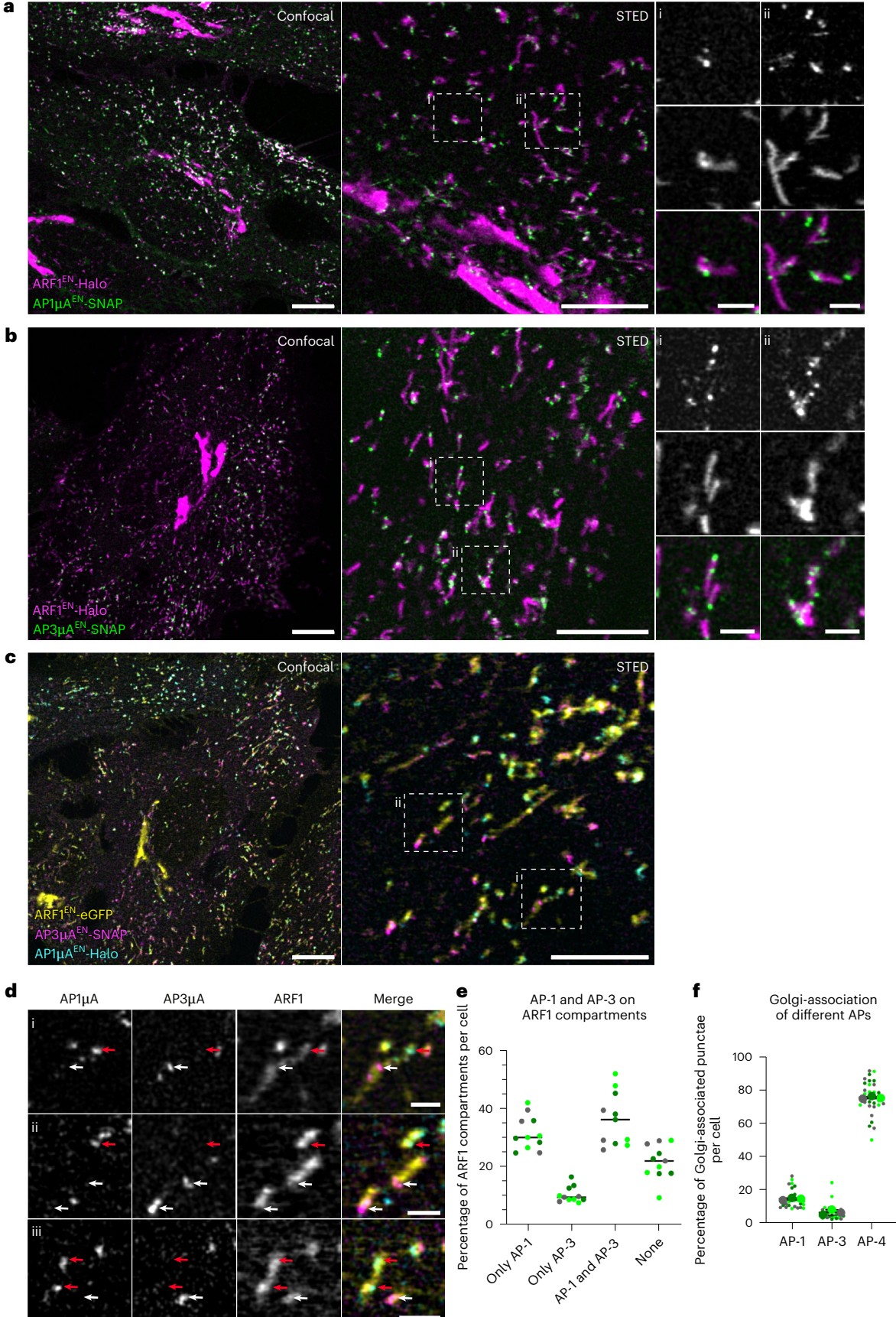

**Fig. 3 | Adaptor protein complexes AP-1 and AP-3 define segregated nanodomains on ARF1 compartments. a,b,** Live-cell confocal and STED imaging of ARF1$^{EN}$-Halo/AP1μA$^{EN}$-SNAP HeLa cells (**a**) and ARF1$^{EN}$-Halo/AP3μA$^{EN}$-SNAP HeLa cells (**b**) labelled with CA-JF$_{571}$ and BG-JFX$_{650}$ show association of AP-1 and AP-3 to ARF1 compartments. **c,** Live-cell confocal and STED imaging (two-colour STED imaging with ARF1$^{EN}$-eGFP imaged in confocal mode) of ARF1$^{EN}$-eGFP/AP3μA$^{EN}$-SNAP/AP1μA$^{EN}$-Halo HeLa cells labelled with CA-JF$_{571}$ and BG-JFX$_{650}$ show that AP-1 and AP-3 localize to segregated nanodomains on ARF1 compartments. **d,** (i) Crops highlight that AP-1 (red arrows) and AP-3 (white arrows) localize to segregated nanodomains on the same compartment.

(ii–iii) In addition, ARF1 compartments harbouring either AP-1 or AP-3 are found. **e,** Quantification of the percentage of ARF1 compartments with specific adaptor identity per cell. In total 11 cells from three independent experiments were analysed, replicates are shown in different colours each dot representing a single cell. **f,** Quantification of Golgi-associated puncta positive for AP-1, AP-3 and AP-4. In total 30 cells from three independent experiments were analysed for each condition, replicates are shown in different colours and each small dot represents a single cell of the replicate. Scale bars, 10 μm (confocal overview), 5 μm (STED images) and 1 μm (STED crops). Source numerical data are available in source data.

where they coordinate bi-directional transport between the TGN and endosomes (AP-1)[1,33,34] or transport to late endosomes and melanosomes (AP-3)[17,32]. As both adaptors are recruited by ARF1 (refs. 3,4), we wondered whether they are present on ARF1 compartments. To test the localization of AP-1 and AP-3, we introduced a SNAP-tag to the C terminus of their μA-subunit in ARF1$^{EN}$-Halo KI cells and created double KI cell lines (Fig. 3a,b). Of note, as observed for clathrin (Fig. 1a), live-cell STED highlighted nanodomains of both AP-1 and AP-3 on ARF1 compartments (Fig. 3a,b). We observed the same localization pattern when we endogenously tagged the large AP1γ1 or AP3δ1 subunit (Extended Data Fig. 2a,b) suggesting that the placement of the tag within the complex does not impact AP localization. Simultaneous tagging and visualization of medium and large subunits of AP complexes showed both subunits to colocalize within the same nanodomains, indicating that AP complexes still form when one or more subunits are tagged (Extended Data Fig. 2c,d). In addition, AP-1-dependent clathrin recruitment was unaffected by Halo-tagging of the μ-subunit (Extended Data Fig. 2e,f), indicating that interaction of AP-1 with accessory proteins is unaffected by addition of the imaging tags.

Seeing both adaptors on ARF1 compartments, we questioned whether these are multi-functional sorting endosomes or distinct compartments defined by different adaptors. To understand the nature and function of the compartments, we created an ARF1$^{EN}$-eGFP/AP1μA$^{EN}$-Halo/AP3μA$^{EN}$-SNAP triple KI cell line (Fig. 3c). Live-cell confocal and STED imaging revealed that both adaptors localize to segregated nanodomains on ARF1 compartments. The most abundant class of ARF1 compartments is decorated with AP-1 and AP-3 (~38%) (Fig. 3c,d(i),e). However, we could additionally observe ARF1 compartments that only supported AP-1 (~30%) or AP-3 recruitment (~10%) (Fig. 3d(ii)–(iii),e). ARF1 compartments that were seen in the perinuclear area and emerging from the Golgi were only positive for AP-1. In contrast, tubules observed in the cell periphery were found to be positive for both AP-1 and AP-3 (Fig. 3a–c). This suggests that functionally distinct populations of ARF1 compartments may co-exist. About 20% of all ARF1 compartments did not harbour any AP-1 or AP-3, in agreement with a role for ARF1 tubules in retrograde Golgi-to-ER transport[13,23]. Both AP-1 and AP-3 were often shown to localize predominantly to the TGN/Golgi area in fixed cells[5,6]. However, in living gene-edited cells, we found that only around 16% of total AP-1 and 6% of total AP-3 punctae were confined to the perinuclear area

(Fig. 3f). Fixation is known to disrupt tubulo-vesicular cytoplasmic membranes[23] possibly leading to an overestimation of the fraction of Golgi-associated adaptors. Additionally, while a population of AP-1 punctae associated with the Golgi was very prominent, AP-3 was mainly excluded from the Golgi area (Fig. 3f). AP-4, the other adaptor protein complex recruited by ARF1 (refs. 23,35), was predominantly associated with the Golgi (Fig. 3f and Extended Data Fig. 2g). This suggests that ARF1 compartments serve as the main hub for AP-1 and AP-3-dependent sorting and might fulfil distinct intracellular functions based on the adaptors present.

Next, we wanted to investigate whether both AP-1 and AP-3 can recruit clathrin in living cells. Clathrin binding to AP-1 is well established[36]. Although AP-3 was shown to bind clathrin in vitro[6], the interaction of AP-3 with clathrin in mammalian cells is debated[17,37–39]. To resolve this issue, we created triple KI cell lines that allowed simultaneous visualization of ARF1, clathrin and AP-1 (ARF1$^{EN}$-eGFP/AP1μA$^{EN}$-SNAP/Halo-CLCa$^{EN}$) or AP-3 (ARF1$^{EN}$-eGFP/AP3μA$^{EN}$-SNAP/Halo-CLCa$^{EN}$) (Fig. 4a–d). Live-cell STED imaging of CLCa and AP1μA revealed a perfect colocalization of AP-1 and clathrin on ARF1 compartments (Fig. 4b). In contrast, AP-3 and clathrin localized to segregated nanodomains (Fig. 4d). Colocalization analysis shows a strong overlap between AP-1 and clathrin, whereas AP-3 and clathrin colocalization was like the negative control (AP-3 versus AP-1; Fig. 4e). To further elucidate the role of AP-1 on ARF1 compartments, we created a CRISPR-Cas9 AP1μA knockout (KO) in the ARF1$^{EN}$-Halo/SNAP-CLCa$^{EN}$ KI cell line. Notably, loss of AP1μA led to the formation of elongated ARF1 compartments, suggesting a defect in fission (Fig. 4f). Clathrin recruitment to peripheral ARF1 compartments, but not to the Golgi, was impaired (Fig. 4g). Hence, AP-1 is responsible for the recruitment of clathrin and fission machinery to ARF1 compartments. Residual clathrin puncta were observed on ARF1 compartments (arrows in Fig. 4f) suggesting recruitment via other clathrin adaptors such as GGAs (Golgi-localized, γ-ear containing, ADP-ribosylation factor binding)[10].

To conclude, ARF1 compartments harbour AP-1 and AP-3 nanodomains, with clathrin being recruited exclusively to AP-1 nanodomains. The different classes of tubules, harbouring different classes of adaptors may have a role in channelling different cargoes from ARF1 compartments into segregated downstream pathways. Beyond its role in cargo selection, AP-1 might also be required for the recruitment of fission factors.

**Fig. 4 | AP-1 recruits clathrin to ARF1 compartments and promotes their fission. a,** Live-cell confocal and STED imaging (two-colour STED imaging with ARF1$^{EN}$-eGFP imaged in confocal mode) of ARF1$^{EN}$-eGFP/AP1μA$^{EN}$-SNAP/Halo-CLCa$^{EN}$ HeLa cells labelled with CA-JF$_{571}$ and BG-JFX$_{650}$ highlight that clathrin and AP-1 are recruited to the same nanodomains on ARF1 compartments. **b,** (i–iii) Examples of different compartments and line profiles showing perfect colocalization of AP-1 with clathrin. **c,** The same analysis on ARF1$^{EN}$-eGFP/AP3μA$^{EN}$-SNAP/Halo-CLCa$^{EN}$ HeLa cells labelled with CA-JF$_{571}$ and BG-JFX$_{650}$ shows that clathrin and AP-3 do not colocalize on ARF1 compartments. **d,** (i–iii) Examples of different compartments and line profiles. **e,** Colocalization analysis using the Manders coefficient shows high correlation of AP-1 with clathrin but low correlation of AP-3 with clathrin, comparable with the correlation of AP-1 with AP-3. In total 30 cells from three independent experiments were analysed, replicates

are shown in different colours each dot representing a single cell. **f,** Live-cell confocal imaging shows that AP1μA KO in ARF1$^{EN}$-Halo/SNAP-CLCa$^{EN}$ HeLa cells labelled with CA-JF$_{552}$ and BG-JFX$_{650}$ leads to the formation of aberrant long tubular ARF1 compartments. Clathrin recruitment to the long tubules is reduced but not completely abolished (yellow arrows highlight clathrin nanodomains in crops (i,ii)). **g,** Quantification of fluorescence intensity of clathrin punctae on peripheral ARF1 compartments normalized to the intensity of Golgi-associated clathrin punctae in control and AP1μA KO HeLa cells. In total 27 cells from three independent experiments were analysed for each condition, replicates are shown in different colours and each dot represents a single cell of the replicate. $P$ value of nested two-sided $t$-test is 0.0064; **$P$ < 0.01. Scale bars, 10 μm (confocal overview), 5 μm and 1 μm (crops). Source numerical data and unprocessed blots are available in source data. WT, wild type; norm., normalized.

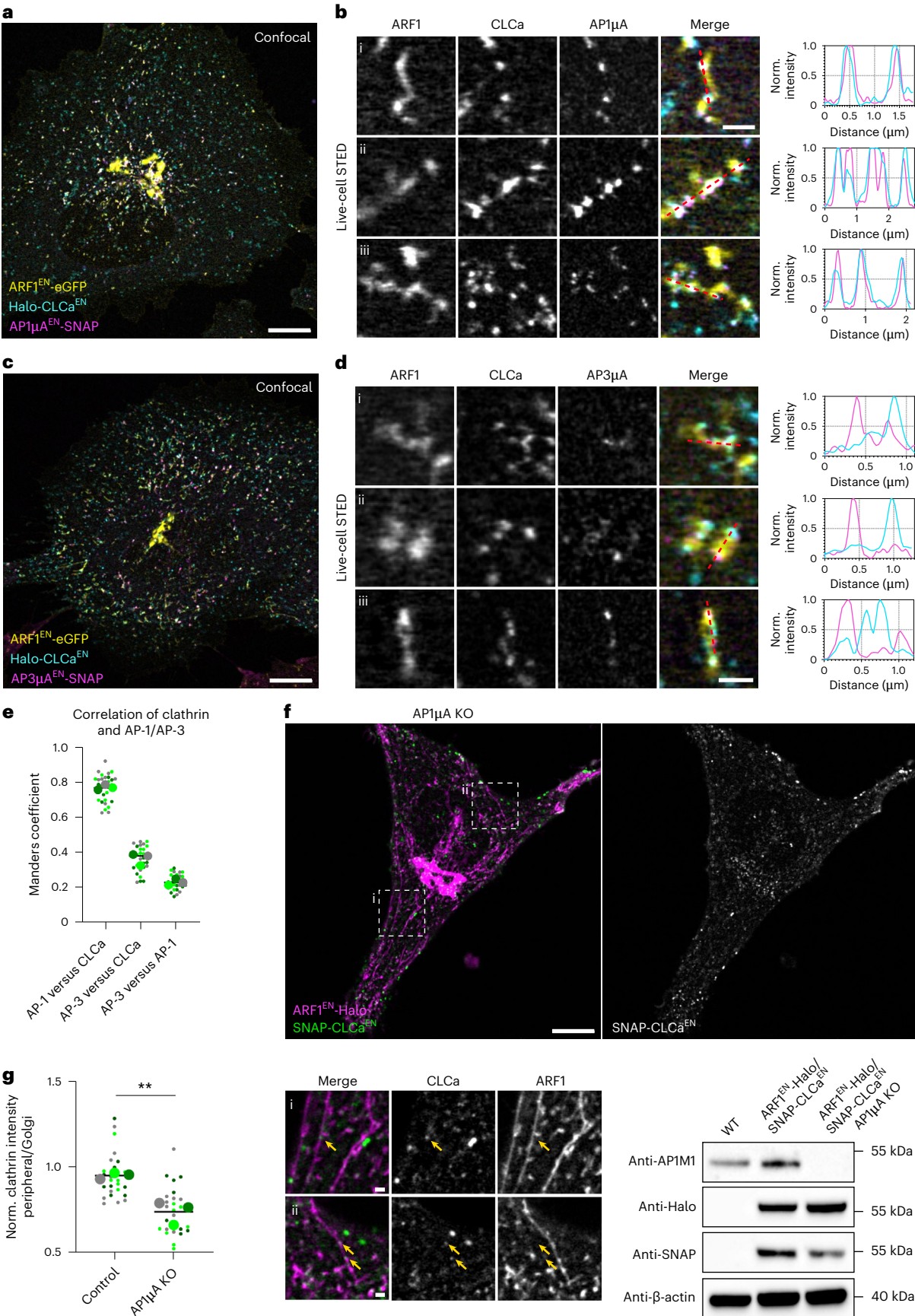

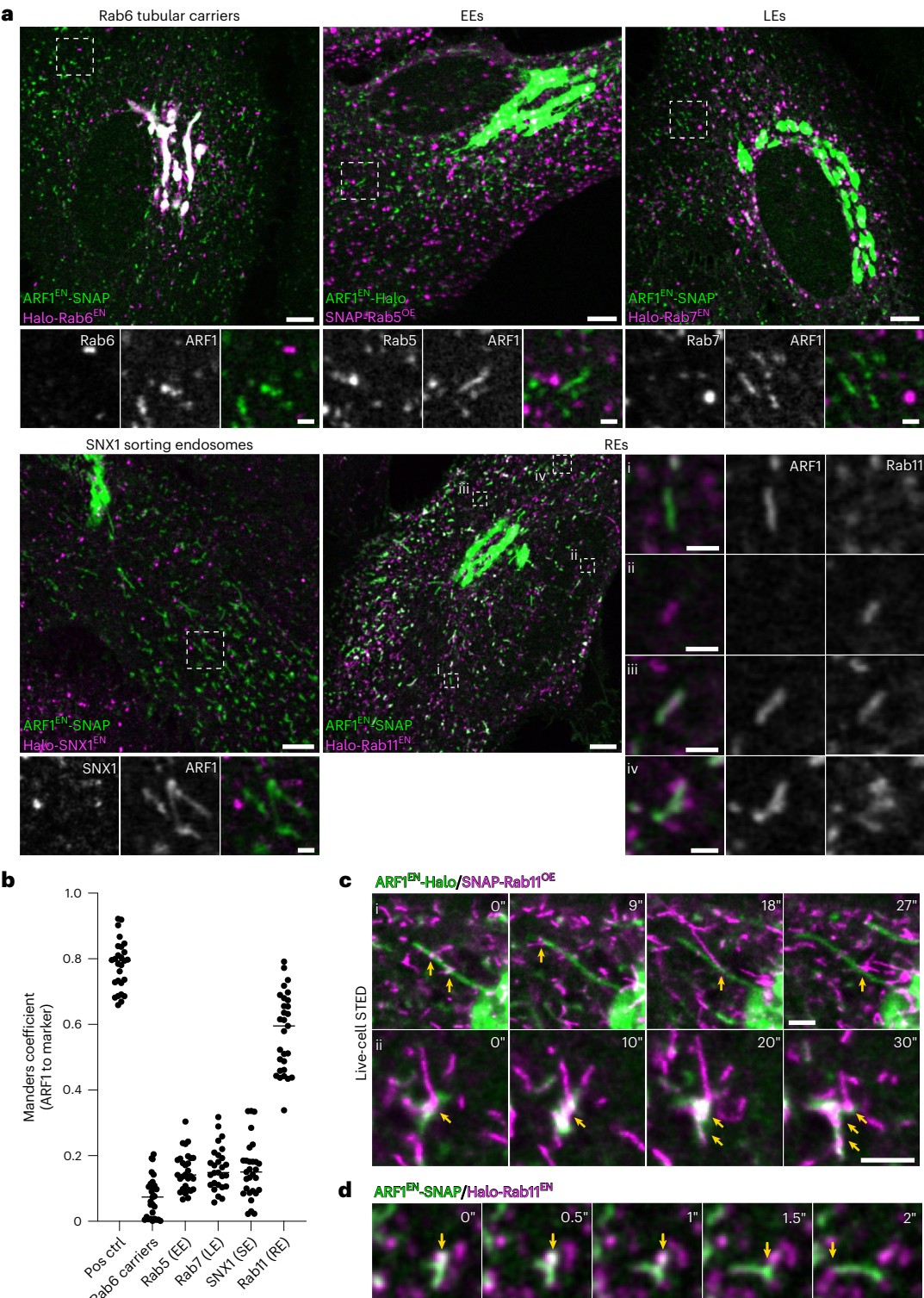

**Fig. 5 | ARF1 compartments are sorting compartments with a partial colocalization with the RE marker Rab11. a**, Live-cell confocal microscopy of ARF1$^{EN}$-SNAP/Halo-Rab6$^{EN}$ (Rab6 secretory carriers), ARF1$^{EN}$-SNAP/Halo-Rab7$^{EN}$ (LEs), ARF1$^{EN}$-SNAP/Halo-SNX1$^{EN}$ (SNX1 SEs), ARF1$^{EN}$-SNAP/Halo-Rab11$^{EN}$ (REs) HeLa cells labelled with BG-JF$_{552}$ and CA-JFX$_{650}$ (ARF1 + Rab6/7/11) or BG-JFX$_{650}$ and CA-JF$_{571}$ (ARF1 + SNX1). ARF1$^{EN}$-Halo HeLa cells transiently expressing SNAP-Rab5$^{OE}$ (EEs) were labelled with BG-JFX$_{650}$ and CA-JF$_{552}$. Confocal imaging highlights ARF1 compartments devoid of markers for different endosomal compartments and a partial overlap of ARF1 compartments with the RE marker Rab11. In particular we could observe compartments positive (i) for ARF1 only (ii), for Rab11 only (iii), for both ARF1 and Rab11 and (iv) ARF1 compartments in close proximity to REs. **b**, Colocalization analysis using the Manders coefficient shows higher correlation of ARF1 compartments with REs compared with other tested markers. At least 27 cells from three independent experiments were analysed for each condition, each dot represents a single cell. **c**, STED microscopy of ARF1$^{EN}$-Halo HeLa cells transiently expressing SNAP-Rab11$^{OE}$ labelled with CA-JF$_{571}$ and BG-JFX$_{650}$ show the dynamic nature of the interaction of ARF1 compartments with REs (sites of interaction indicated with yellow arrows) at the TGN (i) or in the cell periphery (ii). **d**, Time-lapse confocal spinning-disk imaging in ARF1$^{EN}$-SNAP/Halo-Rab11$^{EN}$ HeLa cells labelled with CA-JFX$_{650}$ and BG-JF$_{552}$ shows ARF1 compartments transiently interacting with different REs (sites of interaction indicated with yellow arrows). Scale bars, 5 μm (overview) and 1 μm (crops, time-lapse). Source numerical data are available in source data.

## ARF1 compartments shed ARF1 to mature into recycling endosomes

We found ARF1 compartments to harbour different AP complexes. As these adaptors were described to associate with different endosomal compartments[12,17,33], we set out to visualize ARF1 in combination with known post-Golgi and endosomal membrane markers (Fig. 5). First, we created a double KI cell line expressing edited ARF1 and Rab6, a membrane marker for tubular Golgi-derived carriers mediating direct transport to the PM[40], and found ARF1 and Rab6 to define different post-Golgi carriers in the cell periphery (Fig. 5a). The high degree of colocalization at the Golgi prompted us to test the extent of overlap between Golgi-derived ARF1 compartments and Rab6 carriers (Extended Data Fig. 3). Live-cell STED and confocal time-lapses show that about half of the Golgi-derived compartments are positive for ARF1 and Rab6, while the remaining half are Rab6-only carriers devoid of AP-1 (Extended Data Fig. 3a–c). This suggests the presence of functionally distinct classes of Rab6 carriers. Rab6-only tubules may be the direct Golgi-to-PM carriers, which have been reported previously[41]. Next, to test whether ARF1 compartments are defined by early, late, sorting or recycling endosomal markers, we created double KI cell lines expressing gene-edited ARF1 and Rab7 (late endosome; LE), SNX1 (sorting endosome, SE) and Rab11 (RE) or transiently overexpressed Rab5 (early endosome; EE). Live-cell confocal imaging revealed that ARF1 compartments are not defined by early, late or sorting endosomal markers, but they displayed a partial colocalization with the RE marker Rab11 (Fig. 5a,b). Notably, as reflected by the colocalization analysis, we observed ARF1 compartments positive for ARF1 only (Fig. 5a(i)), RE structures positive for Rab11 only (Fig. 5a(ii)) and ARF1 compartments also positive for Rab11 (Fig. 5a(iii)). Accordingly, AP-1 and clathrin are seen on ARF1 compartments and on a subpopulation of double-positive Rab11-ARF1 compartments (Extended Data Fig. 4a,b). Fast confocal imaging showed co-translocation of AP-1 with ARF1 compartments only (Extended Data Fig. 4c) and quantification of the overlap between ARF1/AP-1/Rab11 revealed a stronger association of AP-1 to ARF1 compartments compared with REs (Extended Data Fig. 4d). In addition, we observed many ARF1 compartments that seem to localize closely to REs in confocal microscopy images (Fig. 5a(iv)). To further characterize the dynamics of these various classes of ARF1 compartments and REs, we started out by surveying the dynamics of ARF1 compartments closely interacting with REs with higher resolution live-cell STED microscopy. Both peripheral and Golgi-derived compartments seemed to be closely associated and interacting with REs (Fig. 5c, Extended Data Fig. 5a,b and Supplementary Videos 3 and 4). Fast confocal microscopy and live-cell STED microscopy could highlight transient interaction of the same ARF1 compartment with different REs (Fig. 5d and Extended Data Fig. 5a,b). Live-cell confocal microscopy revealed that AP-1 is located at the interface of ARF1 compartments and REs in cases where both compartments were found to interact (Extended Data Fig. 5c).

Yet, we were intrigued by the fact that some ARF1 compartments were also defined by Rab11 (Fig. 5a(iii)), raising the possibility that ARF1 compartments may mature into REs. To test this, we endogenously tagged ARF1 with the highly photostable monomeric fluorescent protein StayGold (mStayGold)[42], allowing long-term imaging of ARF1 and Halo-Rab11 dynamics (Fig. 6). We focused on peripheral ARF1 compartments, as the less crowded cell periphery allows for better tracking of structures over time. Notably, we observed that double-positive compartments shed their ARF1 coat and acquired more Rab11 over time (Fig. 6a–c and Supplementary Video 5). We observed ~1–2 maturation events per ~200 μm$^2$ of cell area in ~4 min acquisition time. While increase of Rab11 coating the surface of the compartment was gradual, complete shedding of ARF1 occurred over a short period of ~7–9 s (Fig. 6b). Noticeably, the highly curved ends of the ARF1 compartment were the last parts to uncoat (Fig. 6, yellow arrows), suggesting that ARF1 molecules may be shielded by the presence of the AP-1/clathrin

which were observed to preferentially localize to regions of high curvature. These data suggest that ARF1 compartments mature into REs to potentially deliver their content to the PM.

## ARF1 compartments mediate endocytic recycling and secretory traffic via maturing into recycling endosomes

So far, we have shown that ARF1-positive compartments are uncharacterized organelles that control clathrin and adaptor-dependent post-Golgi trafficking. To determine the exact role of ARF1 compartments in sorting, we investigated the trafficking of endocytic and secretory cargoes in different CRISPR-Cas9 KI cell lines. We started by investigating the role of Golgi-derived perinuclear ARF1 compartments in the export of cargoes from the Golgi (Fig. 7). ARF1 compartments have been shown to mediate Golgi export of vesicular stomatitis virus glycoprotein (VSV-G)[13]. Notably, they were not observed fusing with the PM suggesting that another sorting step is required for the final delivery of VSV-G to the PM. We employed the retention using selective hooks (RUSH) system to release a pulse of various secretory cargoes from the ER[43]. The RUSH system consists of a reporter fused to a fluorescent protein or SNAP-tag and to streptavidin-binding protein (SBP). The reporter is retained in the ER by a streptavidin hook. Upon biotin addition, the reporter is released from the ER and accumulates at the Golgi (~15 min) before reaching the PM (~30 min). We examined the secretory transport of five different RUSH reporters: transferrin receptor (TfR, transmembrane protein), a LAMP1 variant lacking the endocytic motif in its cytoplasmic tail and thus fails to be endocytosed after deposition to the PM (LAMP1Δ, transmembrane protein)[44], TNF (transmembrane protein), VSV-G (transmembrane protein) and a soluble SNAP reporter (sSNAP) with live-cell confocal microscopy in the various gene-edited cell lines. We found that all reporter cargoes exited the Golgi in ARF1/clathrin-positive compartments that did not harbour AP-3 (Fig. 7a,b, Extended Data Fig. 6a–c and Supplementary Video 6). Secretory compartments detached and moved away from the Golgi (Extended Data Fig. 6d). Using two exemplary RUSH cargoes, we quantified the fraction of the tubulo-vesicular carriers leaving the Golgi that were positive for ARF1. We found that ~90% of RUSH cargo-containing tubules were decorated by ARF1 (Fig. 7c).

REs have been shown to serve as an intermediate sorting station for secretory cargo exiting the Golgi en route to the PM[45,46]. As ARF1 compartments do not fuse with the PM[13] but shed ARF1 to mature into REs (Fig. 6), we tested whether secretory RUSH cargoes transit through REs downstream of ARF1 compartments. We followed the transport of RUSH cargoes in Halo-Rab11[EN] KI cells and found them to colocalize with REs (Fig. 7d,e and Extended Data Fig. 6e). By quantifying the kinetics of TfR-RUSH transport to ARF1 compartments and REs, we found that the cargo first fills ARF1 compartments before transitioning to REs (Fig. 7f). These results suggest a model in which ARF1 compartments containing secretory cargo emerge from the Golgi and over time mature into REs before being able to deliver their content to the PM. Furthermore, upon AP1μA KO, both TfR- and TNF-RUSH were retained in long-aberrant perinuclear tubules that were positive for both ARF1 and Rab11 (Fig. 7g and Extended Data Fig. 6f). Additionally, cargo exit from the TGN was delayed in AP1μA KO cells (Fig. 7h). Altogether, this suggests that AP-1 may be required for secretory ARF1 compartments to shed ARF1 and mature into REs.

As secretory cargoes were almost exclusively transported by perinuclear ARF1 compartments, we wondered about the function of peripheral ARF1 compartments. The importance of AP-1 for transferrin (Tfn) recycling[47] motivated us to investigate the role of ARF1 compartments in endocytic recycling (Fig. 8). For this, we performed fluorescent Tfn uptake experiments in different gene-edited cell lines. After internalization, Tfn could be detected in peripheral ARF1 compartments (Fig. 8a) defined by AP-1 and AP-3 nanodomains (Extended Data Fig. 7a–c). These peripheral ARF1 compartments were functionally segregated from the perinuclear secretory ARF1 compartments

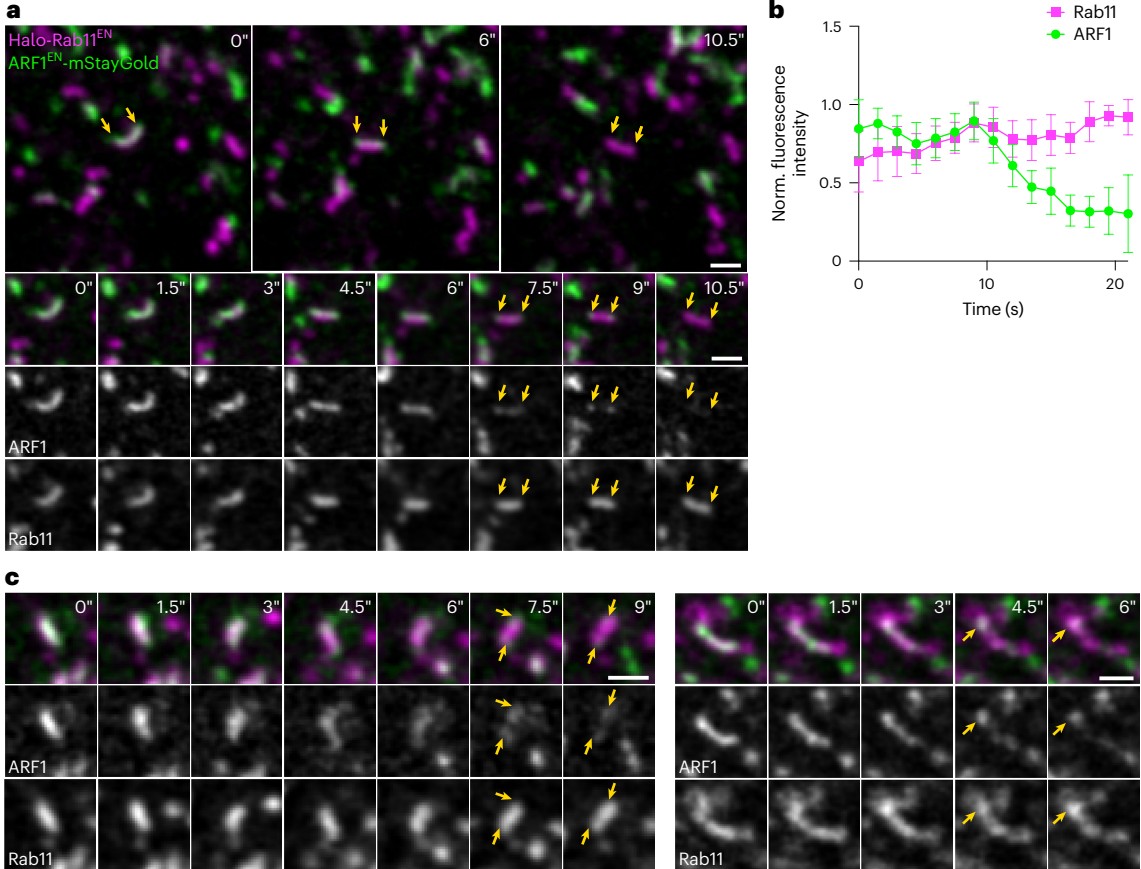

**Fig. 6 | ARF1 compartments mature into Rab11-positive REs. a**, Live-cell confocal microscopy of ARF1[EN]-mStayGold/Halo-Rab11[EN] HeLa cells labelled with CA-JFX$_{650}$. ARF1 compartments are seen to shed ARF1 from their membrane and mature into REs. **b**, Normalized fluorescence intensity of ARF1 and Rab11 signal on maturing endosomal compartments. ARF1 is shed over a short period of 7–9 s. Eleven videos from five different independent biological replicates were analysed, graph shows mean values, s.d. error bars. **c**, Additional examples of maturation events. Yellow arrows indicate the curved ends of the tubules which uncoat last. Scale bars, 1 μm (crops, time-lapse). Source numerical data are available in source data.

(Fig. 8b). Quantification of the colocalization of fluorescent Tfn with markers for EE (Rab5), ARF1 compartments and REs (Rab11) showed that internalized Tfn first localizes to Rab5-positive EEs, then to ARF1 compartments and REs (Fig. 8c and Extended Data Fig. 7d). At 10 min post-internalization Tfn empties out from ARF1 compartments but continues to fill REs (Fig. 8c). Dynamic live-cell microscopy further demonstrates that ARF1 compartments are not simply sorting sub-domains of Rab5-positive EEs or derive from EEs (Extended Data Fig. 7e).

We then wanted to test whether flow of Tfn cargo is dependent on the maturation of ARF1 compartments into REs. For this, we applied fluorescent Tfn to Rab11 and ARF1 double KI cells and followed the fate of an ARF1/Rab11 compartment filled with Tfn. Over the time of ~9 s ARF1 dissociated from the membrane of the Tfn filled compartment resulting in the formation of a Rab11-only positive RE (Fig. 8d and Supplementary Video 7). Loss of AP1μA delayed Tfn export from ARF1 compartments (Fig. 8e). Of note, ARF1 compartments involved in Tfn recycling did not seem elongated as a result of the AP1μA KO (Extended Data Fig. 7f), suggesting that fission defects caused by the loss of AP-1 are limited to perinuclear secretory ARF1 compartments. Furthermore, we observed that AP-1 stays associated with ARF1 compartments during the maturation process and dissociates from the membrane once maturation is complete (Fig. 8f and Supplementary Video 8).

Collectively, these data suggest that ARF1 compartments mediate endocytic recycling via maturation into REs, a process that depends on AP-1.

## Discussion

Clathrin-coated vesicles are thought to mediate cargo exchange between the TGN and different endosomal compartments as small punctate structures are commonly seen travelling around the cytoplasm of cells[6–8]. Endogenous tagging of ARF1 and clathrin together with CLEM allowed us to visualize the membrane underlying clathrin-positive vesicle-like structures and revealed that the vast majority of non-endocytic clathrin is associated with a tubulo-vesicular network of ARF1 compartments (Figs. 1 and 2) which direct cargo flow along the secretory and endocytic recycling routes though shedding of ARF1 and maturation into REs (Fig. 8g). Previous studies have shown clathrin as well as AP-1 and AP-3 on tubular endosomal structures; however the identity and the role of these compartments were never assessed further[14,16,17]. Here, we identify functionally distinct ARF1 compartments: perinuclear secretory compartments harbouring AP-1 and peripheral endocytic recycling compartments harbouring both AP-1 and AP-3 (Fig. 3). Clathrin translocates and remains associated with ARF1 compartments, and no budding events are observed (Fig. 1). However, visualization of budding events from moving objects is technically challenging and would require fast volumetric imaging to detect rapid uncoating events after vesicle formation.

ARF1 compartments are not defined by endosomal markers, but partial colocalization was observed with Rab11, a Rab GTPase commonly used as an RE marker (Fig. 5a,b). We propose that ARF1 compartments represent a tubulo-vesicular organelle with a key role in the distribution of secretory and endocytic recycling cargo along

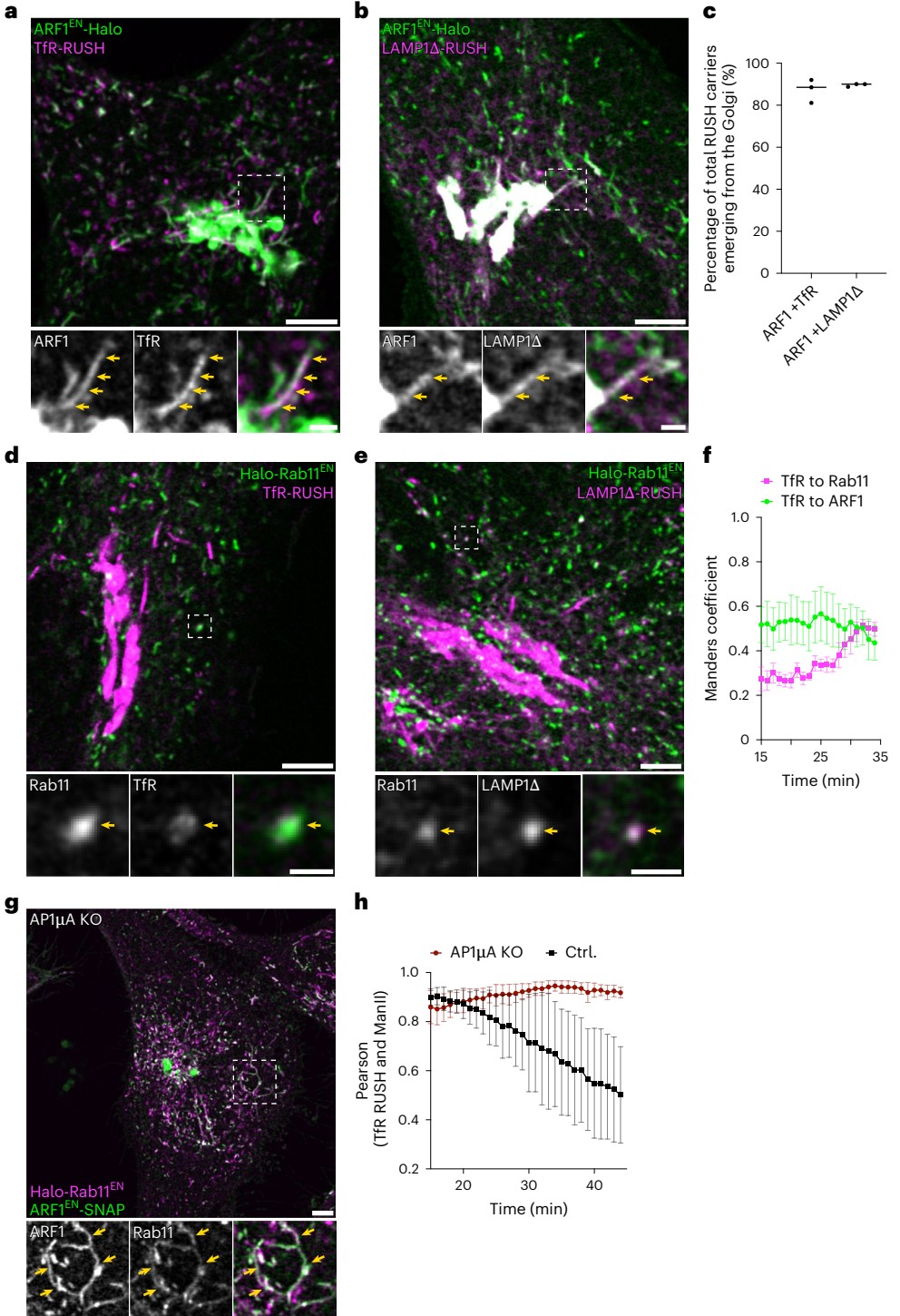

**Fig. 7 | Perinuclear ARF1 compartments transport secretory cargoes and loss of AP-1 delays cargo exit from the TGN. a,b**, ARF1$^{EN}$-Halo HeLa cells transiently expressing the RUSH constructs streptavidin-KDEL/TfR-SBP-SNAP (**a**) or streptavidin-KDEL/ssSBP-SNAP-LAMP1Δ (**b**) (w/o QYTI) labelled with CA-JF$_{552}$ and BG-JFX$_{650}$ were imaged with confocal microscopy 20 min after addition of biotin with a frame rate of 6 s per frame. A qualitative example shows secretory RUSH cargo leaving the Golgi in ARF1 compartments at 21 min (TfR) or 24 min (LAMP1Δ) post biotin addition. **c**, Manual quantification of the total RUSH cargo carriers emerging from the Golgi reveals that most secretory RUSH cargo exits via ARF1 compartments, each dot represents a single cell. **d,e**, Live-cell confocal imaging of Halo-Rab11$^{EN}$ HeLa cells transiently expressing streptavidin-KDEL/TfR-SBP-GFP labelled with CA-JFX$_{650}$ (**d**) and streptavidin-KDEL/ssSBP-SNAP-LAMP1Δ labelled with CA-JF$_{552}$ and BG-JFX$_{650}$ (**e**) shows that secretory RUSH cargo is sorted

through REs en route to the PM. A qualitative example is shown at 21 min (TfR) or 24 min (LAMP1Δ) post biotin addition. **f**, Colocalization analysis using the Manders coefficient shows that correlation of TfR-RUSH cargo with REs increases over time while correlation with ARF1 compartments shows a downward trend. Each dot represents the average of 5 cells. s.e.m. error bars. **g**, Live-cell confocal imaging of ARF1$^{EN}$-SNAP/Halo-Rab11$^{EN}$ AP1μA KO HeLa cells labelled with CA-JF$_{552}$ and BG-JFX$_{650}$ exhibit formation of long-aberrant Rab11/ARF1 compartments near the TGN. **h**, Pearson correlation coefficient of secretory TfR-RUSH cargo and the Golgi (masked by the Golgi-marker ManII) reveals that upon AP1μA KO, Golgi exit of secretory cargo is impaired in comparison to control cells, each dot represents the average of four cells, s.d. error bars. Scale bars, 5 μm (overviews) and 1 μm (crops). Source numerical data are available in source data.

post-Golgi routes. While vesicular exchange between the Golgi and endosomes has been proposed, our data rather suggest that maturation of ARF1 compartments into REs directs the flow of anterograde cargoes exiting the Golgi (Figs. 6–8). Our findings close an important gap in the understanding of communication between the Golgi and post-Golgi organelles. Peripheral ARF1 compartments are the sorting station downstream of Rab5 EEs (Fig. 8c). Dynamic imaging shows that ARF1 compartments are not simply tubulo-vesicular compartments derived from EEs but rather a stand-alone organelle (Extended Data Fig. 7e). Peripheral ARF1 compartments containing fluorescent transferrin are seen shedding ARF1 and acquiring the identity of REs, explaining that re-deposition of cargoes back to the PM is mediated by maturation (Fig. 8d). It is likely that ARF1 compartments emerging from the Golgi (which are also marked by Rab6) would undergo a similar transition to acquire endosomal identity and would shed ARF1 before being able to fuse with the PM. Previous reports in yeast and *Drosophila melanogaster* have postulated the presence of a Rab6-to-Rab11 cascade at the Golgi and for dense granule and exosome biogenesis suggesting this GTPase switch may be conserved across kingdoms[48,49]. Some secretory cargoes have been shown to follow an indirect route from the Golgi to the PM via REs[45,46] and the maturation of TGN-derived ARF1 compartments into Rab11-positive endosomes explains why cargoes are observed in endosomes downstream of the Golgi. Unfortunately, the detection of maturation events of TGN-derived tubules filled with secretory cargoes is challenging as these tubules move in and out on the plane on their way to the PM. However, the comparable behaviour of the compartments would suggest similar sorting mechanisms. Additionally, a maturation defect could explain the formation of long-aberrant Rab11 and ARF1-positive tubules filled with secretory cargo upon KO of AP-1 (Fig. 7g and Extended Data Fig. 6f). Impaired TGN export could additionally be explained by a defect in the retention of TGN-resident proteins or possibly defective retrograde endosome-to-Golgi recycling[34].

What is the role of AP-1 on ARF1 compartments? AP-1 is a versatile adaptor that acts in many trafficking steps. Because of this plethora of functions in many different cell types, it has been difficult to reach a consensus about the core functions of AP-1. AP-1 is thought to coordinate bi-directional transport between the TGN and endosomes[1,33,34]. In yeast, an additional role for AP-1 in intra-Golgi recycling of Golgi-resident proteins has also been suggested[50]. AP-1 was recently proposed to function exclusively in retrograde transport from endosomes to the Golgi[14]. Our data show that AP-1 localizes solely on the Golgi and ARF1 compartments. Assuming AP-1's role in retrograde transport to the Golgi, it is conceivable that a compartment may have to retrieve all retrograde cargoes before becoming competent for transport to the PM

(Fig. 8g). AP-1 could sequester AP-1 cargoes from ARF1 compartments, whether they have escaped the Golgi (perinuclear compartments) or upon internalization after endocytosis (peripheral compartments). Once retrograde cargoes are exported from ARF1 compartments, ARF1 would dissociate from the membrane. Interestingly, shedding of ARF1 from the membrane seems to be slowest at the regions of high curvature on the compartment where AP-1/clathrin localize (Figs. 6 and 8f), suggesting that the coat may shield ARF1 from GTP hydrolysis. We speculate that ARF1 shedding would be triggered by recruitment of specific ARF-GAPs, possibly recruited by Rab11. Notably, ARF1 compartments are seen closely interacting with Rab11-positive REs (Fig. 5 and Extended Data Fig. 5) with a similar behaviour to that observed for other endosomal compartments that undergo kiss-and-run for cargo exchange[9,28]. We speculate that such close interactions may be necessary for ARF1 compartments to acquire maturation factors and/or allow material exchange. However, further experiments would be required to prove this concept.

The presence of AP-1 and AP-3 nanodomains on ARF1 compartments involved in Tfn recycling (Extended Data Fig. 7a–c) tempts us to speculate that AP-3 could, in a similar manner, direct transport of endocytosed proteins to endo-lysosomes from peripheral ARF1 compartments. AP-3 was reported to facilitate sorting to melanosomes but its role in non-specialized cells is controversial[32]. Generally, AP-3 might be involved in the sorting of lysosomal cargoes as LAMP1 is mistargeted in AP-3 deficient mice[51], calling for further investigation on the role of ARF1 compartments in transport to lysosomes. Although it was speculated that AP-3 does not bind clathrin in vivo[37,38], the presence of a clathrin box in the β-subunit of AP-3 as well as its ability to bind clathrin in vitro indicated a possible AP-3/clathrin interaction[6]. Of note, we only see association of clathrin with AP-1 but not with AP-3 (Fig. 4a–e), indicating that AP-3 and clathrin do not interact in vivo.

Upon AP1µA KO we observed aberrant secretory ARF1 compartment containing anterograde cargo suggesting a fission defect (Fig. 7g,h and Extended Data Fig. 6f). We speculate AP-1 is responsible for the recruitment of yet-to-be identified fission factors. A potential role for the membrane-remodelling GTPase dynamin in fission events at the TGN is controversial, as several approaches suggested a contribution of dynamin to the fission of clathrin-coated vesicles from the TGN and endosomes[52–54], while in vivo studies show that it exclusively promotes fission of endocytic vesicles[55,56]. ARF1 has been proposed to mediate dynamin 2 recruitment, and depletion of dynamin 2 led to long tubular extensions from the TGN[53], reminiscent of the elongated ARF1 compartments we observe upon AP-1 depletion. Notably, peripheral ARF1 compartments involved in Tfn recycling remain morphologically unchanged upon AP1µA KO, while Tfn recycling is affected (Fig. 8e and

**Fig. 8 | ARF1 compartments mediate endocytic recycling and direct cargo flow via maturation into REs. a**, Transferrin (Tfn) recycling assays were performed using fluorescently labelled Tfn (Tfn-AlexaFluor488). Live-cell confocal imaging in ARF1$^{EN}$-Halo HeLa cells labelled with CA-JFX$_{650}$ shows Tfn in ARF1 compartments 5 min after addition of Tfn. **b**, Live-cell confocal imaging of ARF1$^{EN}$-Halo HeLa cells transiently expressing streptavidin-KDEL/TNF-SBP-SNAP labelled with BG-JFX$_{650}$ and CA-JF$_{552}$ shows that when performing both RUSH and Tfn recycling assay in parallel, both cargoes are in separate ARF1 compartments: (i) peripheral ARF1 compartments containing only endocytic recycling cargo and (ii) perinuclear ARF1 compartments containing only secretory RUSH cargo. **c**, Tfn recycling assays using Tfn-AlexaFluor488 were performed in ARF1$^{EN}$-Halo, Halo-Rab6$^{EN}$, Halo-Rab11$^{EN}$ HeLa cells and HeLa cells transiently expressing SNAP-Rab5 (SNAP-Rab5$^{OE}$) labelled with CA-JFX$_{650}$ or BG-JFX$_{650}$. Cells were fixed at indicated time points post addition of Tfn to the culture media. Colocalization analysis using the Manders correlation coefficient showed that Tfn first enters EE, then ARF1 compartments and REs. Halo-Rab6$^{EN}$ cells were used as a negative control (neg. ctrl.). Each data point represents the average of 10 cells of two independent experiments, s.e.m. error bars. **d**, Live-cell confocal microscopy of Tfn recycling using Tfn-AlexaFluor488 in ARF1$^{EN}$-SNAP/Halo-Rab11$^{EN}$ HeLa cells labelled with BG-JFX$_{552}$ and CA-JFX$_{650}$. Tfn-containing ARF1 compartments are seen to shed

ARF1 from their membrane and mature into REs. **e**, Correlation analysis using the Manders coefficient of ARF1 and Tfn in ARF1$^{EN}$-Halo HeLa cells or ARF1$^{EN}$-Halo/ AP1µA KO HeLa cells shows that Tfn retains longer in ARF1 compartments upon KO of AP1µA. Each dot represents the average of 5 cells, s.d. error bars. **f**, Live-cell confocal microscopy of Tfn recycling using Tfn-AlexaFluor488 in ARF1$^{EN}$-Halo/ AP1µA$^{EN}$-SNAP HeLa cells labelled with BG-JFX$_{552}$ and CA-JFX$_{650}$. AP-1 localizes to maturing ARF1 compartments that contain Tfn. **g**, Model illustrating how ARF1 compartments orchestrate cargo flow via maturing into RE. Clathrin-dependent post-Golgi pathways are mediated by two classes of ARF1 compartments that harbour AP nanodomains, allowing for site-specific cargo enrichment. Secretory cargoes flow is mediated by maturation of ARF1 compartments into Rab11-positive REs, whereas retrieval to the Golgi transport would be driven by AP-1 carriers (grey arrow). A segregated Rab6-dependent pathway coordinates direct Golgi-to-PM traffic (magenta arrow). Endocytic cargo is first taken up in Rab5-positive early endosomes. Downstream of Rab5, maturation of ARF1 compartments into Rab11-positive REs would allow recycling of cargoes back to the PM (green arrow). It is unclear whether Rab11-positive endosomes are the last compartment that can fuse with the PM (indicated by '?'). Scale bars, 5 µm (overviews **a**–**c**), 10 µm (overviews in **d** and **f**) and 1 µm (crops). Source numerical data are available in source data.

Extended Data Fig. 7)[47]. Differential recruitment and specific function of AP-1 may be driven by distinct subsets of interacting proteins and co-adaptors, including specific lipid-modifying enzymes such as phosphatidylinositol-4-kinases[57]. Loss of AP1μA led to a pronounced reduction of clathrin association on long-aberrant ARF1 compartments, without affecting clathrin recruitment to the Golgi (Fig. 4f,g).

Clathrin could be recruited to ARF1 compartments or the TGN via other ARF-dependent adaptors such as GGA proteins[34]. Sub-populations of AP-1, either associated with GGAs or alone, could drive differential effector recruitment. Additionally, the role of non-classic adaptor proteins, such as EpsinR[58], in intracellular clathrin-dependent sorting is not well explored.

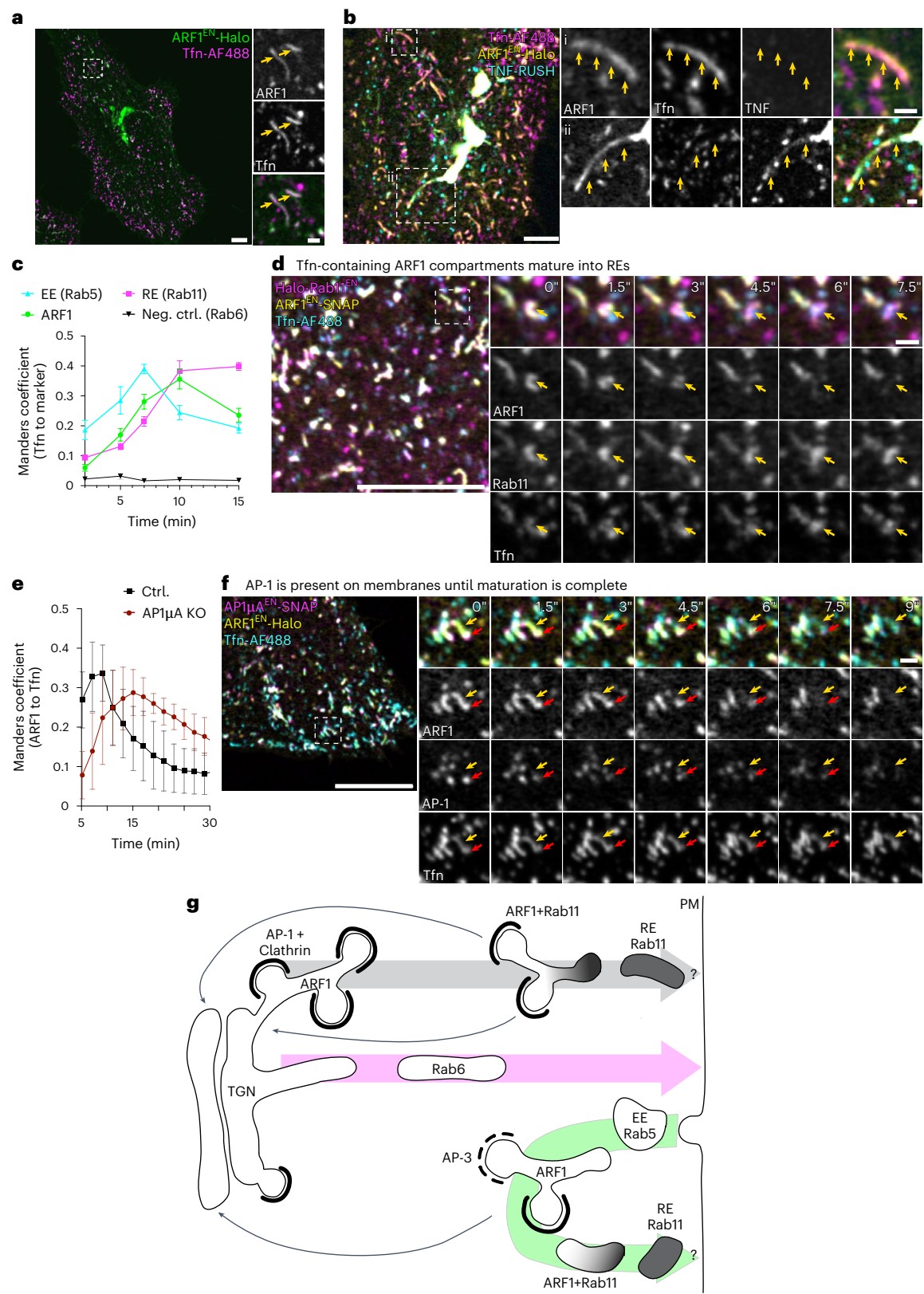

The concept of endosomal maturation has already been described, as early endosomes were seen to shed Rab5 and acquire the late endosome marker Rab7 (refs. 21,22). We here propose a similar mechanism for conversion of ARF1 compartments into REs, which may be driven by an ARF-Rab cascade as shown for the Rab5-to-Rab7 conversion. It remains elusive whether ARF1-to-Rab11 conversion is important for the biogenesis of all REs. We envision that Golgi-derived membranes may generate ARF1 compartments and REs via complex fission and fusion mechanisms that may require fast volumetric imaging to postulate a compelling model.

As the organization of the endosomal system differs strongly between organisms and cell types, due to the presence of specialized endosomes and distinct metabolic needs, we expect cell type-dependent variations of the here described mechanism[59]. ARFs are abundant and ubiquitously expressed proteins in all organisms, suggesting their key role in post-Golgi trafficking may be conserved. Diversity of function in different cell types and organisms might also be driven by the presence of different adaptors and effector proteins. In polarized epithelia cells, the tissue-specific adaptor complex AP-1B is additionally expressed[60], adding another layer of complexity to the sorting process.

Overall, by using advanced imaging methods to visualize endogenous sorting machinery, we provide evidence for intracellular material exchange being facilitated by a tubulo-vesicular network that connects the TGN, the endosomal system and the PM and drives cargo flow via maturation.

## Online content

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

## Methods

### Mammalian cell culture

CCL-2 HeLa cells (cat. no. 93021013 ECACC General Collection) and HAP1 cells (a kind gift of T. Brummelkamp's laboratory, Netherlands Cancer Institute)[61] were grown in a humidified incubator at 37 °C with 5% $CO_2$ in Dulbecco's Modified Eagle Medium (DMEM; Gibco) supplemented with 10% fetal bovine serum (Corning), 100 U l$^{-1}$ penicillin and 0.1 g l$^{-1}$ streptomycin (Fisher Scientific). Jurkat T cells (cat. no. ACC 282 DSMZ) were grown in a humidified incubator at 37 °C with 5% $CO_2$ in Roswell Park Memorial Institute medium (Gibco) supplemented with 10% fetal bovine serum (Corning), 100 U l$^{-1}$ penicillin and 0.1 g l$^{-1}$ streptomycin (Fisher Scientific).

For transient transfection of plasmids encoding for SNAP-Rab5, GFP-ManII or RUSH cargoes into HeLa cells, a NEPA21 electroporation system was used (Nepa Gene). For this, 1 million cells were washed twice with Opti-MEM (Gibco), resuspended in 90 µl Opti-MEM and mixed with 5 µg for SNAP-Rab5, 1 µg for ManII-GFP, 10 µg for RUSH cargo of DNA in an electroporation cuvette with a 2-mm gap. The electroporation reaction consists of two poring pulse (125 V, 3 ms length, 50 ms interval, with decay rate of 10% and + polarity) and five consecutive transfer pulse (25 V, 50 ms length, 50 ms interval, with a decay rate of 40% and ± polarity).

### Generation of plasmids for overexpression and gene editing

For all plasmids the used HaloTag originates from pH6HTC His6HaloTag T7 (Promega, JN874647) and the SNAP-tag from pSNAPf (New England Biolabs, N9183S). All plasmids were verified through sequencing. Sequences of all primers are provided in Supplementary Table 1.

### Generation of overexpression plasmids

For generating SNAP-Rab5A, the Rab5 fragment was amplified from eGFP-Rab5A (Addgene plasmid #49888) and cloned into a pSNAP$_f$ backbone using BamHI and NotI.

For the generation of SNAP-tagged RUSH cargoes, streptavidin-KDEL/TNF-SBP-eGFP and streptavidin-KDEL/TfR-SBP-eGFP were used[43]. eGFP was exchanged to SNAP (streptavidin-KDEL/TfR and TNF-SBP-SNAP) by amplifying SNAP as a SbfI/XbaI fragment from pSNAPf and inserting it into the SbfI/XbaI linearized eGFP plasmids.

For generating the soluble SNAP (sSNAP) cargo (streptavidin-KDEL/ss(signal sequence)-SBP-SNAP), a ssSBP-SNAP fragment was synthesized as a gBlock and inserted into the XbaI/EcoRI linearized streptavidin-KDEL/SBP-SNAP-colX vector[43]. For creating streptavidin-KDEL/ssSBP-SNAP-LAMP1Δ (w/o QYTI), the vector Str-KDEL-SBP-SNAP-CollagenX was PacI/XbaI digested and SNAP-LAMP1Δ was inserted as gBlock through Gibson assembly. ManII-eGFP[62] and streptavidin-li/ssSBP-eGFP-VSV-G[43] were described previously.

### Generation of CRISPR knock-in cell lines

**Cloning of guide and homology repair plasmids.** All guide RNAs were designed using the online tool Benchling (https://www.benchling.com) and cloned into either the SpCas9 pX459 plasmid (Addgene plasmid #62988)[63] or the SpCas9 pX330 plasmid (Addgene plasmid #42230)[64] by annealing oligonucleotides and ligation into the vector which was linearized with BbsI. A detailed list of guide RNA sequences is provided in Supplementary Table 2.

**Cloning of HR plasmids.** To avoid re-cutting from the Cas9, the protospacer-adjacent motif site was mutated in the HR plasmids (if part of the coding region a silent mutation was introduced). The G418 resistance (G418$^R$ containing SV40 promoter, ORF, polyadenylation signal) cassette originates from pEGFP-N1 (Clontech), the puromycin resistance cassette (Puro$^R$ containing SV40 promoter, ORF, polyadenylation signal) from pPUR (Clontech), the hygromycin resistance cassette (hygromycin$^R$ containing SV40 promoter, ORF, polyadenylation signal)

from pcDNA5/FRT/TO V5 (Addgene plasmid #19445[65]) and the SV40 polyadenylation signals (PolyA) sequence originates from pEGFP-N1.

*HR plasmids AP1µA.* The HR plasmid was designed with ~1-kb homology arms and was synthesized by Twist Bioscience. A glycine–serine (GS) linker and a BamHI and EcoRI site were added between the two homology arms for insertion of tags and resistance cassette. The coding sequences of HaloTag, SNAP-tag and mStayGold, including a small epitope tag, were integrated between the homology arms, followed by a polyA sequence and a G418$^R$ (in combination with HaloTag), a Puro$^R$ (in combination with SNAP-tag) or a hygromycin$^R$ (in combination with mStayGold). The coding sequences of HaloTag, SNAP-tag, were obtained via PCR using sense primers with a BamHI restriction site and antisense primers with a NheI restriction site. The SNAP-tag antisense primer included the sequence of the V5 epitope tag. The SV40 polyA sequence was amplified from pEGFP-N1 using a PolyA NheI sense and a PolyA NotI for pEGFP-C1 using a G418 NotI sense and G418 EcoRI antisense primer. The Puro$^R$ cassette was amplified using a Puro NotI sense and Puro EcoRI antisense primer. The various fragments were cloned into the HR vector linearized with BamHI and EcoRI. To generate the AP1µA$^{EN}$-mStayGold-PolyA-hygromycin$^R$ HR plasmid, the AP1µA$^{EN}$-Halo-PolyA-G418$^R$ HR plasmid was digested with BamHI and NheI and the coding sequence of mStayGold was inserted as gBlock to replace the Halo coding sequence. In a second step the hygromycin$^R$ cassette was amplified using a hygromycin ClaI sense and hygromycin EcoRI antisense primer and inserted into the AP1µA-StayGold HR vector linearized with ClaI and EcoRI.

*HR plasmid AP2µ.* The HR plasmid was designed according to the design of the HR plasmid for AP1µA and was synthesized by Twist Bioscience. To generate the AP2µ$^{EN}$-SNAP-V5-PolyA-Puro$^R$ HR plasmid the entire insert (SNAP-V5-PolyA-Puro$^R$) was excised from the AP1µA$^{EN}$-SNAP-V5-PolyA-Puro HR plasmid using the BamHI and the EcoRI site and inserted into the ordered AP2µ plasmid, linearized with BamHI and EcoRI.

*HR plasmid AP3µA.* The HR plasmid was designed according to the design of the HR plasmid for AP1µA and was synthesized by Twist Bioscience. To generate the AP3µA$^{EN}$-SNAP-V5-PolyA-Puro HR plasmid the entire insert (SNAP-V5-PolyA-Puro$^R$) was excised from the AP1µA$^{EN}$-SNAP-V5-PolyA-Puro$^R$ HR plasmid using the BamHI and the EcoRI site and inserted into the ordered AP3µA plasmid, linearized with BamHI and EcoRI. To generate the AP3µA$^{EN}$-Halo-ALFA-PolyA-G418$^R$ HR plasmid the PolyA-G418 fragment was excised from AP1µA$^{EN}$-Halo-PolyA-G418$^R$ HR using NheI and EcoRI. The coding sequences of HaloTag were obtained via PCR using a sense primer with a BamHI restriction site and an antisense primer with a NheI restriction site which included the sequence of the ALFA epitope tag. Both fragments were inserted into the ordered AP3µA HR plasmid, linearized with BamHI and EcoRI.

*HR plasmids AP4µ.* The HR plasmid was designed according to the design of the HR plasmid for AP1µA and was synthesized by Twist Bioscience. To generate the AP4µ$^{EN}$-eGFP-PolyA-G418$^R$ HR plasmid the PolyA-G418$^R$ fragment was excised from AP1µA$^{EN}$-Halo-PolyA-G418$^R$ HR using NheI and EcoRI. The coding sequence of eGFP was obtained via PCR from pEGFP-C1 using a sense primer with a BamHI restriction site and an antisense primer with a NheI restriction site.

*HR plasmids AP1γ1 and AP3δ1.* The HR plasmids were designed according to the design of the HR plasmid for AP1µA and was synthesized by Twist Bioscience with homology arms shortened to 500 bp. To generate the AP1γ1$^{EN}$-SNAP-V5-PolyA-Puro$^R$ and AP3δ1$^{EN}$-SNAP-V5-PolyA-Puro$^R$ HR plasmid the entire insert (SNAP-V5-PolyA-Puro$^R$) was excised from the AP1µA$^{EN}$-SNAP-V5-PolyA-Puro$^R$ HR plasmid using the BamHI and

the EcoRI site and inserted into the ordered HR plasmids, linearized with BamHI and EcoRI.

*HR plasmids CLCa*. The HR plasmid designs of LoxP-Puro[R]-LoxP-SNAP-CLCa[EN] and LoxP-G418[R]-LoxP-Halo-CLCa[EN] were previously described[23].

*HR plasmids ARF1*. The design of ARF1[EN]-Halo/SNAP-PolyA-G418[R] HR plasmid was previously described[13]. To generate the ARF1[EN]-eGFP HR plasmid an alternative version of the HR plasmid was synthesized by Twist Bioscience including ~1-kb homology arms with a GS linker and a NheI and BamHI site between the two homology arms. The coding sequences of eGFP were obtained via PCR from pEGFP-C1 using sense primers with a NheI restriction site and antisense primers with a BamHI restriction site. The fragment was cloned into the HR plasmid linearized with NheI and BamHI.

To generate the ARF1[EN]-SNAP-V5-PolyA-Puro[R] HR plasmid, first a SNAP-V5-tag fragment was obtained via PCR using a sense primer and antisense with a NheI restriction site and inserted in the ARF1 HR plasmid linearized with NheI. In a second step, the PolyA-Puro[R] fragment was obtained via PCR using a sense with a NheI site and antisense primer with a BamHI restriction site and inserted in the ARF1[EN]-SNAP-V5 HR plasmid linearized with NheI and BamHI. To generate the ARF1[EN]-mStayGold HR plasmid, an alternative version of the HR plasmid was synthesized by Twist Bioscience including 500 bases homology arms with a GS linker and a BamHI and EcoRI site between the two homology arms. An mStayGold-PolyA-Hygromycin[R] fragment was excised from the AP1µA[EN]-mStayGold-PolyA-hygromycin[R] HR plasmid using BamHI and the EcoRI and inserted into the new ARF1 HR plasmid, linearized with BamHI and EcoRI.

*HR plasmids Rab11A and Rab7A*. The HR plasmid for endogenously tagging Rab11A N-terminally was designed with ~1,000-bp homology arms based on the region around the start codon. The HR plasmid was constructed starting from pHaloSec61ß plasmid, which is based on pEGFP-C1 and contains a Halo tag. The homology arms were amplified from genomic wild-type HeLa DNA. First, the left homology arm (LHA) was cloned using AseI and NheI and then the right homology arm (RHA) using MluI and BglII. A GS linker was included before the RHA. Last, a G418[R] flanked by loxP sites was cloned into the full HR plasmid as a NheI fragment. The HR plasmid for endogenously tagging Rab7A N-terminally was generated like Rab11A with the exception that the homology arms were ordered as gBlocks and inserted through Gibson assembly at the AseI/NheI (LHA) and MluI/BglII (RHA) site.

*HR plasmids Rab6 and SNX1*. The HR plasmids for endogenously tagging Rab6 and SNX1 N-terminally were designed with ~700-bp homology arms based on the region around the start codon and synthesized by GeneScript (SNX1) or Thermo Fisher Scientific (Rab6) containing a GS linker located before the RHA. The designed HR plasmids contain the restriction sites NheI/SpeI (SNX1) or NheI/BamHI (Rab6) for insertion of tags and additional DNA between the two homology arms. Generally, for the addition of an N-terminal tag, a plasmid was constructed in the laboratory, containing a G418[R] flanked by loxP sites, 3xALFA tag (ordered as a G-Block from IDT and inserted through Gibson assembly) and a Halo tag containing NheI/SpeI. This loxP-G418[R]-loxP-3xALFA-Halo fragment was excised and cloned into the synthesized SNX1 HR linear plasmid at the NheI/SpeI site. For inserting the N-terminal tag into the synthesized Rab6 HR plasmid the loxP-G418[R]-loxP-3xALFA-Halo was amplified as a NheI/BamHI fragment and inserted into the NheI/BamHI linearized Rab6 HR plasmid.

## Generation of CRISPR KI cell lines
For the generation of HeLa KI cell lines, cells at 70–80% confluency were transiently transfected with both guide and HR plasmids using FuGENE HD (Promega) according to the supplier's protocol.

For the generation of HAP1 KI cell lines, 1 million HAP1 cells were washed twice with Opti-MEM, resuspended in 90 µl Opti-MEM and mixed with 5 µg of guide and HR plasmid DNA in an electroporation cuvette with a 2-mm gap. The electroporation reaction consists of two poring pulse (125 V, 3 ms length, 50 ms interval, with decay rate of 10% and + polarity) and five consecutive transfer pulses (20 V, 50 ms length, 50 ms interval, with a decay rate of 40% and ± polarity).

In both cases, G418, puromycin or hygromycin were added to the cells 3 days after transfection at a concentration of 1.5 mg ml$^{-1}$ (G418) or 2 µg ml$^{-1}$ (puromycin) or 0.4 mg ml$^{-1}$ (hygromycin) and the medium was exchanged every 2–3 days until selection was complete (G418 7–10 days, puromycin 2–5 days, hygromycin 6–8 days). After the selection of N-terminal KIs, cells were again transfected with a Cre-recombinase (Addgene plasmid #11923)[66] using FuGENE HD to remove the loxP-flanked resistance cassette.

For generation of KI Jurkat cell lines, 5 million Jurkat T cells were washed twice with Opti-MEM (Gibco), resuspended in 90 µl Opti-MEM and mixed with 2.5 µg of the guide and HR plasmids DNA in an electroporation cuvette with a 2-mm gap. The electroporation reaction consists of two poring pulse (150 V, 5 ms length, 50 ms interval, with decay rate of 10% and + polarity) and five consecutive transfer pulses (20 V, 50 ms length, 50-ms interval, with a decay rate of 40% and ± polarity). G418 was added to the cells 4 days after transfection at a concentration of 3 mg ml$^{-1}$ and medium was exchanged every 2–3 days for 7 days. Cells were again electroporated using the same protocol with a Cre-recombinase (Addgene plasmid #11923)[66] to remove the loxP-flanked resistance cassette. After 3 days, G418 was added to the cells at a concentration of 3 mg ml$^{-1}$ medium and was exchanged every 2–3 days for 5 days.

All KI cell lines were validated via western blotting. An overview of plasmids, base cell lines and selection method used for creation of new KI cell lines is given in Supplementary Table 3.

## Generation of AP1µA knockout cell lines
To achieve an AP1µA KO, the AP1M1 gene was targeted with to guide RNAs binding to sequences in exon 2 and exon 5. Guides were designed with the online tool Benchling (https://www.benchling.com) and cloned into the SpCas9 pX459 plasmid (Addgene plasmid #62988)[63] by annealing oligonucleotides and ligation into the vector, which was linearized with BbsI. The KO guide RNA sequences are provided in Supplementary Table 2.

For generation of AP1µA KO cell lines, cells at 70–80% confluency were transiently transfected with both guide plasmids using FuGENE HD (Promega) according to the supplier's protocol. One day after transfection, transfected cells were selected with 2 µg ml$^{-1}$ puromycin. Single-cell clones were obtained via serial dilution. KO of AP1µA in single-cell clones was confirmed via western blot.

An overview of plasmids, base cell lines used for creation of AP1µA KO cell lines is given in Supplementary Table 3.

## SDS–PAGE and western blot
Cells lysates were loaded on 4–12% SDS–PAGE gels (Life Technologies) and after electrophoresis, proteins were transferred to a nitrocellulose membrane (Amersham) via wet blotting. Membranes were blocked with 5% (wt/vol) milk powder and 1% BSA in PBST and incubated with primary antibodies overnight. For detection a secondary horseradish peroxidase-coupled antibody to the primary antibody was used. To develop the membrane the ECL western blot substrate was added for 2 min and then the membrane was imaged. Used antibodies are listed in Supplementary Table 4.

## Labelling for live-cell imaging
For live-cell imaging, cells were seeded on a glass-bottom dish (3.5 cm, no. 1.5; Cellvis) coated with fibronectin (Sigma). Labelling with HaloTag and SNAP-tag substrates was carried out for 1 h at 37 °C in culture

medium. All dye-conjugates were used at a concentration of 1 µM. After the staining, cells were washed in growth medium at 37 °C for at least 1 h. Live-cell imaging was performed in live-cell imaging solution (FluoroBrite DMEM (Gibco) supplemented with 10% FBS, 20 mM HEPES (Gibco) and 1× GlutaMAX (Gibco)).

For live-cell labelling of non-adherent T cells, 200,000 cells were labelled with Halo and SNAP substrates (1 µM) for 1 h at 37 °C in culture medium in a volume of 100 µl. After staining, cells were washed in growth medium three times via centrifugation.

### Imaging and image processing
Microscopy data were collected on an Abberior STED microscope using the Imspector software from Abberior Instruments (version 16.3). Line-scanning confocal and STED imaging was carried out on a commercial expert line Abberior STED microscope equipped with 485 nm, 561 nm and 640 nm excitation lasers. For two-colour STED experiments both dyes were depleted with a 775 nm depletion laser. The detection windows were set to 498 to 551 nm, 571 to 630 nm and 650 to 756 nm. Multi-colour STED images were recorded sequentially line by line. For confocal imaging, probes that were detected in the 498 to 551 nm and the 650 to 756 nm detection windows were recorded simultaneously. If required for quantitative analysis, the laser power was kept constant between images. For live-cell confocal imaging the pixel size was set to 60 nm and for STED imaging to 30 nm (live-cell) or 20 nm (fixed cell). Live-cell imaging was performed at 37 °C.

Spinning-disk confocal imaging was carried out using a CSU-W1 SoRa spinning disk (Nikon) with NIS-Elements software (version 4.50). The microscope was equipped with a dual camera system for simultaneous dual-colour detection. All imaging was performed with a ×60 Plan Apo oil objective (NA = 1.4). For experiments, 488 nm, 561 nm and 636 nm laser lines were used for excitation. For simultaneous dual-colour imaging, the quad-bandpass filter was used with relevant centre wavelengths of 607 nm and 700 nm and full-width half maximums of 34 nm and 45 nm, respectively. For three-colour imaging, eGFP and far-red ($JFX_{650}$) signal were detected simultaneously, and the orange channel ($JF_{552}$) was detected separately. Again, the quad-bandpass filter was used with relevant centre wavelengths of 521 nm, 607 nm and 700 nm and full-width half maximums of 21 nm, 34 nm and 45 nm, respectively.

To reduce noise, confocal images were background subtracted and Gaussian blurred (1 pixel s.d.) using Fiji[67] (ImageJ version 2.7.0). STED videos in Fig. 5c and images in Extended Data Figs. 3a, 4a,b and 5a,b were deconvoluted using Richardson–Lucy deconvolution from the Python microscopy PYME package (https://python-microscopy.org). Line profiles shown in Fig. 4b,d and Extended Data Fig. 5a were obtained by drawing a perpendicular line to the direction of the membrane with Fiji on Gaussian blurred (1 pixel s.d.) STED images. The line profile data were then normalized and plotted using GraphPad Prism (GraphPad Software; https://www.graphpad.com).

### FIB-SEM CLEM
For 3D CLEM, ARF1$^{EN}$-Halo/SNAP-CLCa$^{EN}$ HeLa cells were grown on IBIDI gridded glass coverslips. Cells were stained with SNAP-tag and HaloTag substrates as for standard light microscopy. Additionally, lysosomes were labelled as SiX lysosomes[68] (0.5 µM) after Halo and SNAP labelling was complete, to visualize lysosomes. Cells were fixed for 15 min at 37 °C using 3% PFA and 0.2% glutaraldehyde. Following the fixation, the reaction was quenched using 0.1% $NaBH_4$ in PBS for 7 min and cells were rinsed three times with PBS afterwards. Cells were then directly imaged in PBS. From chosen cells confocal z-stacks with 200 nm increments were recorded. All three colours were recorded sequentially line by line.

Following light microscopy, cells were postfixed with 2% glutaraldehyde in 0.1 M cacodylate buffer. Osmification was performed with 1% $OsO_4$ and 1.5% potassium cyanoferrate (III) in 0.1 M cacodylate buffer on ice, followed by aqueous 1% $OsO_4$ at room temperature. Following washing, 1% uranyl acetate was applied and dehydration in ascending ethanol concentration was performed. Ultrathin embedding into Durcupan resin was performed[69]. After centrifugation of excessive resin amount, coverslips with cells were put into the heating cupboard for polymerization.

Following resin polymerization, coverslips were trimmed with a glass-cutting pen and mounted onto SEM pin stabs, sputter coated with carbon (30–50 nm) and imaged in SEM (Helios 5CX Dual Beam, Thermo Fisher Scientific). Low-resolution overviews of coverslip surface were used to navigate on the coverslip and find the corresponding quadrant with the cell of interest. The sum projection confocal microscopy stack was overlaid with a SEM cell image (ETD, secondary electrons) in the Thermo Fischer MAPS 3 software to specify part of cell for autoslice and view. FIB milling was performed at the 7-nm step, SEM imaging was performed at $x = 3.37$ nm and $y = 4.6$ nm, dwelling time was 5 µs and back-scattered electrons were detected by in-Column detector at 2 kV and 0.34 nA.

Stack alignment was performed in freeware Microscopy Image Browser (version 2.83). The 3D stacks were binned to 7-nm isotropic resolution and further CLEM alignment, feature segmentation and visualization was performed using Dragonfly (Comet Group, version 2022.2) software. In short, SiX-labelled lysosomes were used for rough alignment, fine alignment of light microscopy and electron microscopy volumes was performed by orienting on cell membrane edges and PM-coated pits, which were clearly visible in FIB stack as well as light microscopy. Following alignment, numerous ARF1 stained tubules were inspected and all of them are correlated with corresponding tubular compartment in electron microscopy images. ARF1 compartments and clathrin-coated compartments were segmented manually. The diameter of compartments and clathrin-coated membranes (Fig. 2d) was measured using the ruler tool in Dragonfly. The diameter of ARF1 compartments was measured in irregular distances within the ARF1 compartments shown in Fig. 2.

### Fixed-cell STED z-stack
To preserve tubulo-vesicular structures for acquisition of the 3D z-stack (Extended Data Fig. 5b), cells were fixed with 3% PFA and 0.2% glutaraldehyde, as described for FIB-SEM CLEM fixation. The z-stack was recorded with 200-nm increments.

### Immunostaining of AP1µA-Halo HAP1 cells
AP1µA-Halo HAP1 cells were seeded on fibronectin-coated coverslips and on the next day live-cell labelled with CA-$JFX_{650}$. After washing the cells were rinsed twice with PBS and fixed with 4% PFA for 10 min. Samples were rinsed three times with PBS and incubated in permeabilization buffer (0.3% NP40, 0.05% Triton-X 100 and 0.1% BSA (IgG free) in PBS) for 3 min. Following this, samples were blocked for 1 h in blocking buffer (0.05% NP40, 0.05% Triton-X 100 and 5% goat serum (Jackson ImmunoResearch) in PBS), incubated with primary anti-CHC antibody (1:1000 in blocking buffer) overnight and washed three times with washing buffer (0.05% NP40, 0.05% Triton-X 100 and 0.2% BSA in PBS). Finally, samples were incubated for 1 h with Alexa594-labelled secondary antibody (1:5,000 dilution in blocking buffer), washed three times for 5 min with washing buffer and mounted with ProLong Gold (Life Technologies).

### Trafficking assays
For the RUSH assay (Fig. 7a–f,h and Extended Data Fig. 6), plasmids encoding for the different RUSH cargoes as described in the figure legend were transiently transfected into HeLa cells. Then, 18 h after transfection, cells were stained with live-cell imaging dyes. The biotin stock solution ($c = 585$ mM in dimethylsulfoxide) was diluted to a final concentration of 500 µM (in live-cell imaging solution) before addition to the cells. Live-cell confocal imaging was started 15 min,

20 min, 35 min or 70 min after biotin addition to investigate post-Golgi trafficking of the cargo.

For the Tfn assays (Fig. 8a,d–f and Extended Data Fig. 7), live-cell labelled HeLa cells were put on ice for 10–15 min before incubated with Tfn-AF488 (25 μM, Thermo Fisher) in live-cell imaging solution at 37 °C for 3 min. Cells were washed once with live-cell imaging solution before live-cell imaging with holo-Tfn (1 mM, Thermo Fisher) was added to prevent re-endocytosis of labelled Tfn.

For the Tfn assays (Fig. 8c), live-cell labelled HeLa cells were put on ice for 10 min before incubated with Tfn-AF488 (25 μM, Thermo Fisher) in live-cell imaging solution at 37 °C for 2 min. For collecting the 2 min time point, cells were immediately washed 3× with PBS and then fixed with 3% PFA and 0.2% glutaraldehyde, as described for FIB-SEM CLEM fixation. For the later time points (5, 7, 10 and 15 min), cells were washed once with live-cell imaging solution before live-cell imaging with holo-Tfn (1 mM, Thermo Fisher) was added to prevent re-endocytosis of labelled Tfn until the indicated time point. Cells were then washed 3× with PBS and fixed. Finally, the samples were mounted using ProLong Gold.

For the simultaneous RUSH and Tfn uptake assay (Fig. 8b), cells were prepared for the RUSH assay as described above. Live-cell imaging solution containing biotin was added to the cells (final concentration biotin of 500 μM) for 12 min at 37 °C allowing it to pulse the secretory RUSH cargo out of the ER. Then, the medium was changed to live-cell imaging solution containing Tfn-AF488 (final concentration of 50 μM) for 30 s at 37 °C allowing Tfn entry into the cell followed by live-cell imaging.

## Image quantification

To estimate the association of clathrin with ARF1 compartments and AP-2 (Fig. 1c), all clathrin punctae were manually counted in 10 ARF1[EN]-eGFP/AP2μ[EN]-SNAP/Halo-CLCa[EN] HeLa cells from three independent experiments and categorized into three categories depending on their colocalization with either ARF1 or AP-2 or neither of them. A fourth category (ARF1 + AP-2) shows the percentage of clathrin puncta where the association could not be unambiguously determined. From each cell, an image at the Golgi plane and an image at the apical PM was acquired and used for quantification. For quantification of ARF1 compartments with different adaptor identity (Fig. 3d) ARF1[EN]-eGFP/AP1μA[EN]-SNAP/AP3μA[EN]-Halo HeLa cells were analysed manually. Compartments were categorized based on which AP was found associated with the structure.

To estimate the amount of association of different APs with the Golgi (Fig. 3f), dual-colour confocal images of ARF1[EN]-Halo/AP1μA[EN]-SNAP, ARF1[EN]-Halo/AP3μA[EN]-SNAP and ARF1[EN]-Halo/AP4μ[EN]-eGFP HeLa cells were used for the analysis in Fiji. A mask of the Golgi was drawn using the ARF1-signal as reference. Using the 'Find Maxima'-Tool of Fiji the number of AP puncta in the Golgi area and in the entire cell measured to estimate the percentage of Golgi-associated AP.

To quantify the correlation of clathrin with AP-1 and AP-3 (Fig. 4e) the Manders correlation coefficient was used. Confocal images of ARF1[EN]-eGFP/AP1μA[EN]-SNAP/Halo-CLCa[EN], ARF1[EN]-eGFP/AP3μA[EN]-SNAP/Halo-CLCa[EN] and ARF1[EN]-eGFP/AP1μA[EN]-SNAP/AP3μA[EN]-Halo HeLa cells were background subtracted, Gaussian blurred and the cell was selected as the region of interest (ROI). After automatic thresholding with Fiji the Manders correlation coefficient was determined.

To estimate the change in clathrin recruitment upon AP1μA KO (Fig. 4g), confocal images of ARF1[EN]-Halo/SNAP-CLCa[EN] and ARF1[EN]-Halo/SNAP-CLCa[EN] AP1μA KO HeLa cells were analysed. Each cell was imaged at the PM plane and the Golgi plane. In each cell the fluorescence intensity of ten clathrin puncta which were associated with ARF1 compartments were measured and averaged. For normalization, the average intensity of ten clathrin plaques at the PM was used.

The overlap of ARF1 compartments and different endosomal markers (Fig. 5b) was quantified through a Manders correlation

coefficient. For this, dual-colour overview images of HeLa cells expressing ARF1[EN]-SNAP/Halo-SNX1[EN]/Halo-Rab6/7/11[EN] (labelled with BG-JF[552] and CA-JFX[650]) and ARF1[EN]-Halo/SNAP-Rab5[OE] (labelled with BG-JFX[650] and CA-JF[552]) and as a positive control ARF1[EN]-Halo HeLa cells (labelled with CA-JFX[650] and CA-JF[552]) were used. The confocal images were background subtracted, Gaussian blurred and the ROI was selected by hand excluding the nucleus and Golgi. The Manders values were calculated with a CellProfiler[70] (version 4.2) script summarized in Supplementary Table 5.

To estimate the overlap of compartments exiting the Golgi positive for ARF1 and Rab6 (Extended Data Fig. 3b), live-cell confocal time-lapses of ARF1[EN]-SNAP/Halo-Rab6[EN] HeLa cells labelled with BG-JF[552] and CA-JFX[650] were taken at 10 s per frame imaging speed. Carriers emerging from the Golgi were counted manually from three different cells.

The correlation of ARF1, Rab11 and AP-1 in Extended Data Fig. 4d was investigated by live-cell confocal imaging of ARF1[EN]-eGFP/Halo-Rab11[EN]/AP1μA[EN]-SNAP HeLa cells labelled with CA-JFX[650] and BG-JF[552] by taking square confocal images of the cytosol. The images were background subtracted and Gaussian blurred and thresholded with Fiji before determining the Manders correlation coefficient with the JaCOP Fiji PlugIn[71]. To avoid higher values due to coincidental overlap, the obtained Manders values were subtracted by the Manders value measured once one channel is rotated by 90 degrees, revealing the normalized Manders values shown in Extended Data Fig. 4d.

For visualization and quantification of maturation events, cells were imaged at a frame rate of 1.5 s per frame (Fig. 6). For 11 maturation events, the total fluorescence intensity of ARF1 and Rab11 was measured using Fiji. The images were background subtracted and Gaussian blurred, and the compartment was selected as ROI and the total fluorescence signal for each channel was measured for the ROI. The ROI was adapted for each frame. The datasets were overlayed and averaged based on the timing of the drop in ARF1 fluorescence signal.

To quantify the percentage of secretory tubules exiting the Golgi positive for ARF1 (Fig. 7c), RUSH assays were performed in ARF1[EN]-Halo HeLa cells transiently overexpressing streptavidin-KDEL/TfR-SBP-SNAP labelled with CA-JF[552] and BG-JFX[650]. Live-cell confocal time-lapses were taken 20 min after biotin addition at 6 s per frame imaging speed. Tubules emerging from the Golgi were counted manually from three different cells.

To understand the directionality of secretory cargo transfer between ARF1 compartments and RE (Fig. 7f), RUSH assays of streptavidin-KDEL/TfR-SBP-SNAP transiently overexpressed in ARF1[EN]-Halo or Halo-Rab11[EN] HeLa cells were performed. Images were captured 15 min after biotin at 1 min per frame. Confocal images were background subtracted, Gaussian blurred and the ROI was selected by hand excluding the nucleus and Golgi. The threshold was obtained for every channel with Fiji and then Manders correlation coefficient was determined with the JaCOP plugin.

For determining the Golgi export delay upon AP1μA KO (Fig. 7h), RUSH assays were performed in ARF1[EN]-Halo (control) or ARF1[EN]-Halo AP1μA KO HeLa cells transiently overexpressing streptavidin-KDEL/TfR-SBP-SNAP and ManII-GFP (Golgi mask) labelled with BG-JFX[650]. Live-cell confocal overview images were taken 15 min after biotin at 1 min per frame. The ROI was selected by hand excluding the nucleus and the Pearson correlation coefficient over time was determined using the JaCOP plugin of Fiji.

To place ARF1 compartments along the endocytic recycling route (Fig. 8c), Tfn assays were performed in ARF1[EN]-Halo, SNAP-Rab5[OE] (early endosomes), Halo-Rab11[EN] (REs), and Halo-Rab6[EN] (negative control) HeLa cells labelled with CA/BG-JFX[650]. After the indicated time point, cells were fixed as described for FIB-SEM CLEM fixation and a total of ten cells (per time point of each endosomal marker) from two biological replicates were captured as confocal overview images.

Images were background subtracted and Gaussian blurred in Fiji and the Manders values were calculated with a CellProfiler[70] summarized in Supplementary Table 6.

To quantify the effect of an AP1µA KO on Tfn trafficking (Fig. 8e), Tfn assays were performed in ARF1$^{EN}$-Halo and ARF1$^{EN}$-Halo AP1µA KO HeLa cells labelled with CA-JFX$_{650}$. Tfn recycling was recorded every 2 min for 30 min and the Manders correlation coefficient of ARF1 and Tfn was determined for every time point.

## Antibodies and dyes

All live-cell imaging dyes used in the study are indicated in the figure legends and were provided from the Lavis lab[72,73]. All antibodies used in this study are listed in Supplementary Table 4.

## Statistics and reproducibility

GraphPad Prism 9.3.0 was used to generate all graphs and to perform statistical analysis. Data distribution was assumed to be normal but this was not formally tested. Datasets containing continuous data from different biological replicates were presented as superplots[74]. All statistical tests used are indicated in the figure legends. No statistical method was used to predetermine sample size and sample sizes are indicated in the figure legends. No data were excluded from the analyses. The experiments were not randomized. The Investigators were not blinded to allocation during experiments and outcome assessment. All schematics were generated with Affinity Designer. Microscopy images and western blots are shown as representative images. Micrographs in Figs. 1a,d–h, 3a–c, 4a,c,f, 5a,c,d, 7g and 8a,b,d,f and in Extended Data Figs. 1a–c, 2a–e,g, 3a–c, 4a–c, 5a–c, 6a–f and 7a–e are representatives of at least three independent experiments. The correlative light microscopy/FIB-SEM experiment (Fig. 2) was repeated three times with similar results. The presented dataset was chosen as the alignment of FIB-SEM images worked best and allowed best visualization of the underlying membrane of ARF1 compartments. Micrographs in Fig. 6 are representative of five independent experiments. The western blot shown in Fig. 4f was repeated twice with comparable results.

## Reporting summary

Further information on research design is available in the Nature Portfolio Reporting Summary linked to this article.

## Data availability

Source data are provided with this paper. All other data supporting the findings of this study are available from the corresponding author on reasonable request.

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

## Acknowledgements

This project was supported by the Deutsche Forschungsgemeinschaft (German Research Foundation) grants SFB958 (project A25 to F.B.), TRR186 (Projects A20 to F.B., A08 to V.H. and Z02 to A.Z. and H.E.) and DFG grant HA 2686/24-1 (V.H.). C.R.-R. is supported by a Human Frontier Science Program early career award to F.B. We thank L. Lavis (Janelia Research Campus) for providing all live-cell dyes we used in this study and A. Butkevich (Max Planck Institute for Biophysical Chemistry) for providing the SiX lysosomal probe. We are grateful to D. Gershlick, G. Boncompain and F. Perez for sharing plasmids and expertise for RUSH experiments. We thank S. Donat and J. Schmoranzer of the AMBIO imaging centre (Charité Berlin) for the technical assistance for spinning-disk experiments and our laboratory manager G. Carai for general technical assistance.

## Author contributions

A.S., P.A. and F.B. conceived the project. A.S., P.A., V.N., A.H., A.K., G.B., S.H., C.R.-R., C.S., A.G., J.O.M., H.E., D.P., V.H. and F.B. designed and performed experiments. A.S., P.A., V.N., A.H., A.K., S.H., A.G., A.Z. and C.R.-R. performed image analysis. A.S., P.A., V.N., A.H., A.K., S.H., C.R.-R., C.S., E.Ö., P.L., A.G., E.F. and J.O.M. generated plasmids and knock-in/knockout cell lines. A.S., P.A. and F.B. wrote the paper with input from all authors.

## Funding

## Competing interests

The authors declare no competing interests.

## Additional information

**Extended data** is available for this paper at https://doi.org/10.1038/s41556-024-01518-4.

**Correspondence and requests for materials** should be addressed to Francesca Bottanelli.

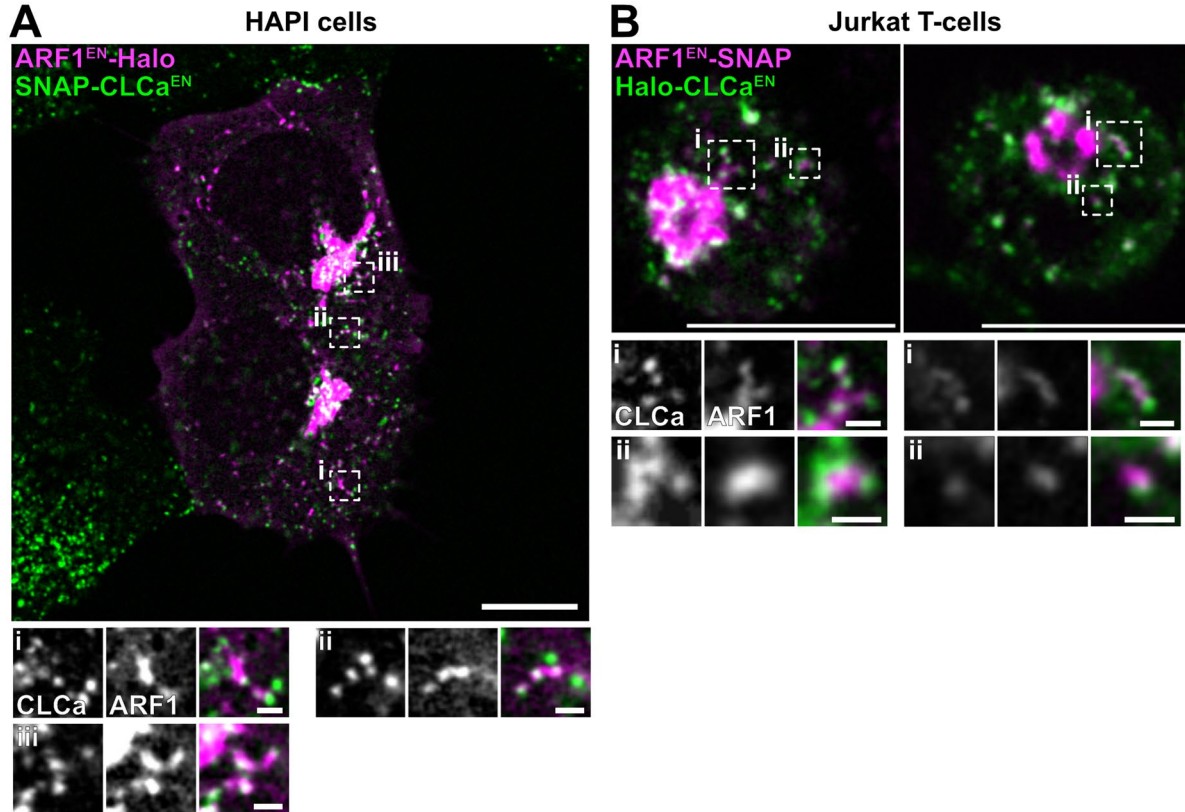

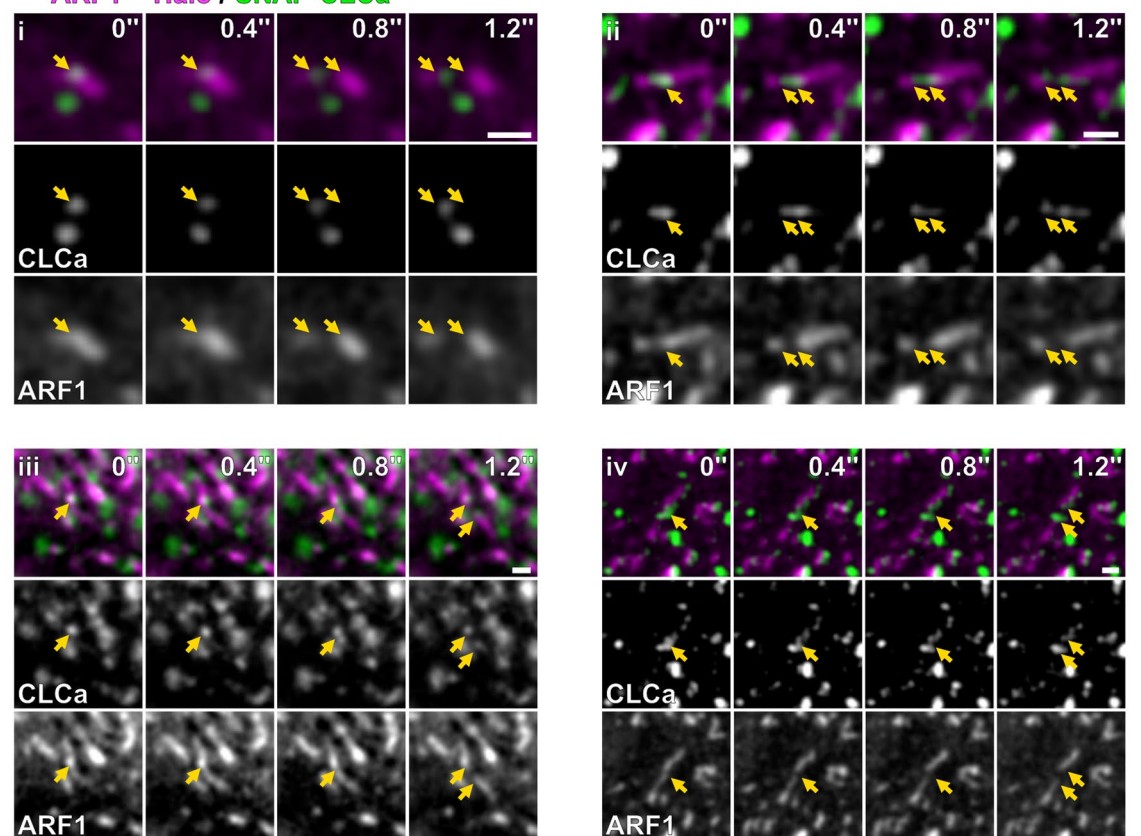

**Extended Data Fig. 1 | See next page for caption.**

**Extended Data Fig. 1 | Clathrin is associated to ARF1 compartments in different cell types and fission occurs at sites of clathrin enrichment.**
(**a-b**) Live-cell confocal imaging of ARF1$^{EN}$-Halo/SNAP-CLCa$^{EN}$ haploid HAP1 cells labelled with CA-JF$_{552}$ and BG-JFX$_{650}$ and ARF1$^{EN}$-SNAP/Halo-CLCa$^{EN}$ Jurkat T cells labelled with CA-JF$_{571}$ and BG-JFX$_{650}$ highlight ARF1 compartments decorated by clathrin domains. (**c**) (**i-iv**) Multiple examples of time-lapse confocal spinning disk imaging of ARF1$^{EN}$-Halo/SNAP-CLCa$^{EN}$ HeLa cells labelled with CA-JF$_{552}$ and BG-JFX$_{650}$ showing clathrin at the site of fission on ARF1 compartments (yellow arrows highlight the localization of clathrin). Selected frames are shown, movie was taken with a frame rate of 5 frames/s. EN=endogenous, HaloTag substrate CA=chloroalkane, SNAP-tag substrate BG=benzylguanine. Scale bars: 10 μm (overviews) and 1 μm (crops).

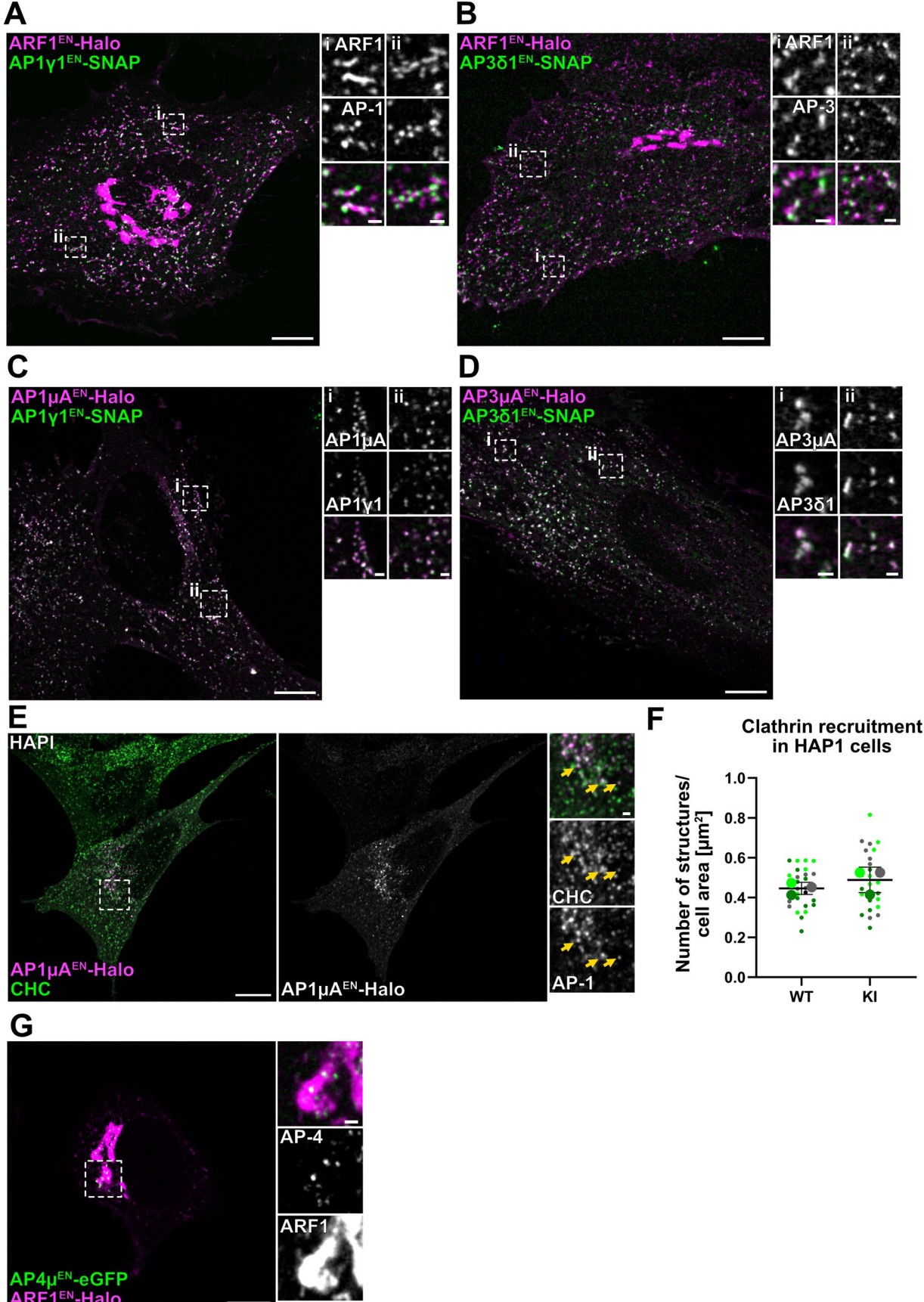

**Extended Data Fig. 2 | See next page for caption.**

**Extended Data Fig. 2 | Endogenous tagging of AP subunits does not affect AP function.** (**a-b**) Live-cell confocal imaging of ARF1$^{EN}$-Halo/AP1γ1$^{EN}$-SNAP and ARF1$^{EN}$-Halo/AP3δ1$^{EN}$-SNAP HeLa cells labelled with CA-JF$_{552}$ and BG-JFX$_{650}$ shows that C-terminal tagging of a large subunit of the adaptor complexes leads to comparable association with ARF1 compartments as observed for C-terminal tagging of the μ-subunit. (**c-d**) Live-cell confocal imaging of AP1μA$^{EN}$-Halo/AP1γ1$^{EN}$-SNAP and AP3μA$^{EN}$-Halo/AP3γ1$^{EN}$-SNAP HeLa cells labelled with CA-JF$_{552}$ and BG-JFX$_{650}$ shows that C-terminal tagging of two subunits of an APs has no effect on the localization of the complexes. (**e**) Confocal imaging of fixed AP1μA$^{EN}$-Halo HAP1 cells labelled with CA-JFX$_{650}$ before fixation and then immunostained with an anti-CHC (clathrin heavy chain) antibody.

The recruitment of clathrin to AP-1 domains is unaffected by tagging of AP1μA (yellow arrows indicate domains where CHC and AP1μA$^{EN}$-Halo colocalize). (**f**) Quantification of clathrin puncta normalized to the cell area in wild-type (WT) and AP1μA$^{EN}$-Halo knock-in (KI) HAP1 cells show no change in the number of cytosolic clathrin structures. In total 30 cells from 3 independent experiments for each condition were analysed, replicates are shown in different colours and each dot represents a single cell, SD error bars. (**g**) Live-cell confocal imaging of ARF1$^{EN}$-Halo/AP4μ$^{EN}$-eGFP HeLa cells labelled with CA-JFX$_{650}$ showing AP-4 puncta at the TGN. EN=endogenous, HaloTag substrate CA=chloroalkane, SNAP-tag substrate BG=benzylguanine. Scale bars: 10 μm (overviews) and 1 μm (crops). Source numerical data and unprocessed blots are available in source data.

**A**

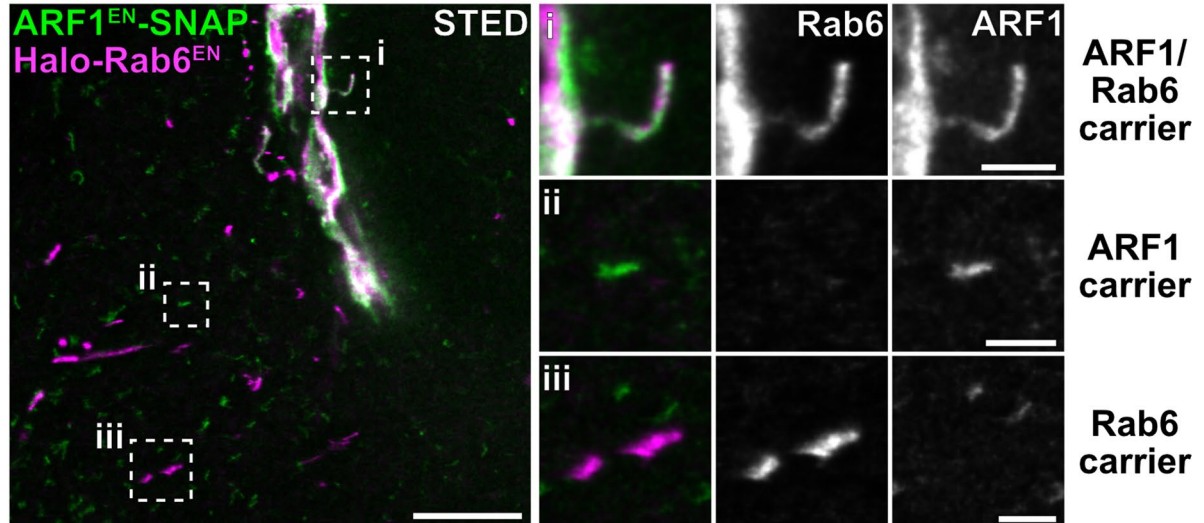

**B** ARF1$^{EN}$-SNAP / Halo-Rab6$^{EN}$

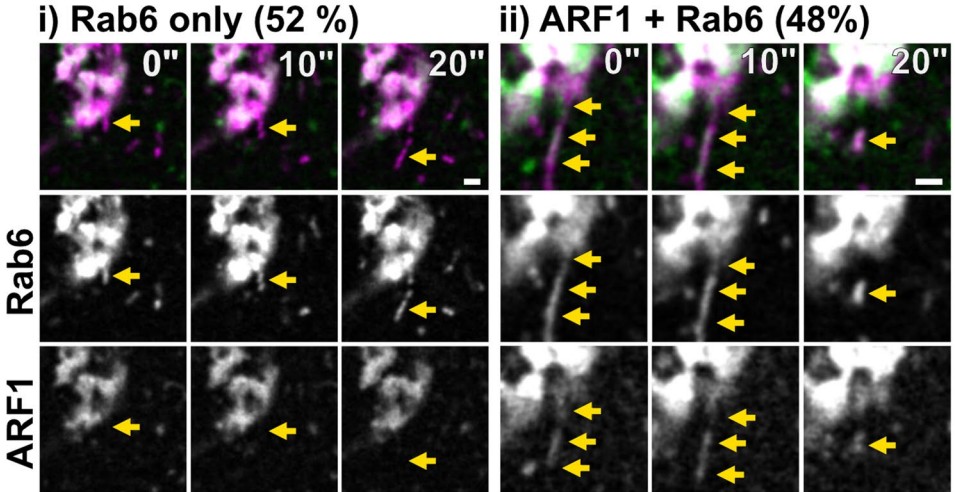

**C**

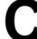

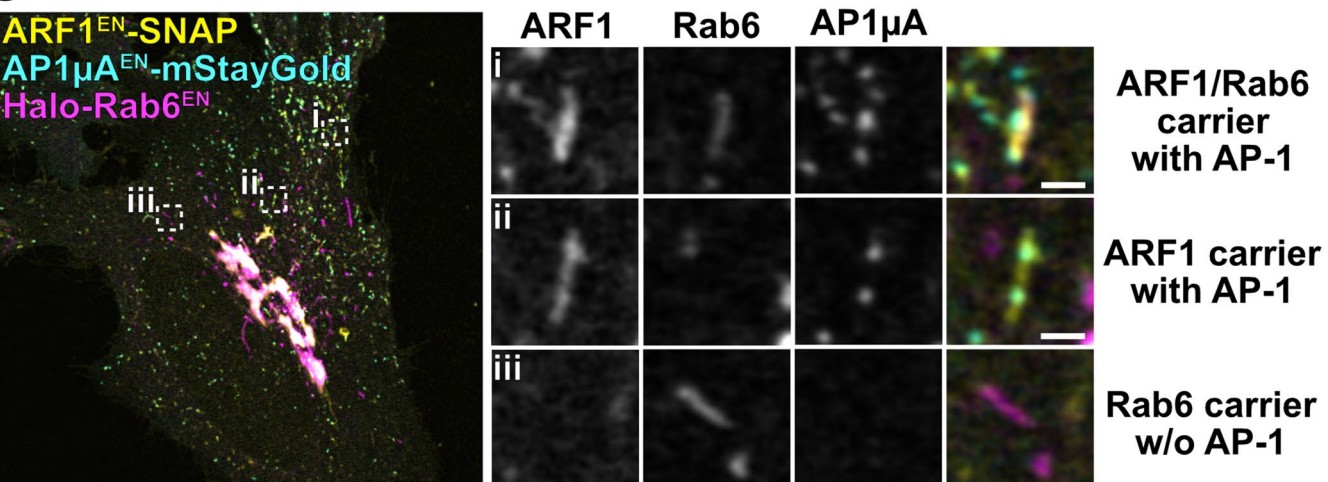

**Extended Data Fig. 3 | See next page for caption.**

**Extended Data Fig. 3 | ARF1 defines a subpopulation of Golgi-derived Rab6 carriers.** (**a**) Live-cell STED imaging of ARF1$^{EN}$-SNAP/Halo-Rab6$^{EN}$ HeLa cells labelled with CA-JF$_{571}$ and BG-JFX$_{650}$ shows that (**i**) ARF1 and Rab6 on carriers that form at the TGN whereas (**ii**) peripheral ARF1 compartments are devoid of Rab6 and (**iii**) peripheral Rab6 carriers are devoid of ARF1. (**b**) Live-cell confocal imaging ARF1$^{EN}$-SNAP/Halo-Rab6$^{EN}$ HeLa cells labelled with BG-JF$_{552}$ and CA-JFX$_{650}$ shows that two distinct populations of Rab6 carriers emerging from the TGN: (**i**) half of the Rab6 carriers are devoid of ARF1 and (**ii**) half are marked by ARF1.

Carriers emerging from the Golgi were counted manually from 3 different cells. (**c**) Live-cell confocal imaging of ARF1$^{EN}$-SNAP/Halo-Rab6$^{EN}$/AP1μA$^{EN}$-mStayGold HeLa cells labelled with BG-JF$_{552}$ and CA-JFX$_{650}$ shows that AP-1 only localizes to (**i**) ARF1/Rab6 double-positive carriers and (**ii**) ARF1 only but not to (**iii**) Rab6 only carriers. EN=endogenous, HaloTag substrate CA=chloroalkane, SNAP-tag substrate BG=benzylguanine. Scale bars: 5 μm (overview) and 1 μm (crops). Source numerical data are available in source data.

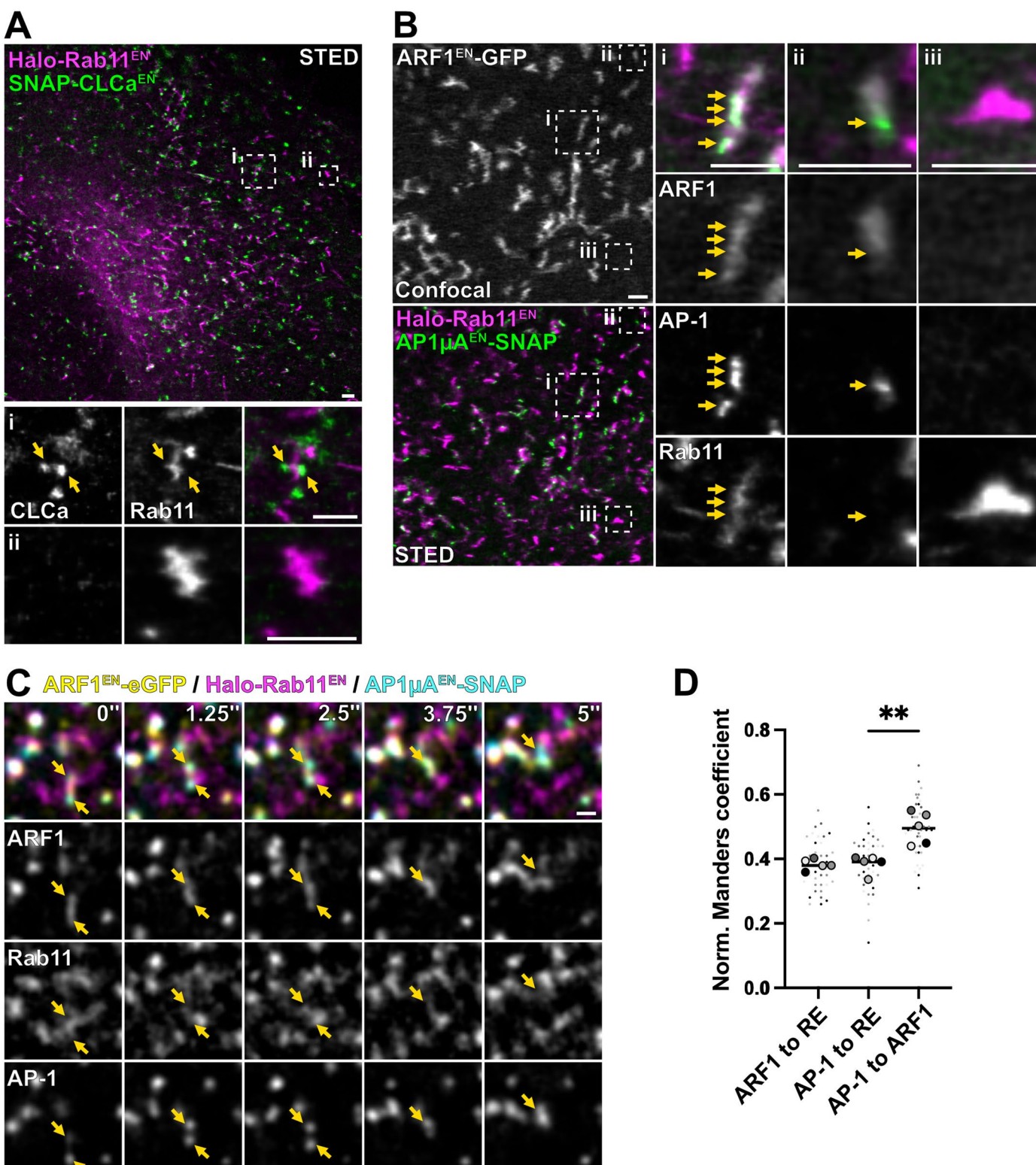

**Extended Data Fig. 4 | AP-1 and clathrin are recruited by ARF1 to RE membranes.** (**a**) Live-cell STED microscopy of CLCa$^{EN}$-SNAP/Halo-Rab11$^{EN}$ HeLa cells labelled with CA-JFX$_{650}$ and BG-JF$_{585}$ highlights (**i**) close proximity of clathrin-coated structures to a RE and (**ii**) a RE devoid of clathrin. (**b**) Live-cell STED imaging of ARF1$^{EN}$-eGFP/Halo-Rab11$^{EN}$/AP1μA$^{EN}$-SNAP HeLa cells (two-colour STED imaging with ARF1$^{EN}$-eGFP imaged in confocal mode) labelled with BG-JFX$_{650}$ and CA-JF$_{571}$ shows (**i**) AP-1 on ARF1/Rab11 double-positive compartments, (**ii**) AP-1 on ARF1 compartments devoid of Rab11, (**iii**) Rab11-positive RE devoid of AP-1 and separated from ARF1 compartments. (**c**) Time-lapse confocal spinning disk imaging shows a moving ARF1 compartment that harbours AP-1 domains (yellow arrows) (d) Quantitative colocalization analysis between ARF1, AP-1 and RE. Normalized Manders coefficient shows comparable correlation of AP-1 to RE and ARF1 to RE, whereas correlation of AP-1 to ARF1 is significantly higher. Five replicates are shown in different colours each small dot representing a single cell of the replicate. P value of an unpaired two-tailed t-test is 0.0023, **$P < 0.01$. EN=endogenous, HaloTag substrate CA=chloroalkane, SNAP-tag substrate BG=benzylguanine. Scale bars: 1 μm. Source numerical data are available in source data.

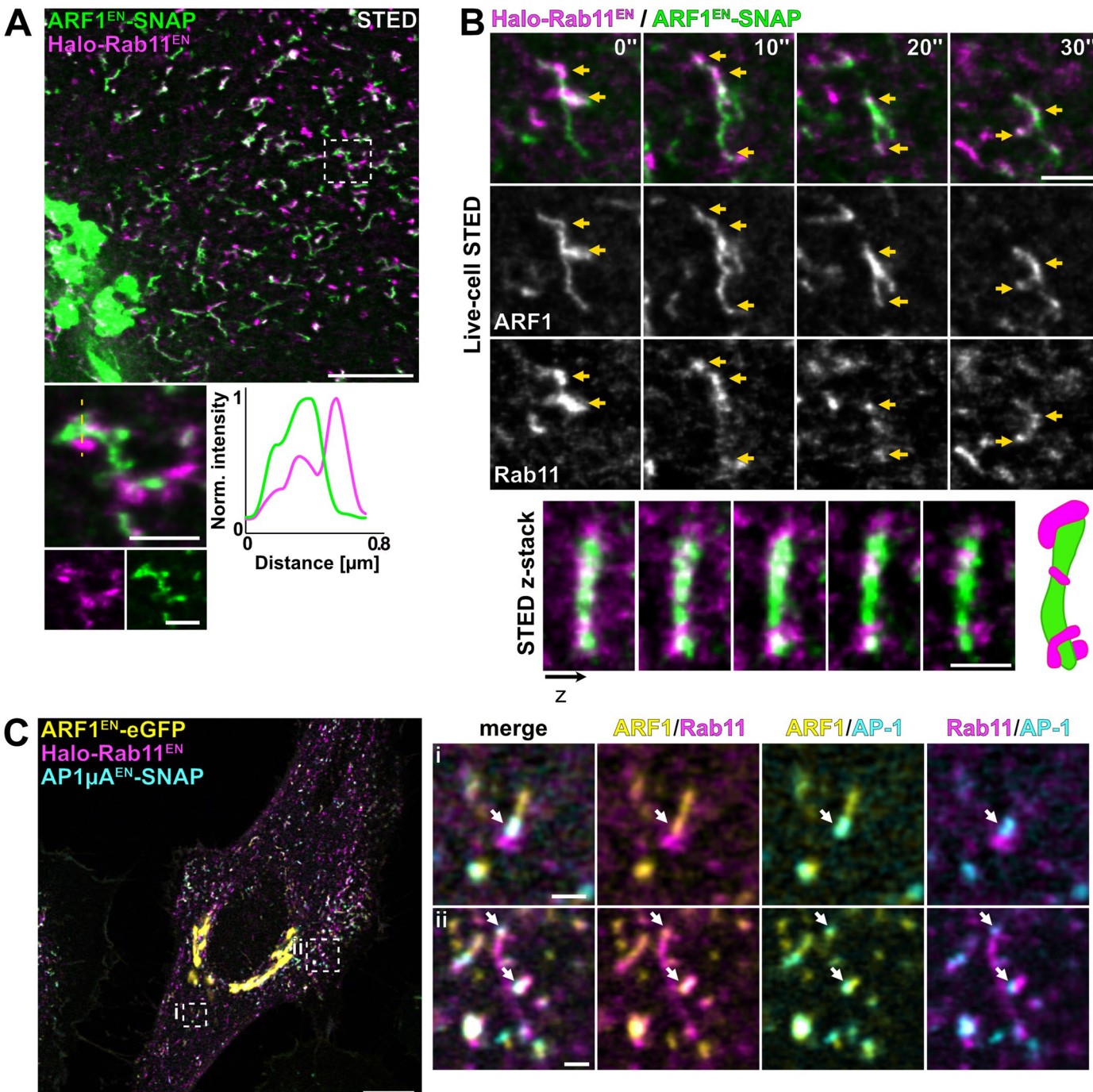

**Extended Data Fig. 5 | ARF1 compartments transiently interact with REs.**
(**a**) Live-cell STED microscopy of ARF1[EN]-SNAP/Halo-Rab11[EN] HeLa cells labelled
with CA-JF$_{571}$ and BG-JFX$_{650}$ highlights ARF1 compartments and REs at nanoscale
resolution. Line profile showing close association of ARF1 compartments
and REs. (**b**) Live-cell STED imaging of ARF1[EN]-SNAP/Halo-Rab11[EN] HeLa cells
labelled with CA-JF$_{571}$ and BG-JFX$_{650}$ shows transient interaction of a peripheral
ARF1 compartments and REs. Fixed-cell 3D-STED microscopy highlights close

proximity of a single ARF1 compartment to multiple REs. (**c**) Live-cell confocal
imaging of ARF1[EN]-eGFP/Halo-Rab11[EN]/AP1μA[EN]-SNAP HeLa cells labelled with
CA-JFX$_{650}$ and BG-JF$_{552}$ shows AP-1 at the interface of ARF1 compartment and RE
located in the (**i**) periphery and (**ii**) perinuclear area of the cell. EN=endogenous,
HaloTag substrate CA=chloroalkane, SNAP-tag substrate BG=benzylguanine.
Scale bars: 10 μm (confocal overview), 5 μm (STED overview) and 1 μm (z-stack,
crops and time-lapse), Δz=0,2 μm.

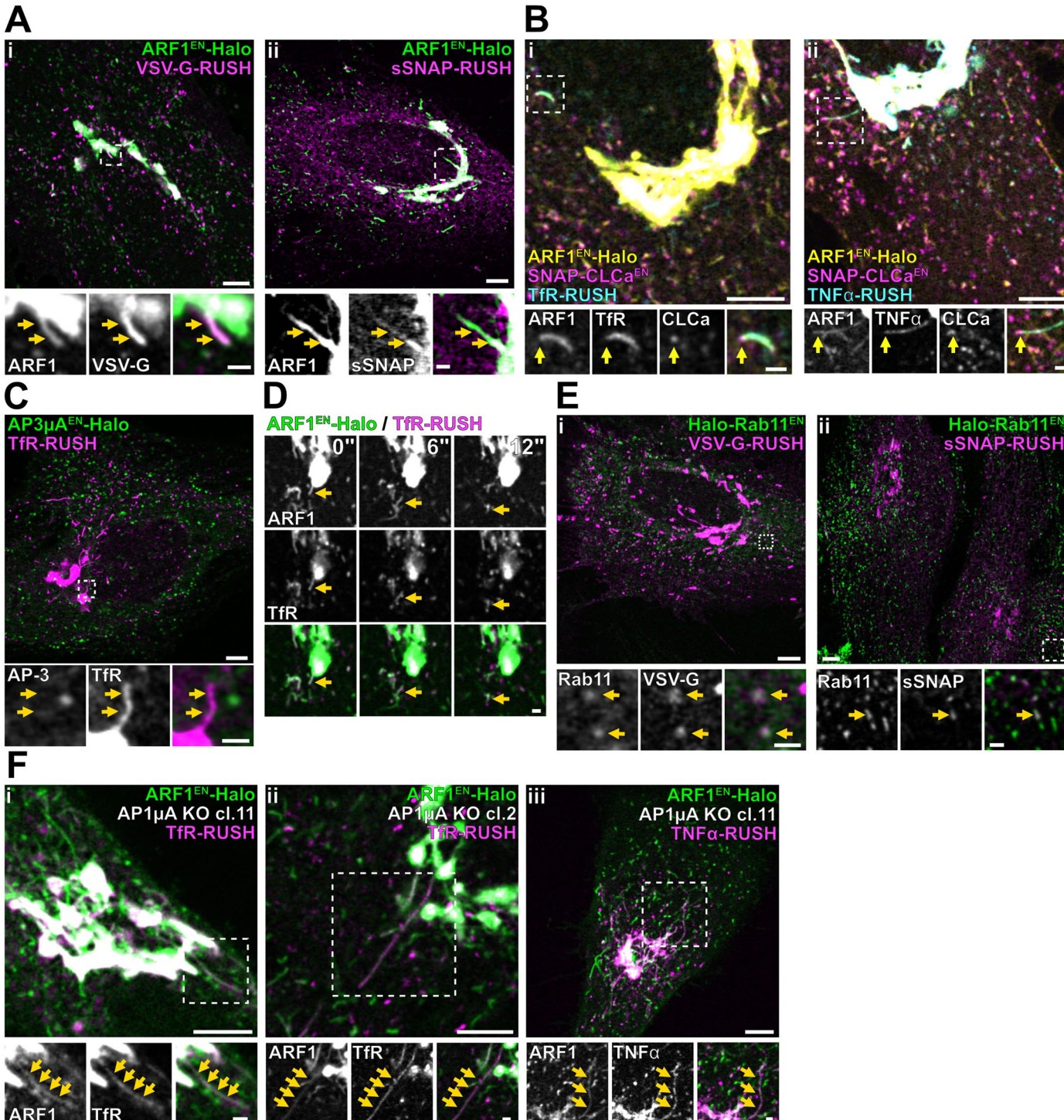

**Extended Data Fig. 6 | See next page for caption.**

**Extended Data Fig. 6 | Perinuclear ARF1 compartments mediate secretory transport.** (**a**) Live-cell confocal imaging of ARF1[EN]-Halo HeLa cells transiently expressing (**i**) Streptavidin-li/ssSBP-eGFP-VSV-G (labelled with CA-JFX$_{650}$) or (**ii**) Streptavidin-KDEL/ssSBP-SNAP (soluble SNAP (sSNAP), labelled with BG-JFX$_{650}$ and CA-JF$_{552}$) highlights the transmembrane (VSV-G) and soluble RUSH cargo in ARF1 compartments. Image taken at 21 min (VSV-G) and 52 min (sSNAP) after addition of biotin. (**b**) Live-cell confocal imaging of ARF1[EN]-Halo/ SNAP-CLCa[EN] HeLa cells transiently expressing (**i**) Streptavidin-KDEL/TfR-SBP-eGFP or (**ii**) Streptavidin-KDEL/TNFα-SBP-eGFP labelled with BG-JFX$_{650}$ and CA-JF$_{552}$ highlights transmembrane RUSH cargo (TfR, TNFα) in ARF1 compartments decorated by clathrin (yellow arrows highlight clathrin on ARF1 compartments). Images were taken (**i**) 20 min and (**ii**) 16 min after addition of biotin. (**c**) Live-cell confocal imaging of AP3µA[EN]-Halo HeLa cells transiently expressing Streptavidin-KDEL/TfR-SBP-eGFP labelled with CA-JFX$_{650}$ show that RUSH carriers are devoid of AP-3. Image taken 23 min after addition of biotin. (**d**) Live-cell confocal time-lapses of ARF1[EN]-Halo HeLa cells transiently expressing Streptavidin-KDEL/TfR-SBP-SNAP labelled with BG-JFX$_{650}$ and CA-JF$_{552}$ show

ARF1 compartments decorated by TfR-RUSH detaching from the Golgi and moving away (highlighted by a yellow arrow). The first frame was taken at 23 min after addition of biotin. (**e**) Live-cell confocal imaging of Halo-Rab11[EN] HeLa cells transiently expressing (**i**) Streptavidin-li/ssSBP-eGFP-VSV-G (labelled with CA-JFX$_{650}$) or (**ii**) Streptavidin-KDEL/ssSBP-SNAP (labelled with BG-JFX$_{650}$ and CA-JF$_{552}$) highlights the transmembrane (VSV-G) and soluble (sSNAP) RUSH cargo localization to recycling endosomes. Image taken at 25 min (VSV-G) and 70 min (sSNAP) after addition of biotin. (**f**) Live-cell confocal imaging of ARF1[EN]-Halo/ AP1µA KO HeLa cells transiently expressing (**i**) Streptavidin-KDEL/TfR-SBP-SNAP (**ii**) Streptavidin-KDEL/TfR-SBP-eGFP or (**iii**) Streptavidin-KDEL/TNFα-SBP-SNAP labelled with (**i, iii**) BG-JFX$_{650}$ and CA-JF$_{552}$ or (**ii**) CA-JFX$_{650}$ show transmembrane RUSH cargo in aberrant elongated ARF1 compartments. (**i-iii**) Different CRISPR-Cas9 KO clones display the same phenotype. Images taken (**i**) 32 min or (**ii**) 25 min or (**iii**) 30 min after addition of biotin. EN=endogenous, HaloTag substrate CA=chloroalkane, SNAP-tag substrate BG=benzylguanine, ss=signal sequence. Scale bars: 5 µm (confocal overviews), 1 µm (crops).

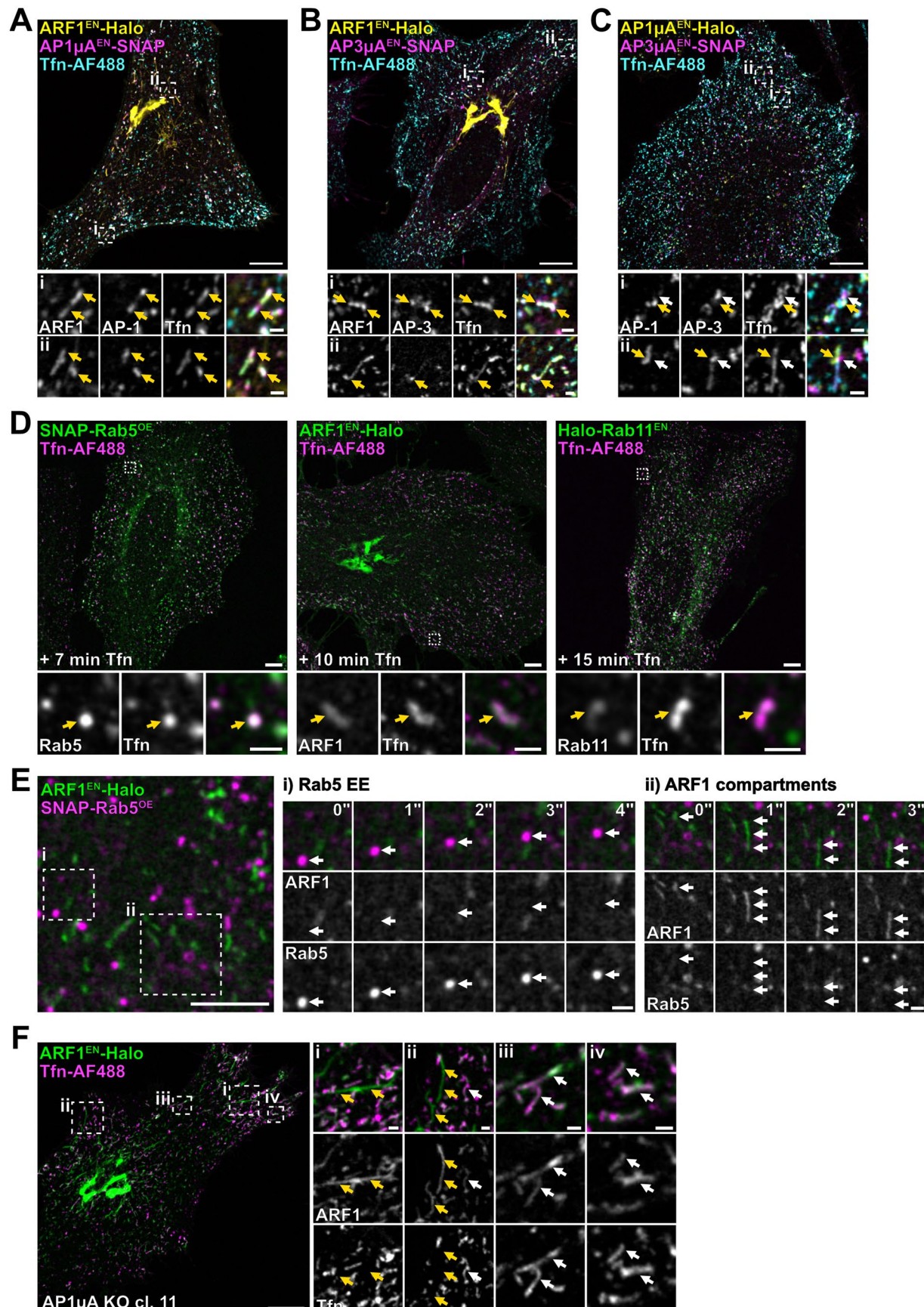

**Extended Data Fig. 7 | See next page for caption.**

**Extended Data Fig. 7 | Peripheral ARF1 compartments mediate endocytic recycling of Tfn and are not derived from early endosomes.** Live-cell confocal imaging of fluorescently labelled Tfn (Tfn-AlexaFluor488) in different KI and KO HeLa cells. (**a-b**) ARF1$^{EN}$-Halo/AP1μA$^{EN}$-SNAP and ARF1$^{EN}$-Halo/AP3μA$^{EN}$-SNAP HeLa cells labelled with CA-JF$_{552}$ and BG-JFX$_{650}$. Fluorescent transferrin localizes to ARF1 compartments harbouring AP-1 and AP-3 domains (yellow arrows). (**c**) AP1μA$^{EN}$-Halo/AP3μA$^{EN}$-SNAP HeLa cells labelled with CA-JF$_{552}$ and BG-JFX$_{650}$. Fluorescent transferrin localizes to ARF1 compartments harbouring AP-1 (yellow arrows) and AP-3 (white arrows) domains. (**d**) Tfn recycling assays using Tfn-AlexaFluor488 were performed in ARF1$^{EN}$-Halo, Halo-Rab6$^{EN}$, Halo-Rab11$^{EN}$ HeLa cells and Hela cells transiently expressing SNAP-Rab5 (SNAP-Rab5$^{OE}$) labelled with CA-JFX$_{650}$ or BG-JFX$_{650}$. Cells were fixed at indicated timepoints post addition of Tfn to the culture media. Exemplary images showing colocalization of Tfn with Rab5 (7 mins post Tfn addition), ARF1 (10 mins) and Rab11 (15 mins). (**e**) Live-cell confocal imaging of gene-edited ARF1$^{EN}$-Halo HeLa cells transiently overexpressing SNAP-Rab5$^{OE}$ labelled with CA-JF$_{503}$ and BG-JFX$_{650}$. ARF1 compartments are not derived from early endosomes (EE) and (**i**) Rab5 positive EE and (**ii**) ARF1 compartments move independently. (**f**) ARF1$^{EN}$-Halo/AP1μA KO HeLa cells labelled with CA-JFX$_{650}$. Tfn fills ARF1 compartments which morphology is unaffected upon loss of AP1μA. (**i-ii**) yellow arrows highlight aberrant elongated ARF1 compartments devoid of Tfn. (**ii-iv**) white arrows highlight shorter ARF1 compartments filled with Tfn. EN=endogenous, HaloTag substrate CA=chloroalkane, SNAP-tag substrate BG=benzylguanine. Scale bars: 10 μm (A-C, E overviews), 5 μm (D overviews), 1 μm (crops).

# Reporting Summary

## Statistics

For all statistical analyses, confirm that the following items are present in the figure legend, table legend, main text, or Methods section.

| n/a | Confirmed | |
|---|---|---|
| ☐ | ☒ | The exact sample size (*n*) for each experimental group/condition, given as a discrete number and unit of measurement |
| ☐ | ☒ | A statement on whether measurements were taken from distinct samples or whether the same sample was measured repeatedly |
| ☐ | ☒ | The statistical test(s) used AND whether they are one- or two-sided *Only common tests should be described solely by name; describe more complex techniques in the Methods section.* |
| ☐ | ☒ | A description of all covariates tested |
| ☐ | ☒ | A description of any assumptions or corrections, such as tests of normality and adjustment for multiple comparisons |
| ☐ | ☒ | A full description of the statistical parameters including central tendency (e.g. means) or other basic estimates (e.g. regression coefficient) AND variation (e.g. standard deviation) or associated estimates of uncertainty (e.g. confidence intervals) |
| ☐ | ☒ | For null hypothesis testing, the test statistic (e.g. *F*, *t*, *r*) with confidence intervals, effect sizes, degrees of freedom and *P* value noted *Give P values as exact values whenever suitable.* |
| ☒ | ☐ | For Bayesian analysis, information on the choice of priors and Markov chain Monte Carlo settings |
| ☒ | ☐ | For hierarchical and complex designs, identification of the appropriate level for tests and full reporting of outcomes |
| ☐ | ☒ | Estimates of effect sizes (e.g. Cohen's *d*, Pearson's *r*), indicating how they were calculated |

*Our web collection on statistics for biologists contains articles on many of the points above.*

## Software and code

Policy information about availability of computer code

| Data collection | Microscopy data was collected on an Abberior STED microscope using the Imspector software from Abberior instruments (Version 16.3) or on a Nikon CSU-W1 SoRa spinning disk microscope using the NIS-Elements software (Version 4.50). Electron micrographs were taken with an Helios 5 CX Dual Beam (Thermo Fisher Scientific). |
|---|---|
| Data analysis | For analysis of microscopy data Fiji (ImageJ 2.7.0) was used. The JaCOP plugin was used. For aquisition and analysis of CLEM data the follwing softwares were used: Aquisition: MAPS 3 (Thermo Fisher), Analysis: Microscopy Imaging Browiser (MIB) Version 2.83 and Dragonfly (Comet Group) were used (Software version Dragonfly 2022.2). Statistical analysis was perfomed using Graphpad Prism 9.3.0. For image quantification Cell Profiler 4.2 was used . Cell Profiler pipelines are provided as Supplementary Table 5 and 6. |

For manuscripts utilizing custom algorithms or software that are central to the research but not yet described in published literature, software must be made available to editors and reviewers. We strongly encourage code deposition in a community repository (e.g. GitHub). See the Nature Portfolio guidelines for submitting code & software for further information.

## Data

Policy information about availability of data

All manuscripts must include a data availability statement. This statement should provide the following information, where applicable:

- Accession codes, unique identifiers, or web links for publicly available datasets
- A description of any restrictions on data availability
- For clinical datasets or third party data, please ensure that the statement adheres to our policy

All data presented in plots and diagrams is provided together with the manuscript as Source Data Files (Unprocessed Western Blots and Numerical Source Data). Microscopy data is available upon reasonable request. Light microscopy images shown in the manuscript are representative of at least 3 biological replicates.

## Human research participants

Policy information about studies involving human research participants and Sex and Gender in Research.

| | |
|---|---|
| Reporting on sex and gender | NA |
| Population characteristics | NA |
| Recruitment | NA |
| Ethics oversight | NA |

Note that full information on the approval of the study protocol must also be provided in the manuscript.

# Field-specific reporting

Please select the one below that is the best fit for your research. If you are not sure, read the appropriate sections before making your selection.

☒ Life sciences      ☐ Behavioural & social sciences      ☐ Ecological, evolutionary & environmental sciences

For a reference copy of the document with all sections, see nature.com/documents/nr-reporting-summary-flat.pdf

# Life sciences study design

All studies must disclose on these points even when the disclosure is negative.

| | |
|---|---|
| Sample size | No sample size calculations were performed and no statistical method was used to determine sample size. Most experiments were replicated (biological replicates) at least three times to ensure reproducibility apart from the Transferrin uptake assays in Figure 8C (n=2). The sample sizes were chosen based on experience made in previous publications/experiments (PMID: 37102998, PMID: 28428254). Microscopy images and Western Blots are shown as representative images. |
| Data exclusions | No data was excluded from this study. |
| Replication | Sample sizes are indicated in the figure legends. All replicates were successful and yielded similiar outcomes. |
| Randomization | Data was not randomized in this study. Randomization was not relevant as analysis of microscopy data was done without prior expectations. |
| Blinding | Data was not blinded in this study. Microscopy data with KI cell lines was analyzed without previous expectations. For quantification of imaging data such as Manders and Pearson correlation coefficient data was automatically analyzed with mageJ and Cell Profiler. |

# Reporting for specific materials, systems and methods

We require information from authors about some types of materials, experimental systems and methods used in many studies. Here, indicate whether each material, system or method listed is relevant to your study. If you are not sure if a list item applies to your research, read the appropriate section before selecting a response.

## Materials & experimental systems

| n/a | Involved in the study |
|-----|----------------------|
| ☐ | ☒ Antibodies |
| ☐ | ☒ Eukaryotic cell lines |
| ☒ | ☐ Palaeontology and archaeology |
| ☒ | ☐ Animals and other organisms |
| ☒ | ☐ Clinical data |
| ☒ | ☐ Dual use research of concern |

## Methods

| n/a | Involved in the study |
|-----|----------------------|
| ☒ | ☐ ChIP-seq |
| ☒ | ☐ Flow cytometry |
| ☒ | ☐ MRI-based neuroimaging |

# Antibodies

| | |
|---|---|
| Antibodies used | CHC antibody: (Catalogue nr.: NB300-613, Supplier: Novus Biologicals , Clone: X22, Lot Number: UG285318)<br>β-Actin antibody: (Catalogue nr.: 8H10D10, Supplier: Cell Signaling , Lot Number: 20)<br>AP1M1 antibody: (Catalogue nr.: ab230273, Supplier: Abcam, Lot Number: GR3311789-9)<br>Halo antibody: (Catalogue nr.: G921, Supplier: Promega, Lot Number: 0000121291 )<br>SNAP antibody: (Catalogue nr.: A00684, Supplier: Genscript) |
| Validation | CHC antibody was validated by western blot of human brain lysates, Immunocytochemistry/Immunofluorescence in HeLa, U251 and NCI-H460 Cells and Flow cytometry of HeLa cells by the supplier.<br>β-Actin antibody was validated by western blot of various cell lines (HeLa, COS, C2C12, C6, CHO), Immunocytochemistry/Immunofluorescence in NIH/3T3 cells and Flow cytometry of HeLa cells by the supplier.<br>AP1M1 antibody was validated by western blot in HeLa, Jurkat, Hep2G and A-375 cell lysates.<br>Halo and SNAP antibodies were both tested by the suppliers for western blot applications. |

# Eukaryotic cell lines

Policy information about cell lines and Sex and Gender in Research

| | |
|---|---|
| Cell line source(s) | HeLa WT cells: ECACC General Cell Collection (93021013),<br>Jurkat cells: DSMZ (ACC 282)<br>HAP1 cells were described in Essletzbichler et. al (2014).Gift from Thijn Brummelkamp lab, Netherlands Cancer Institute |
| Authentication | HeLa and Jurkat cells were analysed and certificated by the supplier (HeLa: Sigma Aldric,  Ref: Cancer Res 1952;12:264; Proc Soc Exp Biol Med 1954;87:480, DNA profile: Amelogenin: X, CSF1PO: 9,10, D5S818: 11,12, D7S820: 8,12, D13S317: 12,13.3, D16S539: 9,10, TH01: 7, TPOX: 8,12, vWA: 16,18, Jurkat cells: DMSZ, Statement from supplier: STR analysis according to the global standard ANSI/ATCC ASN-0002.1-2021 (2021) resulted in an authentic STR profile of the reference STR database) HAP1 cells were not autenticated. |
| Mycoplasma contamination | Cell lines were not tested for any mycoplasma as gene editing was performed at low passage from supplier stock. |
| Commonly misidentified lines<br>(See ICLAC register) | We did not use any commonly misidentified cell lines. |

# Palaeontology and Archaeology

| | |
|---|---|
| Specimen provenance | *Provide provenance information for specimens and describe permits that were obtained for the work (including the name of the issuing authority, the date of issue, and any identifying information). Permits should encompass collection and, where applicable, export.* |
| Specimen deposition | *Indicate where the specimens have been deposited to permit free access by other researchers.* |
| Dating methods | *If new dates are provided, describe how they were obtained (e.g. collection, storage, sample pretreatment and measurement), where they were obtained (i.e. lab name), the calibration program and the protocol for quality assurance OR state that no new dates are provided.* |

☐ Tick this box to confirm that the raw and calibrated dates are available in the paper or in Supplementary Information.

| | |
|---|---|
| Ethics oversight | *Identify the organization(s) that approved or provided guidance on the study protocol, OR state that no ethical approval or guidance was required and explain why not.* |

Note that full information on the approval of the study protocol must also be provided in the manuscript.

# Animals and other research organisms

Policy information about studies involving animals; ARRIVE guidelines recommended for reporting animal research, and Sex and Gender in Research

| | |
|---|---|
| Laboratory animals | *For laboratory animals, report species, strain and age OR state that the study did not involve laboratory animals.* |
| Wild animals | *Provide details on animals observed in or captured in the field; report species and age where possible. Describe how animals were caught and transported and what happened to captive animals after the study (if killed, explain why and describe method; if released, say where and when) OR state that the study did not involve wild animals.* |
| Reporting on sex | *Indicate if findings apply to only one sex; describe whether sex was considered in study design, methods used for assigning sex. Provide data disaggregated for sex where this information has been collected in the source data as appropriate; provide overall numbers in this Reporting Summary. Please state if this information has not been collected.  Report sex-based analyses where performed, justify reasons for lack of sex-based analysis.* |
| Field-collected samples | *For laboratory work with field-collected samples, describe all relevant parameters such as housing, maintenance, temperature, photoperiod and end-of-experiment protocol OR state that the study did not involve samples collected from the field.* |
| Ethics oversight | *Identify the organization(s) that approved or provided guidance on the study protocol, OR state that no ethical approval or guidance was required and explain why not.* |

Note that full information on the approval of the study protocol must also be provided in the manuscript.

# Clinical data

Policy information about clinical studies
All manuscripts should comply with the ICMJE guidelines for publication of clinical research and a completed CONSORT checklist must be included with all submissions.

| | |
|---|---|
| Clinical trial registration | *Provide the trial registration number from ClinicalTrials.gov or an equivalent agency.* |
| Study protocol | *Note where the full trial protocol can be accessed OR if not available, explain why.* |
| Data collection | *Describe the settings and locales of data collection, noting the time periods of recruitment and data collection.* |
| Outcomes | *Describe how you pre-defined primary and secondary outcome measures and how you assessed these measures.* |

# Dual use research of concern

Policy information about dual use research of concern

## Hazards

Could the accidental, deliberate or reckless misuse of agents or technologies generated in the work, or the application of information presented in the manuscript, pose a threat to:

| No | Yes | |
|---|---|---|
| ☐ | ☐ | Public health |
| ☐ | ☐ | National security |
| ☐ | ☐ | Crops and/or livestock |
| ☐ | ☐ | Ecosystems |
| ☐ | ☐ | Any other significant area |

## Experiments of concern

Does the work involve any of these experiments of concern:

No | Yes

☐ ☐ Demonstrate how to render a vaccine ineffective

☐ ☐ Confer resistance to therapeutically useful antibiotics or antiviral agents

☐ ☐ Enhance the virulence of a pathogen or render a nonpathogen virulent

☐ ☐ Increase transmissibility of a pathogen

☐ ☐ Alter the host range of a pathogen

☐ ☐ Enable evasion of diagnostic/detection modalities

☐ ☐ Enable the weaponization of a biological agent or toxin

☐ ☐ Any other potentially harmful combination of experiments and agents

# ChIP-seq

## Data deposition

☐ Confirm that both raw and final processed data have been deposited in a public database such as GEO.

☐ Confirm that you have deposited or provided access to graph files (e.g. BED files) for the called peaks.

Data access links
*May remain private before publication.*

> *For "Initial submission" or "Revised version" documents, provide reviewer access links. For your "Final submission" document, provide a link to the deposited data.*

Files in database submission

> *Provide a list of all files available in the database submission.*

Genome browser session
(e.g. UCSC)

> *Provide a link to an anonymized genome browser session for "Initial submission" and "Revised version" documents only, to enable peer review. Write "no longer applicable" for "Final submission" documents.*

## Methodology

Replicates

> *Describe the experimental replicates, specifying number, type and replicate agreement.*

Sequencing depth

> *Describe the sequencing depth for each experiment, providing the total number of reads, uniquely mapped reads, length of reads and whether they were paired- or single-end.*

Antibodies

> *Describe the antibodies used for the ChIP-seq experiments; as applicable, provide supplier name, catalog number, clone name, and lot number.*

Peak calling parameters

> *Specify the command line program and parameters used for read mapping and peak calling, including the ChIP, control and index files used.*

Data quality

> *Describe the methods used to ensure data quality in full detail, including how many peaks are at FDR 5% and above 5-fold enrichment.*

Software

> *Describe the software used to collect and analyze the ChIP-seq data. For custom code that has been deposited into a community repository, provide accession details.*

# Flow Cytometry

## Plots

Confirm that:

☐ The axis labels state the marker and fluorochrome used (e.g. CD4-FITC).

☐ The axis scales are clearly visible. Include numbers along axes only for bottom left plot of group (a 'group' is an analysis of identical markers).

☐ All plots are contour plots with outliers or pseudocolor plots.

☐ A numerical value for number of cells or percentage (with statistics) is provided.

## Methodology

Sample preparation

> *Describe the sample preparation, detailing the biological source of the cells and any tissue processing steps used.*

Instrument

> *Identify the instrument used for data collection, specifying make and model number.*

| Software | *Describe the software used to collect and analyze the flow cytometry data. For custom code that has been deposited into a community repository, provide accession details.* |
|---|---|
| Cell population abundance | *Describe the abundance of the relevant cell populations within post-sort fractions, providing details on the purity of the samples and how it was determined.* |
| Gating strategy | *Describe the gating strategy used for all relevant experiments, specifying the preliminary FSC/SSC gates of the starting cell population, indicating where boundaries between "positive" and "negative" staining cell populations are defined.* |

☐ Tick this box to confirm that a figure exemplifying the gating strategy is provided in the Supplementary Information.

# Magnetic resonance imaging

## Experimental design

| Design type | *Indicate task or resting state; event-related or block design.* |
|---|---|
| Design specifications | *Specify the number of blocks, trials or experimental units per session and/or subject, and specify the length of each trial or block (if trials are blocked) and interval between trials.* |
| Behavioral performance measures | *State number and/or type of variables recorded (e.g. correct button press, response time) and what statistics were used to establish that the subjects were performing the task as expected (e.g. mean, range, and/or standard deviation across subjects).* |

## Acquisition

| Imaging type(s) | *Specify: functional, structural, diffusion, perfusion.* |
|---|---|
| Field strength | *Specify in Tesla* |
| Sequence & imaging parameters | *Specify the pulse sequence type (gradient echo, spin echo, etc.), imaging type (EPI, spiral, etc.), field of view, matrix size, slice thickness, orientation and TE/TR/flip angle.* |
| Area of acquisition | *State whether a whole brain scan was used OR define the area of acquisition, describing how the region was determined.* |

Diffusion MRI     ☐ Used     ☐ Not used

## Preprocessing

| Preprocessing software | *Provide detail on software version and revision number and on specific parameters (model/functions, brain extraction, segmentation, smoothing kernel size, etc.).* |
|---|---|
| Normalization | *If data were normalized/standardized, describe the approach(es): specify linear or non-linear and define image types used for transformation OR indicate that data were not normalized and explain rationale for lack of normalization.* |
| Normalization template | *Describe the template used for normalization/transformation, specifying subject space or group standardized space (e.g. original Talairach, MNI305, ICBM152) OR indicate that the data were not normalized.* |
| Noise and artifact removal | *Describe your procedure(s) for artifact and structured noise removal, specifying motion parameters, tissue signals and physiological signals (heart rate, respiration).* |
| Volume censoring | *Define your software and/or method and criteria for volume censoring, and state the extent of such censoring.* |

## Statistical modeling & inference

| Model type and settings | *Specify type (mass univariate, multivariate, RSA, predictive, etc.) and describe essential details of the model at the first and second levels (e.g. fixed, random or mixed effects; drift or auto-correlation).* |
|---|---|
| Effect(s) tested | *Define precise effect in terms of the task or stimulus conditions instead of psychological concepts and indicate whether ANOVA or factorial designs were used.* |

Specify type of analysis:     ☐ Whole brain     ☐ ROI-based     ☐ Both

| Statistic type for inference (See Eklund et al. 2016) | *Specify voxel-wise or cluster-wise and report all relevant parameters for cluster-wise methods.* |
|---|---|
| Correction | *Describe the type of correction and how it is obtained for multiple comparisons (e.g. FWE, FDR, permutation or Monte Carlo).* |

## Models & analysis

| n/a | Involved in the study |
|-----|------------------------|
| ☐ ☐ | Functional and/or effective connectivity |
| ☐ ☐ | Graph analysis |
| ☐ ☐ | Multivariate modeling or predictive analysis |

**Functional and/or effective connectivity**

*Report the measures of dependence used and the model details (e.g. Pearson correlation, partial correlation, mutual information).*

**Graph analysis**

*Report the dependent variable and connectivity measure, specifying weighted graph or binarized graph, subject- or group-level, and the global and/or node summaries used (e.g. clustering coefficient, efficiency, etc.).*

**Multivariate modeling and predictive analysis**

*Specify independent variables, features extraction and dimension reduction, model, training and evaluation metrics.*

