## [Peer Review File · Nature Cell Biology]

Peer Review Information

Journal: Nature Cell Biology

Manuscript Title: ARF1 compartments direct cargo flow via maturation into recycling endosomes

Corresponding author name(s): Professor Francesca Bottanelli

Editorial Notes:

Redactions – unpublished data: Parts of this Peer Review File have been redacted as indicated to maintain the confidentiality of unpublished data.

Reviewer Comments & Decisions:

Decision Letter, initial version:

*Please delete the link to your author homepage if you wish to forward this email to co-authors.

Dear Francesca,

Thank you again for submitting your manuscript, "Multi-functional ARF1 compartments serve as a hub for short-range cargo transfer to endosomes", to Nature Cell Biology and I am so sorry for the very long delay in handling your manuscript through peer review and sharing our decision with you. As you know, we gave our reviewers more time, in particular around end-of-the-year holidays. My sincere apologies

for this delay again.

The manuscript has now been seen by 3 referees, who are experts in trafficking, cargo flux (Referee #1); Golgi, trafficking (Referee #2); and ER, Golgi, trafficking (Referee #3). As you will see from their comments (attached below), they found the work of potential interest but have raised substantial concerns, which in our view would need to be addressed with considerable revisions before we can consider publication in Nature Cell Biology.

As you know, Nature Cell Biology editors discuss the referee reports in detail within the editorial team, including the Chief Editor, to identify key referee points that should be addressed with priority, as opposed to requests that are beyond the scope of the current study. To guide the scope of the revisions, I have listed these points below. Our standard revision period is six months, and we are committed to providing a fair and constructive peer-review process, so please feel free to contact me if you would like to discuss any of the referee comments further or if you anticipate any issues or delays addressing the reviews.

We stress that the reviewers' concerns are substantial and valid, and would need to be addressed with experiments and data thoroughly. Reconsideration of the study for this journal and re-engagement of referees will depend on the strength of these revisions. We recommend dedicating efforts in revision towards addressing the following points:

1- The reviewers requested additional characterizations of the tubulo-vesicular ARF1 compartment that is associated with clathrin and different Aps:

Rev#2 points #3, B, C

Rev#3 points #2-3-4-5-7-9-12

2- The reviewers are not convinced that the data support a role for the Arf1 compartment in coordinating Golgi export as well as endocytic recycling; further studies of cargo trafficking are needed to probe the model that this compartment interacts with recycling endosomes and to explore the functional implications:

Rev#1 points #1-2

Rev#2 points #2, D

Rev#3 points #1, #6, #8, #10, #11, #13

3- Minor and technical points, requests for strengthening of existing data, requests for controls, discussion, and other edits should also be addressed.

4- Finally, please pay close attention to our guidelines on statistical and methodological reporting (listed

below) as failure to do so may delay the reconsideration of the revised manuscript. In particular, please provide:

We would be happy to consider a revised manuscript that would satisfactorily address these points, unless a similar paper is published elsewhere, or is accepted for publication in Nature Cell Biology in the meantime.

- ensure that it conforms to our format instructions and publication policies (see below and <https://www.nature.com/nature/for-authors>).

- provide a point-by-point rebuttal to the full referee reports verbatim, as provided at the end of this letter.

- provide the completed Reporting Summary (found here <https://www.nature.com/documents/nr-reporting-summary.pdf>). This is essential for reconsideration of the manuscript will be available to editors and referees in the event of peer review. For more information see <http://www.nature.com/authors/policies/availability.html> or contact me.

When submitting the revised version of your manuscript, please pay close attention to our [href="https://www.nature.com/nature-portfolio/editorial-policies/image-integrity">Digital Image Integrity Guidelines](https://www.nature.com/nature-portfolio/editorial-policies/image-integrity). and to the following points below:

- that unprocessed scans are clearly labelled and match the gels and western blots presented in figures.
- that control panels for gels and western blots are appropriately described as loading on sample processing controls

-- all images in the paper are checked for duplication of panels and for splicing of gel lanes.

Nature Cell Biology is committed to improving transparency in authorship. As part of our efforts in this direction, we are now requesting that all authors identified as 'corresponding author' on published papers create and link their Open Researcher and Contributor Identifier (ORCID) with their account on the Manuscript Tracking System (MTS), prior to acceptance. ORCID helps the scientific community achieve unambiguous attribution of all scholarly contributions. You can create and link your ORCID from the home page of the MTS by clicking on 'Modify my Springer Nature account'. For more information please visit please visit www.springernature.com/orcid.

This journal strongly supports public availability of data. Please place the data used in your paper into a public data repository, or alternatively, present the data as Supplementary Information. If data can only be shared on request, please explain why in your Data Availability Statement, and also in the correspondence with your editor. Please note that for some data types, deposition in a public repository is mandatory - more information on our data deposition policies and available repositories appears below.

[Redacted]

We hope that you will find our referees' comments and editorial guidance helpful. Please do not hesitate to contact me if there is anything you would like to discuss. Thank you again for considering NCB for your work,

Best wishes,

Melina

Melina Casadio, PhD
Senior Editor, Nature Cell Biology
ORCID ID: <https://orcid.org/0000-0003-2389-2243>

Reviewers' Comments:

Reviewer #1:

Remarks to the Author:

Botanelli has previously shown that ARF1 is involved in the formation of long (~3 μm), thin (~110 nm diameter) tubular carriers located beyond the Golgi complex. The tubular carriers are largely free of COPI. In a recent JCB paper, she and her colleagues reported that ARF1 and ARF3 localize to segregated nanodomains on the trans-Golgi network (TGN) and are found on TGN-derived post-Golgi tubules, with Arf1 and Arf3 co-localizing with Golgin 97. Her work showed that ARFs define tubular-vesicular clathrin-positive structures on the TGN and the cell periphery. This submission further characterizes the post-Golgi tubules and highlights their ARF1 content.

New here is beautiful confocal microscopy combined with FIB-SEM and live cell STED. She shows that AP-1 but not AP-3 recruits clathrin to ARF1 compartments, clarifying a question in the field. The tubules do not have Rabs 5, 6 or 7. She then reports using live cell STED that recycling endosomes are tethered to ARF1 compartments. AP1 clusters more with Arf1 than Rab11 but Rab11/ARF1 extended tubules form in AP1 knockout cells for unknown reasons.

A critical experiment is presented in Figure 7, where the authors attempt to use the RUSH system to show that the ARF1 tubules are the carriers of secretory cargo. Importantly, they conclude that 90% of cargo transits through an ARF tubule (7B (is this what is shown?)) and this is messed up if AP1 is knocked out. They monitor the trafficking of the transferrin receptor. Unfortunately, this receptor's normal itinerary is to cycle between the cell surface and recycling endosomes and the authors have not convinced this reader that the pool of TfR seen in REs has not visited the plasma membrane and appeared in an RE after fast recycling—or this cargo that lives in REs may go directly there from the TGN? Here a true secreted cargo or a cell surface resident would be worth testing for comparison.

The work deserves to be presented in Nature Cell Biology because of the beautiful microscopy and potential novelty but the story is missing two things.

1. Rab6 tubules have been seen to depart from the Golgi and travel to the surface (Grigoriev, et al. and Akhmanova (2007) *Developmental Cell* 13, 305-314. How do those tubules/events mesh with the present story? Classical AP1 cargoes such as mannose 6-phosphate receptor should be imaged to tie

that sorting into the story. The authors need to better describe how they determined that all cargo goes via these tubules.

2. The authors have not proven that the secretory cargoes need to undergo a separate sorting step out of these tubules, to be delivered to the cell surface or that the Golgi-derived tubules use kiss and run with REs en route to the cell surface. The title states transfer between compartments but only external interaction is actually shown. To see kiss and run, the authors could try to fill the REs with fluorescent dextran and try to detect mixing in live cells?

Nevertheless, the experiments are state of the art in terms of all the knockin cell lines analyzed and the microscopy is quite beautiful.

Other comments

Line 156 "fusion proteins"? AP proteins are cargo adaptors; Line 199. "ARF1 compartments are multi-functional sorting endosomes". That has not yet been shown, especially at this stage of the paper.

Line 216 should be Fig. 5D? Line 140 she concludes that ARF1 compartments are the major site of clathrin dependent intracellular sorting but this is premature since no cargoes are visualized here—clear is clathrin at the tips of tubules as seen in the JCB paper. The authors should be careful because they may be tracking clathrin assemblies that are static.

Reviewer #2:

Remarks to the Author:

This paper reports an extensive live-cell imaging analysis of the dynamics of Arf1-positive membranes in a mammalian tissue culture cell line. The authors examine the association with the membranes of the clathrin adaptors AP-1 and AP-3, with secreted and endocytic cargo, and their association with other organelles. They conclude that there are two classes of Arf1-positive tubular membranes, one which emanates from the Golgi, and one which receives endocytic cargo, and both have associated AP-1 and AP-3 structures. They find that the exocytic Arf1-positive tubules contact Rab11-positive tubules and suggest that exocytic cargo is transferred between the two in order to facilitate secretion.

This is study of a high technical quality with care taken to tag proteins in the genome to avoid over-expression artefacts, and the data is carefully quantified. However, it also has three area of concern. Overall, I feel that it has the potential to make an excellent study and so I will initially describe the issues of concern, and then outline what could be done to improve the paper.

1) A few of the conclusions have already been published including some by the authors themselves. For instance, the authors have already reported that Arf1 is found on TGN-derived post-Golgi tubules

(PubMed ID 37102998), and it has been reported that AP-1 is found on such tubules as well (PubMed ID 12529433). Likewise, the association of clathrin with AP-1- but not AP-3-coated structures has been reported from immuno-EM studies (15051738). Finally, the transfer of secretory cargo to recycling endosomes has also been reported. The authors do cite these studies, even if not always directly, and their approach is more technically advanced and comprehensive, so the conclusions are valuable, but they need to be more clear about what has been reported and what is new.

2) The evidence in the paper for transfer of cargo from Arf1 tubules to Rab11-positive recycling endosomes is open to alternative interpretations, and their speculation on the mechanism for any such transfer is just that, speculation, given that the light microscopy imaging that they use lacks the resolution to determine if membranes are close, directly contacting or transiently fused. All that they can really say is that the AP-1 coat seems to be often present in the places where the two compartments are closest, and as they note themselves, such a coat is just as likely to inhibit fusion as to promote it. The moments of close proximity of the two compartments could simply reflect that fact that both are moving on microtubules and so one of each structure might transiently be on the same microtubule. Another possibility that is not discussed is that the compartments mature and Arf1 is replaced by Rab11, analogous to the Rab transition or conversion seen in the endocytic system. Certainly, none of the movies presented showed a flooding of cargo from an Arf1-positive carrier into the full extent of a Rab11-positive tubular structure as might be expected from a fusion event between the two.

3) The authors present clear data to show the Arf1 and AP-1/3 positive membranes are of at least two types. One is perinuclear and consists of tubules derived from the TGN and carries exocytic cargo. The other is peripheral and is accessed by endocytosed transferrin. They also show that the two have different dependencies on AP-1 for their fission. This is an interesting observation, but it is introduced only at the end of the paper, and in many of the previous figures the authors quantify aspects of the Arf1-positive membranes without making it clear if they are the peripheral endocytic type or the perinuclear exocytic type.

Despite these concerns I would not wish to see this studied rejected as it undoubtedly has positive features. The first four figures are really good and provide an excellent overview of the dynamics of AP-1 and AP-3 coated pits, and the authors make an excellent point that previous studies that observed AP-1-coated structures moving away from the TGN may have erroneously concluded that they were budded vesicles when in fact they are buds on tubules leaving the TGN. Indeed, this is relevant to recent suggestions from yeast studies that AP-1 may be entirely dedicated to retrieval of proteins to the Golgi rather than export from the Golgi as has been widely assumed. With some more experiments this part of the paper could be expanded to make a very valuable contribution to the field that would stand on its own without the more speculative Rab11 interactions. I would suggest the following:

A). The authors do not consider the GGA coat proteins which also act in this system and may be independent of AP-1 and AP-3. They seem well placed to resolve controversies in this area and so I

suggest that they also tag one of the GGA proteins and compare it to Arf1, AP-1, AP-3 and clathrin. It would also be informative to determine if GGAs are present on the peripheral endocytic Arf1/AP-1-positive structures.

B). The peripheral structures that are labelled with endocytosed markers are interesting. How are these formed? By following a pulse of transferrin uptake, it may be possible to determine at what stage of the endocytic recycling the ligand goes through this compartment, and if the tubules are derived from Rab5-positive early endosomes. In all the experiments the authors need to make clear if the Arf1-positive structures they show or quantify are of the peripheral or perinuclear kind.

C). Cell lines can be generated that lack Arf1 and Arf-3 (PubMed ID 34749397). The could authors look at such cells to see if the tubules still exist and if any of the coats are on them.

D). The cargo transfer aspects of the work are not well proven. The Rab11 experiments should either be removed for a later publication, or discussed briefly with various possible conclusions mentioned as interpretations that are not yet resolved. If the Rab11 data is included it would be valuable to examine both the perinuclear/exocytic and the peripheral/endocytic Arf1-positive compartments.

E) There are several more minor aspects that also need to be addressed:

- 1) The authors should more clearly discuss the evidence, direct or indirect, that the tagged proteins are functional.
- 2) 38% is not a “majority”, but rather the most abundant class of compartment as defined in the experiment (Figure 3C).
- 3) Single channel images should be shown as grey scale as this is easier to see and the intensities are clearer (eg Figure 1B insets, 2C, 3A insets, 3B insets, 3D etc).
- 4) In the discussion the authors should mention and briefly discuss a paper in bioRxiv from Margaret Robinson which addresses similar issues, even if not so expertly (<https://doi.org/10.1101/2023.11.02.565271>).

Reviewer #3:

Remarks to the Author:

The manuscript by Stockhammer et al. describes the identification of a new ARF1-positive post-Golgi compartment that serves as an intermediate for transferring secretory cargo between the TGN and the recycling endosome (RE). The work is of very high technical quality and uses cutting edge technology. The genetically engineered cell lines will be of value for others in the community. Having said this, there are several issues that must be addressed before this work can be accepted for publication. Besides technical points, I think the overall conclusion is not solid enough and other interpretations are also

possible. In addition, even if the basic observation were correct, the work does not provide mechanistic insight into the broader role of this ARF1-positive compartment. Thus, the conceptual advance is in my personal opinion relatively small.

1- What is the functional significance of this ARF1 compartment? Does it transport specific cargos? If not, what fraction of secreted proteins depends on ARF1 for their trafficking?

2- Fig1A: this result needs to be quantified, otherwise it remains anecdotal in nature. Specifically I would suggest to assess

(i) how many of the green puncta are associated with an ARF1 tubule?

(ii) Are there qualitative differences between the green puncta that are free vs. those that are associated with ARF1-tubules? To me, it appears that the CLC structures associated with ARF1 are smaller than the free puncta

3- The authors need to properly quantify the fission events, as seen for instance in Fig.1F. In this figure, I see a large tubule (much longer than the one the authors are focusing on) that does not detach. Moreover, ARF1-tubules have several spots of CLC or AP1 on them, but only some of these puncta become a fission point. The authors propose a model that CLC and or AP1 play a role in the fission of ARF1 tubules. If this were true, then why are there more puncta than fission events? This actually points to the fact that neither CLC nor AP1 are major/essential players in the observed fission event. The images that the authors provide clearly indicate that this is the case. This is one of the instances, where I think that the authors have jumped prematurely to a conclusion and are therefore proposing a model that is likely to be incorrect.

4- Fig.4Di shows that AP3 colocalizes with CLC on an ARF1 structure? However, the text of the authors proposes otherwise.

5- Figure 4: I think that the colocalization in just 11 cells is not sufficient and this number should be increased.

6- In the paragraph describing Figure 5, the authors write “We found ARF1 compartments to be hubs for adaptor-dependent post-Golgi sorting.” This statement is very pre-mature at this stage of the manuscript. The role of this compartment in sorting has not been tested at all at this stage and even later, other conclusions are possible.

7- Figure 5A: These data are not properly quantified. The authors pick a region where Rab6 and ARF1 are not in proximity. However, several other regions within the same picture do show colocalization (or at least close proximity) between ARF1 and RAB6. Therefore, the conclusion of the authors is not correct, because it is based on selective treatment of data.

8- Another example for overinterpretation of the data is the sentence: “The localization of AP-1 at the interface between ARF1 compartments and REs suggests a role for AP-1 in cargo enrichment and transfer between closely interacting compartments”. This is pure conjecture and not supported by data.

9- Figure 6: I understand that longer ARF1 tubules would be expected in the absence of AP1, because the authors propose that AP1 mediates the fission of ARF1 tubules. However, why would Rab11 also

form tubules? This indicates that something else is going on, for instance as an effect on lipid homeostasis due to a depletion of AP1. The fact that the tubules are now completely double-positive for ARF1 and RAB11 is also strange? Have the two compartments (i.e- ARF1 and RE) collapsed to one?

10- A role for AP1 as a diffusion barrier that prevents content mixing is not at all supported by the data and is also not in line with the model of the authors. Such content mixing would require that the RAB11 and ARF1 compartments to be continuous to start with (which they are not). How can a diffusion barrier act between two separate compartments.

11- Figure 7 and Ex. Figure 5: Why is the RUSH cargo located at the tip of the ARF1 tubules? It appears as if the RUSH cargo would be mostly outside the tubules, as if these were two structures that are glued to each other and that ARF1 tubules is at the back of the forward moving cargo.

12- The observations at the end of the manuscript would also be consistent with AP1 being involved in exporting cargo from the ARF1/RAB11 compartment. In the absence of AP1, the compartment remains double positive. This alternative explanation cannot be ruled out by the data. This jeopardizes the entire model.

13- The RUSH data are not fully convincing. The rise in fluorescence in RAB11 positive structures does not coincide with a drop of fluorescence in the ARF1 compartment.

REFERENCES – are limited to a total of 70 for Articles, Resources, Technical Reports; and 40 for Letters. This includes references in the main text and Methods combined. References must be numbered sequentially as they appear in the main text, tables and figure legends and Methods and must follow the precise style of Nature Cell Biology references. References only cited in the Methods should be numbered consecutively following the last reference cited in the main text. References only associated with Supplementary Information (e.g. in supplementary legends) do not count toward the total reference limit and do not need to be cited in numerical continuity with references in the main text. Only published papers can be cited, and each publication cited should be included in the numbered

reference list, which should include the manuscript titles. Footnotes are not permitted.

Methods should be written concisely, but should contain all elements necessary to allow interpretation and replication of the results. As a guideline, Methods sections typically do not exceed 3,000 words. The Methods should be divided into subsections listing reagents and techniques. When citing previous methods, accurate references should be provided and any alterations should be noted. Information must be provided about: antibody dilutions, company names, catalogue numbers and clone numbers for monoclonal antibodies; sequences of RNAi and cDNA probes/primers or company names and catalogue numbers if reagents are commercial; cell line names, sources and information on cell line identity and authentication. Animal studies and experiments involving human subjects must be reported in detail, identifying the committees approving the protocols. For studies involving human subjects/samples, a statement must be included confirming that informed consent was obtained. Statistical analyses and information on the reproducibility of experimental results should be provided in a section titled “Statistics and Reproducibility”.

All Nature Cell Biology manuscripts submitted on or after March 21 2016 must include a Data availability statement as a separate section after Methods but before references, under the heading “Data Availability”. . For Springer Nature policies on data availability see <http://www.nature.com/authors/policies/availability.html>; for more information on this particular policy see <http://www.nature.com/authors/policies/data/data-availability-statements-data-citations.pdf>. The Data availability statement should include:

- Accession codes for primary datasets (generated during the study under consideration and designated as "primary accessions") and secondary datasets (published datasets reanalysed during the study under consideration, designated as "referenced accessions"). For primary accessions data should be made public to coincide with publication of the manuscript. A list of data types for which submission to community-endorsed public repositories is mandated (including sequence, structure, microarray, deep sequencing data) can be found here <http://www.nature.com/authors/policies/availability.html#data>.
- Unique identifiers (accession codes, DOIs or other unique persistent identifier) and hyperlinks for datasets deposited in an approved repository, but for which data deposition is not mandated (see here for details <http://www.nature.com/sdata/data-policies/repositories>).
- At a minimum, please include a statement confirming that all relevant data are available from the authors, and/or are included with the manuscript (e.g. as source data or supplementary information),

listing which data are included (e.g. by figure panels and data types) and mentioning any restrictions on availability.

- If a dataset has a Digital Object Identifier (DOI) as its unique identifier, we strongly encourage including this in the Reference list and citing the dataset in the Methods.

We recommend that you upload the step-by-step protocols used in this manuscript to the Protocol Exchange. More details can found at www.nature.com/protocolexchange/about.

All imaging data should be accompanied by scale bars, which should be defined in the legend. Cropped images of gels/blots are acceptable, but need to be accompanied by size markers, and to retain visible background signal within the linear range (i.e. should not be saturated). The boundaries of panels with low background have to be demarked with black lines. Splicing of panels should only be considered if unavoidable, and must be clearly marked on the figure, and noted in the legend with a statement on whether the samples were obtained and processed simultaneously. Quantitative comparisons between samples on different gels/blots are discouraged; if this is unavoidable, it should only be performed for samples derived from the same experiment with gels/blots were processed in parallel, which needs to be stated in the legend.

The total number of Supplementary Figures (not including the “unprocessed scans” Supplementary Figure) should not exceed the number of main display items (figures and/or tables (see our Guide to Authors and March 2012 editorial <http://www.nature.com/ncb/authors/submit/index.html#suppinfo>; <http://www.nature.com/ncb/journal/v14/n3/index.html#ed>). No restrictions apply to Supplementary Tables or Videos, but we advise authors to be selective in including supplemental data.

GUIDELINES FOR EXPERIMENTAL AND STATISTICAL REPORTING

REPORTING REQUIREMENTS – We are trying to improve the quality of methods and statistics reporting in our papers. To that end, we are now asking authors to complete a reporting summary that collects

information on experimental design and reagents. The Reporting Summary can be found here <https://www.nature.com/documents/nr-reporting-summary.pdf> If you would like to reference the guidance text as you complete the template, please access these flattened versions at <http://www.nature.com/authors/policies/availability.html>.

We strongly recommend the presentation of source data for graphical and statistical analyses as a separate Supplementary Table, and request that source data for all independent repeats are provided when representative experiments of multiple independent repeats, or averages of two independent experiments are presented. This supplementary table should be in Excel format, with data for different figures provided as different sheets within a single Excel file. It should be labelled and numbered as one of the supplementary tables, titled “Statistics Source Data”, and mentioned in all relevant figure legends.

Author Rebuttal to Initial comments

We thank all the reviewers for their input as we believe their suggestions have substantially improved the paper. Here are some general remarks followed by a point-by-point answer to each reviewer's comment.

1) Reviewer #2 suggested that “transfer” of cargoes from ARF1 compartments to recycling endosomes may be due to maturation. This was the first model we proposed as we embarked

into the project. Unfortunately, we were never able to test it due to not having two fluorophores bright and photostable enough to clearly follow the same compartment over a period of time of various minutes. The development of a monomeric variant of green fluorescent protein with great photostability (<https://doi.org/10.1038/s41592-023-02085-6>) called mStayGold pushed us to revisit this idea/model. Indeed, ARF1 compartments shed ARF1 and acquire more Rab11 overtime (New Figure 6 and Figure 8D and F). We believe this is a fundamental discovery that explains why cargoes are first seen in the Golgi and ARF1 compartments and then in recycling endosomes. This mechanism may also explain recycling endosome biogenesis. In the revised paper we have put more emphasis on the maturation model and moved the more speculative part about the interaction of ARF1 compartments with recycling endosomes in the supplement. Introduction and discussion have also been re-focused on the maturation aspect.

2) Reviewer #1 asked us how Rab6 Golgi-to-PM direct carriers mesh into our story. The new data is presented into a new Extended Data Figure 3. We found that ~ 50% of the tubules emerging from the Golgi are positive for Rab6 and the other half are positive for both Rab6 and ARF1. This suggests that there may be more flavors of Rab6 carriers with direct Golgi to PM carriers (Rab6 only) and carriers that are marked by Rab6 and ARF1 which mature into recycling endosomes.

3) A better quantification of the co-localization of ARF1 with organelle markers is now presented in Figure 5B.

4) Reviewer #2 asked to better specify whether the compartments we are looking at are Golgi-derived or peripheral. We have made text changes throughout the manuscript and introduced the distinction into different classes of tubules earlier on in the manuscript. Additionally, we have split the characterization of the two classes of tubules over separate figures. Figure 7 and Extended Data Figure 6 are now dedicated to the characterization of perinuclear compartments and Figure 8 and Extended Data Figure 7 for peripheral compartments.

5) Reviewer #2 asked us to place the peripheral ARF1 compartments on the endocytic recycling pathways. We have quantified the kinetics of Tf_n uptake and we see that ARF1 compartments are downstream of EE positive for Rab5 (Figure 8C). We have also added time lapse microscopy data to show that ARF1 compartments are not sub-domains of EEs or form from EEs (Extended Data Figure 7D).

6) Reviewer #1 had the concern that the transferrin receptor RUSH cargo we observed in recycling endosomes was simply transferrin receptor that had reached the plasma membrane and was being endocytosed. We have included data with more RUSH cargoes to show that secretory cargoes visit recycling endosomes en route from the Golgi to the PM (Figure 7 and Extended Data Figure 6).

To guide the reviewers through the revised figures we shortly explain changes in the layout:

Main document:

Figure 1-4: none or only minor changes.

New Figure 5: The figure was changed to highlight the fact that we observe ARF1 compartments that are positive for Rab11 (Fig. 5Aiii) and ARF1 compartments that transiently interact with recycling endosomes (Fig. 5Aiv). Fig. 5B shows the quantification of the overlap between peripheral ARF1 compartments and the various markers used.

Old Fig. 5B > New Fig. 5D

Old Fig. 5D > New Fig. 5C

Old Fig. 5C and 5Diii > New Extended Data Fig. 5A and B

New Figure 6: New figure highlighting the process of maturation of ARF1 compartments into recycling endosomes.

Old Fig. 6A > New Extended Data Fig. 5C

Old Fig. 6B-C > New Extended Data Fig. 4C-D

Old Fig. 6D > New Fig. 7G

Old Fig. 6E > Deleted as not necessary

New Figure 7: The new Figure 7 now shows the characterization of Golgi-derived ARF1 compartments. We included new data of the RUSH cargo Δ LAMP1, which lack the endocytic motif and cannot be re-endocytosed (New Fig. 7B, C and E). In addition, we moved the AP1 μ A KO data from ARF1/Rab11 KI cells (New Fig. 7G) to this figure, which was previously shown as part of Old Figure 6.

Old Fig. 7E-F > deleted to put more emphasis on the maturation model

Old Fig. 7H > new Fig. 8A

Old Fig. 7I > deleted and replaced with more comprehensive quantification in Fig. 8C

Old Fig. 7J > new Fig. 8E

Old Fig. 7K > new Fig. 8B

New Figure 8: The new Figure 8 displays the characterization of ARF1 compartments involved in endocytic recycling and now includes data that shows maturation of ARF1 compartments which are filled with the endocytic recycling cargo transferrin (New Fig. 8D). In addition, we included data which shows AP-1 on maturing ARF1 compartments (New Fig. 8F). Further we provide new quantification of the overlap of fluorescent transferrin with various markers (New Fig. 8C).

New Figure 9: a new model is provided

Extended Data:

Extended Data Figure 1+2: none or only minor changes.

Extended Data Figure 3: New figure showing the overlap of Golgi-derived ARF1 compartments with Rab6 carriers.

Extended Data Figure 4: Former Extended Data Figure 3. We included data which was previously shown in Figure 6. The data was removed from the main file to improve the flow of the manuscript.

Extended Data Figure 5: New Figure which includes data which was previously shown in Figure 5+6. The figure focusses on the interaction of ARF1 compartments with recycling endosomes, we decided to remove this data from the main figure as we wanted to focus the flow on the maturation of ARF1 compartments into recycling endosomes.

Extended Data Figure 6: Supplement to Figure 7, former Extended Data Figure 4. We included new data to:

- show that a solubleRUSH cargo and a VSVG RUSH cargo also exit the Golgi in ARF1 compartments (Extended Data Fig. 6A).
- show that Golgi-derived compartments are negative for AP-3 (Extended Data Fig. 6C)
- show that a solubleRUSH cargo and a VSVG RUSH cargo are also observed in RE downstream of ARF1 compartments (Extended Data Fig. 6E).

Extended Data Figure 7: Supplement to Figure 8, former Extended Data Figure 5. We included new data showing that ARF1 compartments do not form at early endosomes (Extended Data Fig. 7D).

Reviewer #1:

Remarks to the Author:

Botanelli has previously shown that ARF1 is involved in the formation of long (~3 μm), thin (~110 nm diameter) tubular carriers located beyond the Golgi complex. The tubular carriers are largely free of COPI. In a recent JCB paper, she and her colleagues reported that ARF1 and ARF3 localize to segregated nanodomains on the trans-Golgi network (TGN) and are found on TGN-derived post-Golgi tubules, with Arf1 and Arf3 co-localizing with Golgin 97. Her work showed that ARFs define tubular-vesicular clathrin-positive structures on the TGN and the cell periphery. This submission further characterizes the post-Golgi tubules and highlights their ARF1 content.

New here is beautiful confocal microscopy combined with FIB-SEM and live cell STED. She shows that AP-1 but not AP-3 recruits clathrin to ARF1 compartments, clarifying a question in the field. The tubules do not have Rabs 5, 6 or 7. She then reports using live cell STED that recycling endosomes are tethered to ARF1 compartments. AP1 clusters more with Arf1 than Rab11 but Rab11/ARF1 extended tubules form in AP1 knockout cells for unknown reasons.

A critical experiment is presented in Figure 7, where the authors attempt to use the RUSH system to show that the ARF1 tubules are the carriers of secretory cargo. Importantly, they conclude that 90% of cargo transits through an ARF tubule (7B (is this what is shown?)) and this is messed up if AP1 is knocked out. They monitor the trafficking of the transferrin receptor. Unfortunately, this receptor's normal itinerary is to cycle between the cell surface and recycling endosomes and the authors have not convinced this reader that the pool of TfR seen in REs has not visited the plasma membrane and appeared in an RE after fast recycling—or this cargo that lives in REs may go directly there from the TGN? Here a true secreted cargo or a cell surface resident would be worth testing for comparison.

We understand the concerns of the reviewer.

First, we have tried to clarify the quantification of the RUSH data (now presented in Figure 7C). Here we have quantified the total amount of Transferrin receptor TfR-RUSH tubular-vesicular structures emerging from the Golgi and quantified the percentage of TfR-RUSH tubules which are devoid or positive for ARF1. This quantification does not include cargo that may exit the Golgi in vesicular structures. We have clarified this in the text.

Line 288 “Using two exemplary RUSH cargoes, we then quantified the fraction of the tubulo-vesicular carriers leaving the Golgi that were positive for ARF1. We found that ~ 90% of RUSH cargo-containing tubules were decorated by ARF1 (Fig. 7C).”

As for the second comment we have now included data with other secretory cargoes that would not be endocytosed and accumulate in endosomes (LAMP1 Δ = LAMP1 that is missing endocytic motifs thus accumulates at the PM, VSV-G and soluble signal sequence ssSNAP). For all the cargo analyzed we see strong enrichment into ARF1 tubular-vesicular compartments exiting the Golgi (Fig. 7B and Extended Data Fig. 6A) and RE (Fig. 7E and Extended Data Fig. 6E).

The work deserves to be presented in Nature Cell Biology because of the beautiful microscopy and potential novelty but the story is missing two things.

1. Rab6 tubules have been seen to depart from the Golgi and travel to the surface (Grigoriev, et al. and Akhmanova (2007) Developmental Cell 13, 305-314. How do those tubules/events mesh with the present story?

We thank the reviewer and agree the story would be more complete with additional characterization and comparison with the Rab6 pathway. We believe not all Rab6 carriers emerging from the Golgi are alike and that there may be functionally different Rab6-positive structures. As shown by the now carefully quantified overlap between ARF1 and other markers (Figure 5B), there's very little overlap between peripheral ARF1 and Rab6 positive structures. However, when we looked closely at the tubules emerging from the Golgi (new Extended Data Fig. 3), ~ half of the tubules appeared to be positive for Rab6 while the other half were positive for both Rab6 and ARF1 (Extended Data Fig. 3A-B). The ARF1+Rab6 tubules emerging from the Golgi appeared to be decorated by AP-1 (Extended Data Fig. 3C). This suggests that there may be more flavors of Rab6 carriers with direct Golgi to PM carriers (Rab6 only) and carriers that harbor Rab6 and ARF1 which mature into recycling endosomes. We also now cite the relevant publications (reference #40-41).

Line 217 “The high degree of colocalization at the Golgi prompted us to test the extent of overlap between Golgi-derived ARF1 compartments and Rab6 carriers (Extended Data Fig. 3). Live-cell STED and confocal time lapses show that about half of the Golgi-derived compartments are positive for ARF1 and Rab6, while the remaining half are Rab6-only carriers devoid of AP-1 (Extended Data Fig. 3A-C). This suggests the presence of functionally distinct classes of Rab6 carriers. Rab6-only tubules may be the direct Golgi-to-plasma membrane (PM) carriers, which have been reported previously⁴¹.”

Classical AP1 cargoes such as mannose 6-phosphate receptor should be imaged to tie that sorting into the story. The authors need to better describe how they determined that all cargo goes via these tubules.

We have tested a mannose-6-phosphate RUSH cargo and in agreement with Chen, Gershlick et al. 2017 (doi: 10.1083/jcb.201707172) we observe M6PR leaving the Golgi in vesicular structures which were not associated with ARF1 compartments and not exiting via ARF1 tubular compartments. Chen, Gershlick were also able to show vesicular structures carrying M6PR which were positive for GGA1 (Chen, Gershlick et al. 2017). Therefore, we decided not to

include the data in the paper. We performed RUSH experiments with a M6PR cargo ourselves and a frame of a time lapse 30 mins post-addition of biotin is provided for the reviewer in Figure R1.

Figure R1. GFP-SBP-CD-M6PR was expressed in ARF1-Halo KI HeLa cells labeled with JFX650-CA. Scale bars are 5 μm in the overview and 1 μm in the crop.

2. The authors have not proven that the secretory cargoes need to undergo a separate sorting step out of these tubules, to be delivered to the cell surface or that the Golgi-derived tubules use kiss and run with REs en route to the cell surface. The title states transfer between compartments but only external interaction is actually shown. To see kiss and run, the authors could try to fill the REs with fluorescent dextran and try to detect mixing in live cells?

We thank the reviewers for pushing us to test the model further. We have fed cells with fluorescent dextran and transferrin and we were never able to observe direct transfer between the two organelles. Nevertheless, we expect that it would be rather complicated to observe transfer between two fast moving objects dynamically interacting. Not seeing does not mean that it is not happening... But these negative results pushed us to test alternative models/scenarios.

When we first started characterizing ARF1 compartments we thought that ARF1 compartment could lose ARF1 to mature into recycling endosomes. We were never able to test our hypothesis further because only 1 fluorophore (JFX₆₅₀ used to label Halo tag fusions) was photostable for long enough to be able to detect such events. The development of mStayGold, an incredibly photostable GFP variant, pushed us to revisit this model. For that we generated an ARF1-mStayGold / Halo-Rab11 KI cell line which allowed us to follow both markers in fast live-cell confocal experiments (new Fig. 6). We saw that ARF1 compartments shed ARF1 over a period of ~ 7-9 seconds. As reviewer 2 also pointed out, the extent of co-localization of cargoes

with Rab11-positive RE did not indeed hint to a full fusion event. This cargo transition is better explained by a maturation step. We revisited our model to put more emphasis of the new maturation findings. We were able to show that an ARF1 compartment loaded with fluorescent transferrin matures by shedding the ARF1 coat (Fig. 6) and we were able to show that the last parts of the compartment to shed ARF1 are the highly curved ends of the compartments where AP-1/clathrin localize (Fig. 6, yellow arrows and Fig. 8D and F). While we believe that the new model better explains the transition of cargo from ARF1 compartments to RE, we think that the close interactions between ARF1 compartments and RE (Fig. 5C-D and new Extended Data Fig. 5) may have a functional meaning, for example for the acquisition/transfer of proteins that would promote maturation. However, this is a very speculative model which we have moved away from the main message of the paper. We are currently investigating a possible mechanistic explanation for the close interaction.

Nevertheless, the experiments are state of the art in terms of all the knockin cell lines analyzed and the microscopy is quite beautiful.

Other comments

Line 156 “fusion proteins”? AP proteins are cargo adaptors; Line 199. “ARF1 compartments are multi-functional sorting endosomes”. That has not yet been shown, especially at this stage of the paper.

The term fusion protein rather describes the fact that a protein is fused to an imaging tag. We changed the wording to avoid confusion.

Line 152 “We observed the same localization pattern when we endogenously tagged the large AP1 γ 1- or AP3 δ 1-subunit (Extended Data Fig. 2A-B) suggesting that the placement of the tag within the complex does not impact AP localization.”

We changed the wording to avoid earlier overinterpretation of the data.

Line 204 “To conclude, ARF1 compartments harbor AP-1 and AP-3 nanodomains, with clathrin being recruited exclusively to AP 1 nanodomains. The different classes of tubules, harboring different classes of adaptors may have a role in channeling different cargoes from ARF1 compartments into segregated downstream pathways.”

Line 216 should be Fig. 5D? Line 140 she concludes that ARF1 compartments are the major site of clathrin dependent intracellular sorting but this is premature since no cargoes are visualized here—clear is clathrin at the tips of tubules as seen in the JCB paper. The authors should be careful because they may be tracking clathrin assemblies that are static.

Figure 5 was re-organized and text was changed

We changed the text to make no premature statements:

Line 138 “In summary, these data identify ARF1 compartments as the major site of clathrin recruitment.”

Reviewer #2:

Remarks to the Author:

This paper reports an extensive live-cell imaging analysis of the dynamics of Arf1-positive

membranes in a mammalian tissue culture cell line. The authors examine the association with the membranes of the clathrin adaptors AP-1 and AP-3, with secreted and endocytic cargo, and their association with other organelles. They conclude that there are two classes of Arf1-positive tubular membranes, one which emanates from the Golgi, and one which receives endocytic cargo, and both have associated AP-1 and AP-3 structures. They find that the exocytic Arf1-positive tubules contact Rab11-positive tubules and suggest that exocytic cargo is transferred between the two in order to facilitate secretion.

This is study of a high technical quality with care taken to tag proteins in the genome to avoid over-expression artefacts, and the data is carefully quantified. However, it also has three area of concern. Overall, I feel that it has the potential to make an excellent study and so I will initially describe the issues of concern, and then outline what could be done to improve the paper.

1) A few of the conclusions have already been published including some by the authors themselves. For instance, the authors have already reported that Arf1 is found on TGN-derived post-Golgi tubules (PubMed ID 37102998), and it has been reported that AP-1 is found on such tubules as well (PubMed ID 12529433). Likewise, the association of clathrin with AP-1- but not AP-3-coated structures has been reported from immuno-EM studies (15051738). Finally, the transfer of secretory cargo to recycling endosomes has also been reported. The authors do cite these studies, even if not always directly, and their approach is more technically advanced and comprehensive, so the conclusions are valuable, but they need to be more clear about what has been reported and what is new.

We have re-written parts of the text to highlight what has been previously reported by us and other labs. Figure 1 panel A is not new, but we believe it is an important piece of information to understand the rationale of further characterization and experiments. Text was re-written accordingly.

Line 102 “First, we recapitulated our initial observations²³ and we applied super-resolution live-cell STED microscopy on ARF1^{EN}-Halo/SNAP-CLCa^{EN} (EN=endogenous) knock-in (KI) HeLa cells to highlight the close association of clathrin with ARF1 compartments (Fig. 1A).

Additionally, we observed ARF1-positive clathrin compartments in various cell types (Extended Data Fig. 1A-B). To further characterize ARF1 compartments, we first wanted to test whether non-endocytic clathrin is exclusively associated with ARF1 compartments.”

Line 55 “Various secretory and endocytic recycling cargoes have been found to transit through tubulo-vesicular compartments^{10,11}, some of which were shown to be decorated with clathrin and AP-1¹²⁻¹⁴. In particular, secretory cargo flow from the Golgi to the plasma membrane may occur via direct tubular carriers or clathrin-decorated tubules that deliver their content to endosomes via yet unknown mechanisms^{13,15}. Tubulo-vesicular endosomes harboring the adaptors AP-1 and AP-3 as well as clathrin have been identified using immuno-EM but have not been characterized in detail^{16,17}.”

Line 98 “Previous studies have identified TGN-derived tubulo-vesicular compartments defined by the small GTPase ARF1^{13,23}.”

2) The evidence in the paper for transfer of cargo from Arf1 tubules to Rab11-positive recycling endosomes is open to alternative interpretations, and their speculation on the mechanism for any such transfer is just that, speculation, given that the light microscopy imaging that they use lacks the resolution to determine if membranes are close, directly contacting or transiently

fused. All that they can really say is that the AP-1 coat seems to be often present in the places where the two compartments are closest, and as they note themselves, such a coat is just as likely to inhibit fusion as to promote it. The moments of close proximity of the two compartments could simply reflect that fact that both are moving on microtubules and so one of each structure might transiently be on the same microtubule. Another possibility that is not discussed is that the compartments mature and Arf1 is replaced by Rab11, analogous to the Rab transition or conversion seen in the endocytic system. Certainly, none of the movies presented showed a flooding of cargo from an Arf1-positive carrier into the full extent of a Rab11-positive tubular structure as might be expected from a fusion event between the two.

We thank the reviewers for pushing us to test the model further. As suggested by reviewer 1, we fed cells with fluorescent dextran and transferrin and we were never able to observe direct transfer between ARF1 compartments and RE. Nevertheless, we expect that it would be rather complicated to observe transfer between two fast moving objects dynamically interacting. And indeed, we agree that none of the trafficking assays pointed to flooding cargo from ARF1 compartments to REs.

As replied to reviewer #1, when we first started characterizing ARF1 compartments we thought that ARF1 compartment could lose ARF1 to mature into recycling endosomes. We were never able to test our hypothesis further because only 1 fluorophore (JFX650 used to label Halo tag fusions) was photostable for long enough to be able to detect such events. The development of mStayGold, an incredibly photostable GFP variant, pushed us to revisit this model. For that we generated an ARF1-mStayGold / Halo-Rab11 KI cell line which allowed us to follow both markers in fast live-cell confocal experiments (new Fig. 6). We saw that ARF1 compartments shed ARF1 over a period of ~ 7-9 seconds. This cargo transition is better explained by a maturation step. We revisited our model to put more emphasis on the new maturation findings. We were able to show that an ARF1 compartment loaded with fluorescent transferrin matures by shedding the ARF1 coat (Fig. 6) and we were able to show that the last parts of the compartment to shed ARF1 are the highly curved ends of the compartments where AP-1/clathrin localize (Fig. 6, yellow arrows and Fig. 8D and F). While we believe that the new model better explains the transition of cargo from ARF1 compartments to RE, we think that the close interactions between ARF1 compartments and RE (Fig. 5C-D and new Extended Data Fig. 5) may have a functional meaning, for example for the acquisition/transfer of proteins that would promote maturation. However, this is a very speculative model which we have moved away from the main message of the paper. We are currently investigating a possible mechanistic explanation for the close interaction.

3) The authors present clear data to show the Arf1 and AP-1/3 positive membranes are of at least two types. One is perinuclear and consists of tubules derived from the TGN and carries exocytic cargo. The other is peripheral and is accessed by endocytosed transferrin. They also show that the two have different dependencies on AP-1 for their fission. This is an interesting observation, but it is introduced only at the end of the paper, and in many of the previous figures the authors quantify aspects of the Arf1-positive membranes without making it clear if they are the peripheral endocytic type or the perinuclear exocytic type.

We have introduced the possible functional differentiation in AP-1 perinuclear and peripheral AP-1 and AP-3 positive compartments early on and tried to make sure to clearly state which compartment we are talking about/showing throughout the whole manuscript.

Line 169 “In particular, ARF1 compartments that were seen in the perinuclear area and emerging from the Golgi were only positive for AP-1. In contrast, tubules observed in the cell periphery were found to be positive for both AP-1 and AP-3 (Fig. 3A-C).”

To better explain the functional characterization of different classes of tubules we have now split the results into two figures. Figure 7 and 8 now shows results about the characterization of perinuclear and peripheral compartments respectively. Figure 7 has an associated supplement now called Extended Data Figure 6 while supplementary information for Figure 8 is displayed in Extended Data Figure 7.

Additionally, we have included a panel in Extended Data Figure 6C showing that secretory RUSH tubules are devoid of AP-3. In contrast, endocytic recycling ARF1 compartments harbor both AP-1 and AP-3 (Extended Data Fig. 7A-C).

Despite these concerns I would not wish to see this studied rejected as it undoubtedly has positive features. The first four figures are really good and provide an excellent overview of the dynamics of AP-1 and AP-3 coated pits, and the authors make an excellent point that previous studies that observed AP-1-coated structures moving away from the TGN may have erroneously concluded that they were budded vesicles when in fact they are buds on tubules leaving the TGN. Indeed, this is relevant to recent suggestions from yeast studies that AP-1 may be entirely dedicated to retrieval of proteins to the Golgi rather than export from the Golgi as has been widely assumed. With some more experiments this part of the paper could be expanded to make a very valuable contribution to the field that would stand on its own without the more speculative Rab11 interactions. I would suggest the following:

A). The authors do not consider the GGA coat proteins which also act in this system and may be independent of AP-1 and AP-3. They seem well placed to resolve controversies in this area and so I suggest that they also tag one of the GGA proteins and compare it to Arf1, AP-1, AP-3 and clathrin. It would also be informative to determine if GGAs are present on the peripheral endocytic Arf1/AP-1-positive structures.

We agree with the reviewer that the GGAs are important adaptors to consider in the context of AP-1, clathrin and ARF1-positive membranes. We are currently working on a manuscript in which we elucidate the role of GGAs in post-Golgi transport. [REDACTED].

[REDACTED]

B). The peripheral structures that are labelled with endocytosed markers are interesting. How are these formed? By following a pulse of transferrin uptake, it may be possible to determine at what stage of the endocytic recycling the ligand goes through this compartment, and if the tubules are derived from Rab5-positive early endosomes. In all the experiments the authors need to make clear if the Arf1-positive structures they show or quantify are of the peripheral or perinuclear kind.

We have now included a more extensive quantification showing the kinetics of fluorescent transferrin uptake. We have assessed the co-localization over time with markers for early endosomes (Rab5), ARF1 and recycling endosomes (Rab11, Figure 8C). We show that Tfn fills ARF1 compartments downstream of EE. Furthermore, ARF1 and Rab5 positive EE define different structures (Figure 5B) and move independently from each other (confocal time lapses in Extended Figure 7D) suggesting that ARF1 compartments are not derived from EE or are not simply sorting sub-domains of EE. Figure 8 is now dedicated to the characterization of the peripheral endocytic compartments making it easier to differentiate.

C). Cell lines can be generated that lack Arf1 and Arf-3 (PubMed ID 34749397). The could authors look at such cells to see if the tubules still exist and if any of the coats are on them. We had already performed experiments in ARF KO cell lines and found that recruitment of AP-3 but not AP-1 is impaired in ARF1 KO cells (Fig. R3A, notice the higher cytoplasmic background and less AP-3 punctae observed in ARF1 KO cells). Upon KO of ARF3, which localizes to the same compartments as ARF1 (Wong-Dilworth et al., 2023), both AP-1 and AP-3 are recruited normally. KI of ARF3 in ARF1 KO cells shows that tubular structures still form in the absence of ARF1 (Fig R3C). We are unsure about the underlying mechanisms but assume that ARF3 can compensate for the loss of ARF1 here. However, ARF3 cannot compensate for the loss of ARF1 in the case of AP-3 recruitment to the compartment. We think this information does not add to the message of the manuscript, as it raises more questions than it answers. If the reviewer believes, that this information is relevant, we would be happy to integrate it into the manuscript.

Figure R3: Tubule formation and AP-1 recruitment are not affected by ARF1 KO.

(A) Live-cell confocal imaging of AP1 μ A-Halo and AP3 μ A-Halo in WT, ARF1 KO and ARF3 KO HeLa cells labeled with BG-JFX₆₅₀. AP-1 and AP-3 were endogenously expressed. Imaging shows that recruitment of AP-3 but not AP-1 is impaired in ARF1 KO cells. (B) Western Blots showing endogenous expression of tagged AP-1 and AP-3 μ -subunits in ARF1 and ARF3 KO cells. Whole cell lysates were stained with anti-Halo to detect the fusion protein. (C) Live-cell confocal imaging of ARF3-LAP(Halo)/ARF1 KO HeLa cells showing existence of tubular structures marked by ARF3 in the absence of ARF1. Scale bars: 20 μ m (A), 10 μ m (C).

D). The cargo transfer aspects of the work are not well proven. The Rab11 experiments should either be removed for a later publication, or discussed briefly with various possible conclusions mentioned as interpretations that are not yet resolved. If the Rab11 data is included it would be valuable to examine both the perinuclear/exocytic and the peripheral/endocytic Arf1-positive compartments.

We agree with the reviewer and as explained above we have re-focused the manuscript on the maturation mechanism which we believe has a more solid proof. Nevertheless, we decided to include the data about the interaction as Extended Data Figures and write a very reduced and speculative model about what the function of the interaction could be.

E) There are several more minor aspects that also need to be addressed:

1) The authors should more clearly discuss the evidence, direct or indirect, that the tagged proteins are functional.

We changed the text to be more clear what conclusions we draw from the control experiments. Line #152 “We observed the same localization pattern when we endogenously tagged the large AP1 γ 1- or AP3 δ 1-subunit (Extended Data Fig. 2A-B) suggesting that the placement of the tag within the complex does not impact AP localization. Furthermore, simultaneous tagging and visualization of medium and large subunits of AP complexes showed both subunits to colocalize within the same nanodomains, indicating that AP complexes still form when one or more subunits are tagged (Extended Data Fig. 2C-D). In addition, AP-1-dependent clathrin recruitment was unaffected by Halo-tagging of the μ -subunit (Extended Data Fig. 2E-F), indicating that interaction of AP-1 with accessory proteins is unaffected by addition of the imaging tags.”

2) 38% is not a “majority”, but rather the most abundant class of compartment as defined in the experiment (Figure 3C).

We changed the text accordingly.

Line #167 “The most abundant class of ARF1 compartments is decorated with AP-1 and AP-3 (~38%) (Fig. 3C, Di, E).”

3) Single channel images should be shown as grey scale as this is easier to see and the intensities are clearer (eg Figure 1B insets, 2C, 3A insets, 3B insets, 3D etc).

We changed all single channel images to grey scale.

4) In the discussion the authors should mention and briefly discuss a paper in bioRxiv from Margaret Robinson which addresses similar issues, even if not so expertly (<https://doi.org/10.1101/2023.11.02.565271>).

We have added referenced to the Robinson lab paper in the introduction and the discussion (reference 14). We also further discuss their findings as we believe that the new model is compatible with a role for AP-1 in retrograde transport from ARF1 compartments,

Line #386 “Recently, AP-1 was proposed to act exclusively in retrograde transport of cargoes from endosomes back to the Golgi¹⁴. Our data show that AP-1 localizes solely on the Golgi and ARF1 compartments. Assuming that the role of AP-1 is connected to protein transport back to the Golgi apparatus, it is conceivable that a compartment may have to retrieve all retrograde cargoes before becoming competent for transport to the PM (Fig. 9). AP-1 could sequester AP-1 cargoes from ARF1 compartments, whether they have escaped the Golgi (perinuclear compartments) or upon internalization after endocytosis (peripheral compartments).”

Reviewer #3:

Remarks to the Author:

The manuscript by Stockhammer et al. describes the identification of a new ARF1-positive post-Golgi compartment that serves as an intermediate for transferring secretory cargo between the TGN and the recycling endosome (RE). The work is of very high technical quality and uses cutting edge technology. The genetically engineered cell lines will be of value for others in the community. Having said this, there are several issues that must be addressed before this work can be accepted for publication. Besides technical points, I think the overall conclusion is not solid enough and other interpretations are also possible. In addition, even if the basic observation were correct, the work does not provide mechanistic insight into the broader role of this ARF1-positive compartment. Thus, the conceptual advance is in my personal opinion relatively small.

1- What is the functional significance of this ARF1 compartment? Does it transport specific cargos? If not, what fraction of secreted proteins depends on ARF1 for their trafficking?

For a long time, it was thought that Clathrin coated vesicles would bud from the TGN for transport of cargoes to endosomes. As we have hoped to make clear in the introduction of the paper, we believe this is because Clathrin or AP-1 signal (which indeed looks like punctae moving away from the Golgi) were never observed with the underlying membrane below (ARF1 compartments). Here we show that a pulse of secretory cargo is incorporated into peri-nuclear ARF1 compartments for export and a pulse of endocytic cargo visits the peripheral ARF1 compartments. We also show different functionally distinct compartments with different APs identities. In addition, we are now able to show that some of the ARF1 compartments undergo a maturation step and mature into recycling endosomes, suggesting that maturation is what drives the flow of cargo. This suggests that cargo would flow in an anterograde manner while AP-1 would sequester cargo to be retrieved back to the Golgi (Model in Figure 9). We believe that this is a fundamental discovery that 1) provides a compelling model for the distribution of cargoes in post-Golgi routes 2) provide a possible explanation for the biogenesis of recycling endosomes 3) explains why some secretory cargoes have been seen taking a detour to endosomes before reaching the plasma membrane which has important implications for the understanding of polarized trafficking. While for now we have proven our model with model cargo molecules we look forward to biochemically purifying these organelles to look at their physiological content.

2- Fig1A: this result needs to be quantified, otherwise it remains anecdotal in nature. Specifically I would suggest to assess

(i) how many of the green puncta are associated with an ARF1 tubule?

(ii) Are there qualitative differences between the green puncta that are free vs. those that are associated with ARF1-tubules? To me, it appears that the CLC structures associated with ARF1 are smaller than the free puncta

We hope to be able to clarify this point. The structures that are not associated to ARF1 tubules are most probably AP-2 positive structures on the plasma membrane. We believe that the quantification the reviewer refers to is already provided in Figure 1C. We created an AP-2/ARF1/Clathrin triple KI (Figure 1B) and quantified the amount of ARF1 associated structures vs free clathrin structures (Figure 1C). We found only a very small portion of clathrin not associated with either ARF1 compartments/or AP-2 on the plasma membrane.

The free green puncta that the reviewer refers to in Figure 1A are most probably clathrin-coated pits at the plasma membrane, therefore they are larger in size compared to the ones associated

with ARF1 compartments as clathrin coated pits at the plasma membrane are 100-200 nm in size, bigger than a clathrin coated bud at the endosomes/Golgi which is 70-80 nm in size (also quantified in Fig. 2D). When comparing clathrin structures that are neither associated to AP-2 nor ARF1 (counted for Fig. 1C) to ARF1-associated structures, we do not observe any differences in their size.

3- The authors need to properly quantify the fission events, as seen for instance in Fig.1F. In this figure, I see a large tubule (much longer than the one the authors are focusing on) that does not detach. Moreover, ARF1-tubules have several spots of CLC or AP1 on them, but only some of these puncta become a fission point. The authors propose a model that CLC and or AP1 play a role in the fission of ARF1 tubules. If this were true, then why are there more puncta than fission events? This actually points to the fact that neither CLC nor AP1 are major/essential players in the observed fission event. The images that the authors provide clearly indicate that this is the case. This is one of the instances, where I think that the authors have jumped prematurely to a conclusion and are therefore proposing a model that is likely to be incorrect.

We believe we have already quantified the fission events of ARF1 compartments:

Line #120 “We analyzed more than 100 fission events and found that clathrin was present at >90% of the fission sites (Fig 1G-H, Extended Data Fig. 1C)”.

This analysis was done unbiasedly, by first only looking at the ARF1 channel and identifying fission sites and in a second step, checking these fission sites for presence of clathrin.

Regarding the observation of the reviewer: Not all compartments emerging from the TGN undergo fission. Some tubules extend to retract again. We do want to make clear that AP-1 may have a role in recruiting fission factors to the sites of fission, thus not all the AP-1 punctae will lead to a certain fission event as other factors may be involved. Currently, we do not know how this is regulated. As depletion of AP-1 leads to longer ARF1 compartments (Fig. 4) and both AP-1 and clathrin are present at sites of fission (Fig. 1), strongly suggests that AP-1 is required for recruitment of fission factors.

In lines 118-120 we state “Intriguingly, clathrin localized at the fission site on ARF1 compartments, suggesting that clathrin and associated machinery may be responsible for the recruitment of fission factors.”

We agree with the reviewer that the title for Figure 1 may be mis-leading and we have changed it from “Fission of ARF1 tubules via association of non-endocytic clathrin coat” to “ARF1 compartments are the major site of non-endocytic clathrin assembly”.

4- Fig.4Di shows that AP3 colocalizes with CLC on an ARF1 structure? However, the text of the authors proposes otherwise.

In Fig 4Di an AP-3 structure is shown that is clearly devoid of clathrin. A second smaller AP-3 structure is near a larger clathrin structure as shown by the shifted peaks in the line profile. In our colocalization analysis using the Manders coefficient, we included a negative control (AP-3 vs AP-1) that shows that co-localization of AP-3 and clathrin is comparable to the one of AP-3 with AP-1, indicating that residual overlap of AP-3 with clathrin is likely to be caused by proximity of AP-1 and AP-3 nanodomains. That fact that the Manders of AP-3 with clathrin is slightly higher than the one of AP-3 and AP-1 is likely caused by the higher abundance of clathrin structures compared to AP-1 structures (clathrin coated pits at the plasma membrane).

5- Figure 4: I think that the colocalization in just 11 cells is not sufficient and this number should be increased.

We increased the number to 30 cells per condition.

6- In the paragraph describing Figure 5, the authors write “We found ARF1 compartments to be hubs for adaptor-dependent post-Golgi sorting.” This statement is very pre-mature at this stage of the manuscript. The role of this compartment in sorting has not been tested at all at this stage and even later, other conclusions are possible.

We have changed the text accordingly

Line #210 “We found ARF1 compartments to harbor different AP complexes. As these adaptors were described to associate with different endosomal compartments^{19,23,38} ... “

7- Figure 5A: These data are not properly quantified. The authors pick a region where Rab6 and ARF1 are not in proximity. However, several other regions within the same picture do show colocalization (or at least close proximity) between ARF1 and RAB6. Therefore, the conclusion of the authors is not correct, because it is based on selective treatment of data.

As shown by the now carefully quantified overlap between ARF1 and other markers (Figure 5B), there’s very little overlap between peripheral ARF1 and Rab6 positive structures. However, when we looked closely at the tubules emerging from the Golgi (new Extended Data Fig. 3), ~ half of the tubules appeared to be positive for Rab6 the other half were positive for both Rab6 and ARF1 (Extended Data Fig. 3A-B). The ARF1+Rab6 emerging from the Golgi appeared to be decorated by AP-1 (Extended Data Fig. 3C). This suggests that there may be more flavors of Rab6 carriers with direct Golgi to PM carriers (Rab6 only) and carriers that harbor Rab6 and ARF1 which mature into recycling endosomes. The new results and figures have been integrated in the new version of the manuscript.

8- Another example for overinterpretation of the data is the sentence: “The localization of AP-1 at the interface between ARF1 compartments and REs suggests a role for AP-1 in cargo enrichment and transfer between closely interacting compartments”. This is pure conjecture and not supported by data.

We have moved the data mentioned by the reviewer into supplementary figures (Extended Data Fig. 4-5) and removed the speculative model. We are currently performing more CLEM and are looking forward to gaining a better understanding of what is happening at the ARF1 compartments/RE interface.

9- Figure 6: I understand that longer ARF1 tubules would be expected in the absence of AP1, because the authors propose that AP1 mediates the fission of ARF1 tubules. However, why would Rab11 also form tubules? This indicates that something else is going on, for instance as an effect on lipid homeostasis due to a depletion of AP1. The fact that the tubules are now completely double-positive for ARF1 and RAB11 is also strange? Have the two compartments (i.e- ARF1 and RE) collapsed to one?

Based on the new proposed model, we believe that double positive ARF1 and Rab11 tubules form due to the maturation defects. We hope this is a more compelling model for the reviewer.

10- A role for AP1 as a diffusion barrier that prevents content mixing is not at all supported by the data and is also not in line with the model of the authors. Such content mixing would require that the RAB11 and ARF1 compartments to be continuous to start with (which they are not). How can a diffusion barrier act between two separate compartments.

As explained above we have removed this speculative model and focused on the maturation, which better explains our data.

11- Figure 7 and Ex. Figure 5: Why is the RUSH cargo located at the tip of the ARF1 tubules? It appears as if the RUSH cargo would be mostly outside the tubules, as if these were two structures that are glued to each other and that ARF1 tubules is at the back of the forward moving cargo.

We have removed the old Extended Data Fig. 5D. as we agree it was confusing. While there is RUSH cargo on the tubule, there were many bright structures close to the tubule making it hard for the reader to appreciate the enrichment on the tubule itself. The old figure in grey scales is provided for the reviewer.

ARF1^{EN}-Halo / TNF α -RUSH

Old Extended Data Fig. 5D.

12- The observations at the end of the manuscript would also be consistent with AP1 being involved in exporting cargo from the ARF1/RAB11 compartment. In the absence of AP1, the compartment remains double positive. This alternative explanation cannot be ruled out by the data. This jeopardizes the entire model.

We agree with the reviewer and we believe the new model fits better the observations. AP-1 could sequester AP-1 cargoes from ARF1 compartments, whether they have escaped the Golgi (perinuclear compartments) or upon internalization after endocytosis (peripheral compartments). Once retrograde cargoes are exported from ARF1 compartments, AP-1 would dissociate from the membrane. Depletion of AP-1 may therefore prevent maturation.

13- The RUSH data are not fully convincing. The rise in fluorescence in RAB11 positive structures does not coincide with a drop of fluorescence in the ARF1 compartment.

We believe the reviewer refers to the graph that is now presented in Fig. 7F. Unfortunately, due to photobleaching we could not follow the cargo for longer when performing higher frame rate microscopy. We believe that the not so convincing drop in fluorescence for the ARF1 channel may be due to backed up flow, which is a limitation of using the RUSH system.

Decision Letter, first revision:

Our ref: NCB-A52758A

8th August 2024

Dear Dr. Bottanelli,

Thank you for submitting your revised manuscript "ARF1 compartments direct cargo flow via maturation into recycling endosomes" (NCB-A52758A). It has now been seen by the original referees and their comments are below. The reviewers find that the paper has improved in revision; while Rev#3 had reservations about the depth of understanding of the role of ARF1, they also appreciated the quality of the study and could see the interest and importance in the data. The other two reviewers support publication. Therefore, we'll be happy in principle to publish the manuscript in Nature Cell Biology pending minor revisions to satisfy the referees' final requests and to comply with our editorial and formatting guidelines. Please do address all remaining comments from the reviewers (in particular Rev#1's points) by added text/discussion to the manuscript, edits to the figures as appropriate. No further experimentation is needed in our view.

We are now performing detailed checks on your paper and will send you a checklist detailing our editorial and formatting requirements in about 1-2 weeks. Please do not upload the final materials and make any revisions until you receive this additional information from us.

Thank you again for your interest in Nature Cell Biology. Please do not hesitate to contact me if you have any questions.

Sincerely,

Melina

Melina Casadio, PhD
Senior Editor, Nature Cell Biology
ORCID ID: <https://orcid.org/0000-0003-2389-2243>

Reviewer #1 (Remarks to the Author):

The authors have very thoughtfully and carefully responded to each of my comments and the manuscript is much improved. My only remaining concern regards the model in Fig. 9. The authors describe 3 types of vesicles departing the TGN: Rab6 only, Rab6+Arf1 and MPR alone. Might those be mentioned or indicated? And where is the Rab11? Not clear that RE's fuse directly with PM--maybe a dashed line? [Also, AP1 retrograde transport passes through a Rab7 compartment or? It's ok if some of the details of this sorting will be addressed in other work.]. Overall well done!

Reviewer #2 (Remarks to the Author):

This is a revised version of a paper that uses advanced live-cell imaging to examine the dynamics of Arf1-labelled membranes at the Golgi and endosomes. I felt that the original version had a lot of positive features but felt that more analysis was needed and that the proposed rapid transfer of cargo from an Arf1-positive compartment to a Rab11-positive compartment might instead be a maturation process. In revising this paper, the authors have done a really excellent job of engaging with my comments and suggestions. They have added more analysis, and have separately analysed events at peripheral and central compartments which adds clarity and rigour. They have also fully engaged with the notion of maturation (as was also suggested by Reviewer 1), and have now added compelling data that this is indeed what is happening. Arf1 labelled compartments mature over time to be Rab11-positive recycling endosome compartments. This is a potentially very important finding that would provide a much-needed explanation for the relationship between the trans-Golgi network, the recycling endosome and early endosomes. The results are presented with clarity and with an admirable degree of balance by explaining what remains speculative and what is well proven.

Given all this I am very happy to recommend this excellent manuscript for publication and would like to commend again the authors for their open-minded approach and willingness to engage fully with reviewer comments.

Reviewer #3 (Remarks to the Author):

The authors present an improved revised manuscript with a new working model. They addressed all my initial comments either experimentally, or just verbally. The current model for trafficking from the Golgi is that AP1/clathrin control export from the TGN. The conclusion of the revised manuscript (new model as the authors call it) is that AP1 is the factor responsible for cargo sequestration from the ARF1 compartment. I am not sure how this differs from the existing trafficking model. The ARF1-compartment is part of the TGN, and not an independent compartment. Thus, the new model plainly says that AP1 controls export from the TGN, which is not much different from the current model. I am therefore not sure what novelty this manuscript provides.

The main finding of this manuscript is that ARF1 defines a domain of the TGN where exocytic cargo is segregated for export. How ARF1 performs this job remains unclear. It might alter the physical properties of the membrane (curvature), or it might recruit lipid modifying enzymes, or it might recruit a fission machinery together with AP1 (as the authors claim themselves). However, none of these working hypotheses are tested against each other.

I really appreciate and love the technical excellence and elegance of this work. The authors also improved the data in the revised manuscript. Furthermore, I think that the discovery of ARF1 as a definer of something like an "export domain" at the TGN is interesting and important. Nevertheless, I think that the work remains descriptive and lacks mechanistic insight. Even if ARF1 were defining a

genuine compartment, my initial concern that the current work does not define a physiological cargos, which are depend on this ARF1 compartment is not addressed.

Decision Letter, second revision:

Our ref: NCB-A52758A

15th August 2024

Dear Dr. Bottanelli,

Thank you for your patience as we've prepared the guidelines for final submission of your Nature Cell Biology manuscript, "ARF1 compartments direct cargo flow via maturation into recycling endosomes" (NCB-A52758A). Please carefully follow the step-by-step instructions provided in the attached file, and add a response in each row of the table to indicate the changes that you have made. Ensuring that each point is addressed will help to ensure that your revised manuscript can be swiftly handed over to our production team.

In recognition of the time and expertise our reviewers provide to Nature Cell Biology's editorial process, we would like to formally acknowledge their contribution to the external peer review of your manuscript entitled "ARF1 compartments direct cargo flow via maturation into recycling endosomes". For those reviewers who give their assent, we will be publishing their names alongside the published article.

Nature Cell Biology offers a Transparent Peer Review option for new original research manuscripts

submitted after December 1st, 2019. As part of this initiative, we encourage our authors to support increased transparency into the peer review process by agreeing to have the reviewer comments, author rebuttal letters, and editorial decision letters published as a Supplementary item. When you submit your final files please clearly state in your cover letter whether or not you would like to participate in this initiative. Please note that failure to state your preference will result in delays in accepting your manuscript for publication.

Cover suggestions

COVER ARTWORK: We welcome submissions of artwork for consideration for our cover. For more information, please see our guide for cover artwork.

Nature Cell Biology has now transitioned to a unified Rights Collection system which will allow our Author Services team to quickly and easily collect the rights and permissions required to publish your work. Approximately 10 days after your paper is formally accepted, you will receive an email in providing you with a link to complete the grant of rights. If your paper is eligible for Open Access, our Author Services team will also be in touch regarding any additional information that may be required to arrange payment for your article.

Please note that *Nature Cell Biology* is a Transformative Journal (TJ). Authors may publish their research with us through the traditional subscription access route or make their paper immediately open access through payment of an article-processing charge (APC). Authors will not be required to make a final decision about access to their article until it has been accepted. Find out more about Transformative Journals

For information regarding our different publishing models please see our Transformative Journals page. If you have any questions about costs, Open Access requirements, or our legal forms, please contact

ASJournals@springernature.com.

[Redacted]

Best regards,

Kendra Donahue
Staff
Nature Cell Biology

On behalf of

Melina Casadio, PhD
Senior Editor, Nature Cell Biology
ORCID ID: <https://orcid.org/0000-0003-2389-2243>

Reviewer #1:

Remarks to the Author:

The authors have very thoughtfully and carefully responded to each of my comments and the manuscript is much improved. My only remaining concern regards the model in Fig. 9. The authors describe 3 types of vesicles departing the TGN: Rab6 only, Rab6+Arf1 and MPR alone. Might those be mentioned or indicated? And where is the Rab11? Not clear that RE's fuse directly with PM--maybe a dashed line? [Also, AP1 retrograde transport passes through a Rab7 compartment or? It's ok if some of the details of this sorting will be addressed in other work.]. Overall well done!

Reviewer #2:

Remarks to the Author:

This is a revised version of a paper that uses advanced live-cell imaging to examine the dynamics of Arf1-labelled membranes at the Golgi and endosomes. I felt that the original version had a lot of positive

features but felt that more analysis was needed and that the proposed rapid transfer of cargo from an Arf1-positive compartment to a Rab11-positive compartment might instead be a maturation process. In revising this paper, the authors have done a really excellent job of engaging with my comments and suggestions. They have added more analysis, and have separately analysed events at peripheral and central compartments which adds clarity and rigour. They have also fully engaged with the notion of maturation (as was also suggested by Reviewer 1), and have now added compelling data that this is indeed what is happening. Arf1 labelled compartments mature over time to be Rab11-positive recycling endosome compartments. This is a potentially very important finding that would provide a much-needed explanation for the relationship between the trans-Golgi network, the recycling endosome and early endosomes. The results are presented with clarity and with an admirable degree of balance by explaining what remains speculative and what is well proven.

Given all this I am very happy to recommend this excellent manuscript for publication and would like to commend again the authors for their open-minded approach and willingness to engage fully with reviewer comments.

Reviewer #3:

Remarks to the Author:

The authors present an improved revised manuscript with a new working model. They addressed all my initial comments either experimentally, or just verbally.

The current model for trafficking from the Golgi is that AP1/clathrin control export from the TGN. The conclusion of the revised manuscript (new model as the authors call it) is that AP1 is the factor responsible for cargo sequestration from the ARF1 compartment. I am not sure how this differs from the existing trafficking model. The ARF1-compartment is part of the TGN, and not an independent compartment. Thus, the new model plainly says that AP1 controls export from the TGN, which is not much different from the current model. I am therefore not sure what novelty the this manuscript provides.

The main finding of this manuscript is that ARF1 defines a domain of the TGN where exocytic cargo is segregated for export. How ARF1 performs this job remains unclear. It might alter the physical properties of the membrane (curvature), or it might recruit lipid modifying enzymes, or it might recruit a fission machinery together with AP1 (as the authors claim themselves). However, none of these working hypotheses are tested against each other.

I really appreciate and love the technical excellence and elegance of this work. The authors also improved the data in the revised manuscript. Furthermore, I think that the discovery of ARF1 as a definer of something like an “export domain” at the TGN is interesting and important. Nevertheless, I think that the work remains descriptive and lacks mechanistic insight. Even if ARF1 were defining a

genuine compartment, my initial concern that the current work does not define a physiological cargos, which are depend on this ARF1 compartment is not addressed.

Author Rebuttal, first revision:

We thank all the reviewers for pushing us to improve the manuscript

Reviewer #1 (Remarks to the Author):

The authors have very thoughtfully and carefully responded to each of my comments and the manuscript is much improved. My only remaining concern regards the model in Fig. 9. The authors describe 3 types of vesicles departing the TGN: Rab6 only, Rab6+Arf1 and MPR alone. Might those be mentioned or indicated? And where is the Rab11? Not clear that RE's fuse directly with PM--maybe a dashed line? [Also, AP1 retrograde transport passes through a Rab7 compartment or? It's ok if some of the details of this sorting will be addressed in other work.]. Overall well done!

We have made some changes to the model (now Fig. 8 panel G).

- We have now indicated the Rab6 pathway (magenta arrow)
 - We have clearly indicated that the RE are Rab11-positive organelles (dark grey)
 - We have added a ? to indicate it is unclear if the Rab11-positive endosomes are the last compartments that fuse/deliver cargo to the PM
 - We have decided to not highlight the Manose 6 phosphate pathway as no data was presented in the paper
- The figure legend has been re-written accordingly.

Reviewer #2 (Remarks to the Author):

This is a revised version of a paper that uses advanced live-cell imaging to examine the dynamics of Arf1-labelled membranes at the Golgi and endosomes. I felt that the original version had a lot of positive features but felt that more analysis was needed and that the proposed rapid transfer of cargo from an Arf1-positive compartment to a Rab11-positive compartment might instead be a maturation process. In revising this paper, the authors have done a really excellent job of engaging with my comments and suggestions. They have added more analysis, and have separately analysed events at peripheral and central compartments which adds clarity and rigour. They have also fully engaged with the notion of maturation (as was also suggested by Reviewer 1), and have now added compelling data that this is indeed what is happening. Arf1 labelled compartments mature over time to be Rab11-positive recycling endosome compartments. This is a potentially very important finding that would provide a much-needed explanation for the relationship between the trans-Golgi network, the recycling endosome and early endosomes. The results are presented with clarity and with an admirable degree of balance by explaining what remains speculative and what is well proven.

Given all this I am very happy to recommend this excellent manuscript for publication and would like to commend again the authors for their open-minded approach and willingness to engage fully with reviewer comments.

Reviewer #3 (Remarks to the Author):

The authors present an improved revised manuscript with a new working model. They addressed all my initial comments either experimentally, or just verbally.

The current model for trafficking from the Golgi is that AP1/clathrin control export from the TGN. The conclusion of the revised manuscript (new model as the authors call it) is that AP1 is the factor responsible for cargo sequestration from the ARF1 compartment. I am not sure how this differs from the existing trafficking model. The ARF1-compartment is part of the TGN, and not an independent compartment. Thus, the new model plainly says that AP1 controls export from the TGN, which is not much different from the current model. I am therefore not sure what novelty this manuscript provides. The main finding of this manuscript is that ARF1 defines a domain of the TGN where exocytic cargo is segregated for export. How ARF1 performs this job remains unclear. It might alter the physical properties

of the membrane (curvature), or it might recruit lipid modifying enzymes, or it might recruit a fission machinery together with AP1 (as the authors claim themselves). However, none of these working hypotheses are tested against each other.

I really appreciate and love the technical excellence and elegance of this work. The authors also improved the data in the revised manuscript. Furthermore, I think that the discovery of ARF1 as a definer of something like an “export domain” at the TGN is interesting and important. Nevertheless, I think that the work remains descriptive and lacks mechanistic insight. Even if ARF1 were defining a genuine compartment, my initial concern that the current work does not define a physiological cargos, which are depend on this ARF1 compartment is not addressed.

We are glad the reviewer appreciates the technical excellence and elegance of our work. We believe that our work lays the foundations to understand the connection between the TGN and recycling endosomes and the mechanisms of recycling biogenesis. We provide evidence that maturation of compartments may be the driver of recycling endosomes biogenesis.

Final Decision Letter:

Dear Dr Bottanelli,

I am pleased to inform you that your manuscript, "ARF1 compartments direct cargo flow via maturation into recycling endosomes", has now been accepted for publication in Nature Cell Biology.

Publication is conditional on the manuscript not being published elsewhere and on there being no

announcement of this work to any media outlet until the online publication date in Nature Cell Biology.

You may wish to make your media relations office aware of your accepted publication, in case they consider it appropriate to organize some internal or external publicity. Once your paper has been scheduled you will receive an email confirming the publication details. This is normally 3-4 working days in advance of publication. If you need additional notice of the date and time of publication, please let the production team know when you receive the proof of your article to ensure there is sufficient time to coordinate. Further information on our embargo policies can be found here:

<https://www.nature.com/authors/policies/embargo.html>

Please note that *Nature Cell Biology* is a Transformative Journal (TJ). Authors may publish their research with us through the traditional subscription access route or make their paper immediately open access through payment of an article-processing charge (APC). Authors will not be required to make a final decision about access to their article until it has been accepted. Find out more about Transformative Journals

If you have not already done so, we strongly recommend that you upload the step-by-step protocols used in this manuscript to protocols.io (<https://protocols.io>), an open online resource that allows researchers to share their detailed experimental know-how. All uploaded protocols are made freely available and are assigned DOIs for ease of citation. Protocols and Nature Portfolio journal papers in

which they are used can be linked to one another, and this link is clearly and prominently visible in the online versions of both. Authors who performed the specific experiments can act as primary authors for the Protocol as they will be best placed to share the methodology details, but the Corresponding Author of the present research paper should be included as one of the authors. By uploading your Protocols onto protocols.io, you are enabling researchers to more readily reproduce or adapt the methodology you use, as well as increasing the visibility of your protocols and papers. You can also establish a dedicated workspace to collect your lab Protocols. Further information can be found at <https://www.protocols.io/help/publish-articles>.

With kind regards,

Melina Casadio, PhD
Senior Editor, Nature Cell Biology
ORCID ID: <https://orcid.org/0000-0003-2389-2243>

** Visit the Springer Nature Editorial and Publishing website at www.springernature.com/editorial-and-publishing-jobs for more information about our career opportunities. If you have any questions please click here.**